# Hierarchical Federated Learning with Multi-Timescale Gradient Correction

**Wenzhi Fang**
Purdue University
fang375@purdue.edu

**Dong-Jun Han**
Yonsei University
djh@yonsei.ac.kr

**Evan Chen**
Purdue University
chen4388@purdue.edu

**Shiqiang Wang**
IBM Research
wangshiq@us.ibm.com

**Christopher G. Brinton**
Purdue University
cgb@purdue.edu

## Abstract

While traditional federated learning (FL) typically focuses on a star topology where clients are directly connected to a central server, real-world distributed systems often exhibit hierarchical architectures. Hierarchical FL (HFL) has emerged as a promising solution to bridge this gap, leveraging aggregation points at multiple levels of the system. However, existing algorithms for HFL encounter challenges in dealing with *multi-timescale model drift*, i.e., model drift occurring across hierarchical levels of data heterogeneity. In this paper, we propose a multi-timescale gradient correction (MTGC) methodology to resolve this issue. Our key idea is to introduce distinct control variables to (i) correct the client gradient towards the group gradient, i.e., to reduce *client model drift* caused by local updates based on individual datasets, and (ii) correct the group gradient towards the global gradient, i.e., to reduce *group model drift* caused by FL over clients within the group. We analytically characterize the convergence behavior of MTGC under general non-convex settings, overcoming challenges associated with couplings between correction terms. We show that our convergence bound is immune to the extent of data heterogeneity, confirming the stability of the proposed algorithm against multi-level non-i.i.d. data. Through extensive experiments on various datasets and models, we validate the effectiveness of MTGC in diverse HFL settings. The code for this project is available at https://github.com/wenzhifang/MTGC.

## 1  Introduction

In the past several years, federated learning (FL) has emerged as a prevalent approach for distributed training [17, 22, 11, 52, 24]. Conventional FL has typically considered a star topology training architecture, where clients directly communicate with a central server for model synchronization [35, 26]. Scaling this architecture to large numbers of clients becomes problematic, however, given the heterogeneity in FL resource availability and dataset statistics manifesting over large geographies [14, 17, 10]. In practice, such communication networks are often comprised of a *hierarchical architecture* from clients to the main server, as observed in edge/fog computing [25, 38] and software-defined networks (SDN) [20], where devices are supported by intermediate edge servers that are in turn connected to the cloud.

To bridge this gap, researchers have proposed *hierarchical federated learning* (HFL) which integrates *group aggregations* into FL frameworks [30, 5, 47, 15]. In HFL (see Fig. 1), clients are segmented into multiple groups, and the training within each group is coordinated by a group aggregator node (e.g., an edge server coordinating a cell). Meanwhile, the central server orchestrates the training globally by periodically aggregating models across all client groups, facilitated by the group aggregators.

38th Conference on Neural Information Processing Systems (NeurIPS 2024).

**Fundamental challenges.** One of the key objectives in FL is to reduce communication overhead while maintaining model performance. Research in conventional FL has established how the global aggregation period, i.e., the number of local iterations during two consecutive communications between clients and the server, impacts FL performance according to the degree of non-i.i.d. (non-independent or non-identically distributed) across client datasets: when local datasets are more heterogeneous, longer aggregation periods cause client models to drift further apart. In HFL, the situation becomes more complex, and is not yet well studied. There are multiple levels of aggregations within/across client groups, and the frequency of these aggregations diminishes further up the hierarchy (since the communication costs become progressively more expensive). As a result, *model drift occurs across multiple levels of non-i.i.d., at different timescales*. In the canonical two-level case from Fig. 1, we have (i) intra-group non-i.i.d., similar to conventional FL, and (ii) inter-group non-i.i.d., arising from data heterogeneity across different groups. This introduces (i) *client model drift* caused by local updates on individual datasets, usually at a shorter timescale, as well as (ii) *group model drift* caused by FL over clients within the group, usually at a longer timescale.

In conventional star-topology FL, algorithms like ProxSkip [36], SCAFFOLD [18], and FedDyn [1] have shown promise for correcting client model drift through local regularization and gradient tracking/correction. However, *these approaches are not easily extendable to the HFL scenario due to its multi-timescale communication architecture*. Specifically, when integrating these methods into HFL, control variables introduced to handle data heterogeneity, such as gradient tracking or dynamic regularization, need to be carefully injected at each level of the hierarchy, taking into account their coupled effects in taming non-i.i.d. Convergence analysis elucidating the impact of different updating frequencies for such control variables remains an unsolved challenge. Existing works on HFL have also not aimed to directly correct for multi-timescale model drift. This can be seen by the fact that the convergence bounds in existing HFL methods [30, 5, 47, 13] become worse as the extent of non-i.i.d. in the system increases (e.g., gradient divergence between hierarchy levels in [47]). Some works have proposed adaptive control of the aggregation period in HFL [13, 31], but they require frequent model aggregations to prevent excessive drift. We thus pose the following question:

> *How can we tame multi-timescale model drift in non-i.i.d. hierarchical federated learning to provably enhance model convergence performance while not introducing frequent model aggregations?*

## 1.1 Contributions

In this paper, we propose *multi-timescale gradient correction* (MTGC), a methodology which can effectively address multi-level model drift over the topology of HFL with a theoretical guarantee. As depicted in Fig. 1, our key idea is to introduce coupled gradient correction terms – client-group correction and group-global correction – to (i) correct the client gradient towards the group gradient, i.e., to reduce client model drift caused by local updates based on their individual datasets, and (ii) correct the group gradient towards the global gradient, i.e., to reduce group model drift caused by

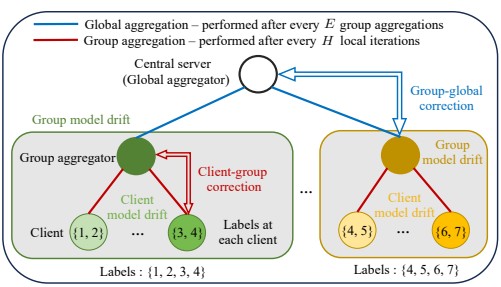

Figure 1: Illustration of multi-timescale gradient correction (MTGC) for multi-level non-i.i.d. in HFL.

FL across clients within the group, respectively. MTGC thus assists each client model to evolve towards improvements in global performance during HFL. We propose a strategy for updating these gradient correction terms after every group aggregation and global aggregation, respectively, and analyze the convergence behavior of MTGC. Due to the coupling of correction terms and their updates being performed at different timescales, additional challenges arise for theoretical analysis compared to prior work. We thoroughly investigate this problem and make the following contributions:

- We develop the multi-timescale gradient correction (MTGC) algorithm for taming leveled model drift in HFL. MTGC incorporates coupled control variables for correcting client gradients and group gradients, effectively tackling model biases arising from various levels of non-i.i.d. data at different timescales. The estimation and update procedures for these control variables rely solely on the model updates, ensuring that no significant additional communication overhead is introduced.

- We characterize the convergence rate for MTGC under the non-convex setup. This rate is immune to the extent of intra and inter-group data heterogeneity, confirming the stability of our approach against multi-level non-i.i.d. statistics. Our theoretical result also demonstrates that MTGC achieves

linear speedup in the number of local iterations, group aggregations, and clients. Also, we show that the convergence rate of MTGC recovers that of SCAFFOLD, i.e., the non-hierarchical case, when the number of groups and group aggregation period reduces to one.

- We conduct extensive experiments using various datasets and models across different parameter settings, which demonstrate the superiority of MTGC in diverse non-i.i.d. HFL environments.

### 1.2 Related Works

**Algorithms for conventional FL.** The seminal work [35] developed the FedAvg algorithm, incorporating multiple local updates into distributed SGD [58] to relieve communication bottlenecks within conventional star-topology FL. However, FedAvg convergence analysis makes assumptions such as bounded gradients [28, 56, 41] or bounded gradient dissimilarity [45, 12], showing it is not resistant to non-i.i.d. data. To tackle this issue, numerous techniques have been proposed in the literature, including incorporating static/dynamic regularizers [27, 1, 57, 16], adaptive control variables [29, 8, 51, 50], and/or gradient tracking methods [18, 32]. Despite these efforts, existing FL algorithms are not easily extendable to HFL due to the timescale mismatch of multi-level model aggregations induced by the hierarchical system topology. Optimizing these algorithms and ensuring their theoretical convergence in the presence of hierarchical model drift remains unsolved. Our paper addresses these issues through a principled multi-timescale gradient correction method.

**Hierarchical FL.** The authors of [5, 30, 13, 49, 55, 47] explored a new FL branch, HFL, tailored for hierarchical systems consisting of a central server, group aggregators, and clients. To tackle the issue of limited communication resources, the authors of [5] developed a FedAvg-like algorithm called hierarchical FedAvg tailored to HFL, and analyzed its convergence behavior. However, their algorithm is built upon an assumption of i.i.d. data. Another work [47] investigated the convergence behavior of hierarchical FedAvg under the non-i.i.d. setup. However, the convergence bound becomes worse as the extent of data heterogeneity increases, making the algorithm vulnerable to non-i.i.d. data characteristics. In [34], ProxSkip-HUB is introduced, but requires clients to compute full batch gradients and upload them to group aggregators after every iteration, which is impractical especially when training large-scale models. Overall, there is still a lack of an algorithm that fully addresses the unique challenge of HFL, i.e., the multi-timescale model drift problem, with theoretical guarantees. We fill this gap by introducing multi-timescale gradient correction and providing theoretical insights.

**Gradient tracking/correction.** Both gradient tracking and gradient correction aim to fix the local updating directions of clients to mitigate the impact of model drift caused by data heterogeneity. The gradient tracking concept was originally proposed and analyzed in [9] and then extended to consider various factors like time-vary graphs and asynchronous updates [37, 40, 42, 54]. Subsequently, SCAFFOLD [18] applied gradient tracking in FL to mitigate the impact of data heterogeneity across clients, ensuring convergence and stability in non-i.i.d. settings. More recently, in [32, 2], the authors demonstrate the effectiveness of gradient tracking in fully decentralized FL, where clients conduct model aggregations through local client-to-client communications. In [6, 43], gradient tracking is further studied in a semi-decentralized FL setup. Compared to all prior research, our work is the earliest attempt to design an algorithm specifically tailored to multi-timescale model drift in HFL and its training process with periodic local/global aggregations. This presents new challenges in our algorithm design and convergence analysis due to the coupling of our correction terms through their updates at different timescales. In Section 5, we empirically validate the effectiveness of our approach over the prior gradient correction method.

## 2 Background and Motivation

### 2.1 Problem Setup: Hierarchical FL

We consider the hierarchical system depicted in Fig. 1. The central server is connected to $N$ group aggregators, each linked to the clients within its region, defined as a group. Each group $j \in \{1, 2, \ldots, N\}$ consists of a set of $n_j$ non-overlapping clients, denoted $\mathcal{C}_j$, resulting in a total of $\sum_{j=1}^{N} n_j$ clients within the system. Each client $i$ has its own local data distribution $\mathcal{D}_i$. The goal of HFL is to construct an optimal global model $x^*$ considering the data distributions of all clients in the system. The role of each group aggregator $j$ involves coordinating the training for the $n_j$ clients within its region, while the central server orchestrates the training across all $N$ groups through

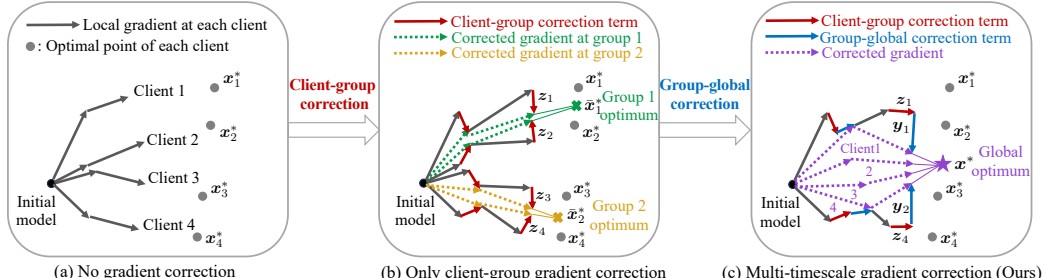

Figure 2: Visualization of the local update process using multi-timescale gradient correction (MTGC) with 4 clients and 2 groups. **(a)** Without any gradient correction (e.g., hierarchical FedAvg), each client model moves towards its respective optimal point, denoted by $\boldsymbol{x}_i^*$. **(b)** When only client-group correction term $\boldsymbol{z}_i$ is applied, the model of client $i \in \mathcal{C}_j$ moves towards the group optimum $\bar{\boldsymbol{x}}_j^*$. **(c)** In MTGC, the gradient of client $i \in \mathcal{C}_j$ is adjusted by both the client-group correction term $\boldsymbol{z}_i$ and the group-global correction variable $\boldsymbol{y}_j$, assisting each client model to converge towards the global optimum $\boldsymbol{x}^*$ during local iterations.

interaction with the group aggregators. We can formally state the HFL learning objective as follows:

$$\min_{\boldsymbol{x}} f(\boldsymbol{x}) := \frac{1}{N} \sum_{j=1}^{N} f_j(\boldsymbol{x}), \text{ where } f_j(\boldsymbol{x}) := \frac{1}{n_j} \sum_{i \in \mathcal{C}_j} F_i(\boldsymbol{x}) \text{ and } F_i(\boldsymbol{x}) := \mathbb{E}_{\xi_i \sim \mathcal{D}_i} \left[ F_i(\boldsymbol{x}, \xi_i) \right]. \quad (1)$$

Here, $f : \mathbb{R}^d \to \mathbb{R}$ denotes the global loss function, $f_j : \mathbb{R}^d \to \mathbb{R}$ is the loss specific to group $j$, and $F_i : \mathbb{R}^d \to \mathbb{R}$ represents the local loss for client $i$. In addition, $\xi_i$ is the data point sampled from distribution $\mathcal{D}_i$. Note that our analysis can be easily extended to a weighted average form of (1) by incorporating positive coefficients for each $f_j(\boldsymbol{x})$ or $F_i(\boldsymbol{x})$. For simplicity, these coefficients are assumed to be included in $F_i(\boldsymbol{x})$ as in previous works [18, 48].

## 2.2 Limitation of Existing Works

In HFL algorithms, group aggregations are conducted after every $H$ local client updates, while global aggregations are performed after every $E$ group aggregations, introducing different timescales. Moreover, different forms of data heterogeneity exist in HFL: (i) intra-group non-i.i.d., due to data heterogeneity across different clients $i \in \mathcal{C}_j$, and (ii) inter-group non-i.i.d., arising from data heterogeneity across different groups $\mathcal{C}_1, ..., \mathcal{C}_N$. These lead to *client model drift* and *group model drift*, respectively. The model drifts induced by multi-level data heterogeneity at different timescales hinder hierarchical FedAvg from converging. In Fig. 2(a), we see that during local training, each local model gradually converges towards the optimal point of its respective client's objective function. Hence, to guarantee theoretical convergence, existing HFL works either assume an i.i.d. setup [5] or rely on a bounded gradient dissimilarity assumption similar to the following [47, 15]:

$$\frac{1}{N} \sum_{j=1}^{N} \|\nabla f_j(\boldsymbol{x}) - \nabla f(\boldsymbol{x})\|^2 \le \delta_1^2, \forall \boldsymbol{x} \text{ and } \frac{1}{n_j} \sum_{i \in \mathcal{C}_j} \|\nabla F_i(\boldsymbol{x}) - \nabla f_j(\boldsymbol{x})\|^2 \le \delta_2^2, \forall \boldsymbol{x}, \forall j. \quad (2)$$

The first inequality is employed to limit the group drift, i.e., the deviation of group gradients from the global gradient, while the second one bounds the client drift, i.e., the divergence of client gradients from their group gradient. As a result, the convergence bounds of algorithms in these works become worse as data heterogeneity increases (i.e., as $\delta_1$ or $\delta_2$ increase) [19]. Our approach, developed next, does not require these assumptions and remains stable regardless of the extent of data heterogeneity.

## 3 Algorithm

### 3.1 Intuition: Gradient Correction in Hierarchical FL

When relying on multiple local SGD iterations as in hierarchical FedAvg, the model update process is not stable even when the model has reached the optimal $\boldsymbol{x}^*$ satisfying $\nabla f(\boldsymbol{x}^*) = \boldsymbol{0}$. Specifically, with $\gamma$ as the learning rate, we have $\boldsymbol{x}^* \neq \boldsymbol{x}^* - \gamma \nabla F_i(\boldsymbol{x}^*)$, as the global optimum $\boldsymbol{x}^*$ may not necessarily be optimal for each client's local loss due to data heterogeneity, i.e., $\nabla F_i(\boldsymbol{x}^*) \neq \boldsymbol{0}$ [39]. Correcting the client gradient $\nabla F_i(\boldsymbol{x}^*)$ to the global gradient $\nabla f(\boldsymbol{x}^*)$ is thus necessary to stabilize the process.[1]

**Motivation and idea.** In HFL, however, due to multi-level aggregations occurring at different timescales, it is infeasible to directly correct the client gradient to the global gradient. In particular,

---

[1]We motivate our approach here using full batch gradients, though our subsequent algorithm and analysis will support stochastic gradients.

---

**Algorithm 1:** HFL with Multi-Timescale Gradient Correction (MTGC)

---

**Input:** Initial model $\bar{\boldsymbol{x}}^0$, global aggregation period $E$, group aggregation period $H$, learning rate $\gamma$, and group-global correction $\boldsymbol{y}_j^0 = -\frac{1}{n_j}\sum_{i\in\mathcal{C}_j}\nabla F_i(\boldsymbol{x}_{i,0}^{t,0}, \xi_{i,0}^{0,0}) + \frac{1}{N}\sum_{j=1}^N \frac{1}{n_j}\sum_{i\in\mathcal{C}_j}\nabla F_i(\boldsymbol{x}_{i,0}^{0,0}, \xi_{i,0}^{0,0}), \forall j$

1   **each global round** $t = 0, 1, \ldots, T-1$ **do**

2     Group model initialization: $\boldsymbol{x}_j^{t,0} = \bar{\boldsymbol{x}}^t, \forall j$

3     Client-group correction initialization:
     $\boldsymbol{z}_i^{t,0} = -\nabla F_i(\boldsymbol{x}_{i,0}^{t,0}, \xi_{i,0}^{t,0}) + \frac{1}{n_j}\sum_{i\in\mathcal{C}_j}\nabla F_i(\boldsymbol{x}_{i,0}^{t,0}, \xi_{i,0}^{t,0}), \forall i\in\mathcal{C}_j, \forall j$

4     **each group communication round** $e = 0, 1, \ldots, E-1$ **do**

5       Local model initialization: $\boldsymbol{x}_{i,0}^{t,e} = \bar{\boldsymbol{x}}_j^{t,e}, \forall i, j$

6       **each local iteration** $h = 0, 1, \ldots, H-1$ **do**

7         $\boldsymbol{x}_{i,h+1}^{t,e} = \boldsymbol{x}_{i,h}^{t,e} - \gamma\left(\nabla F_i(\boldsymbol{x}_{i,h}^{t,e}, \xi_{i,h}^{t,e}) + \boldsymbol{z}_i^{t,e} + \boldsymbol{y}_j^t\right), \forall i\in\mathcal{C}_j, \forall j$      ⬦ Clients do in parallel

8       Group aggregation: $\bar{\boldsymbol{x}}_j^{t,e+1} = \frac{1}{n_j}\sum_{i\in\mathcal{C}_j}\boldsymbol{x}_{i,H}^{t,e}$

9       Client-group corr. update: $\boldsymbol{z}_i^{t,e+1} = \boldsymbol{z}_i^{t,e} + \frac{1}{H\gamma}\left(\boldsymbol{x}_{i,H}^{t,e} - \bar{\boldsymbol{x}}_j^{t,e+1}\right), \forall i\in\mathcal{C}_j, \forall j$ ⬦ Clients do in parallel

10    Global aggregation: $\bar{\boldsymbol{x}}^{t+1} = \frac{1}{N}\sum_{j=1}^N \bar{\boldsymbol{x}}_j^{t,E}$

11    Group-global corr. update: $\boldsymbol{y}_j^{t+1} = \boldsymbol{y}_j^t + \frac{1}{HE\gamma}\left(\bar{\boldsymbol{x}}_j^{t,E} - \bar{\boldsymbol{x}}^{t+1}\right), \forall j$      ⬦ Group aggregators in parallel

---

clients are not able to communicate with the central server directly, and there are multiple group aggregation steps before the group aggregators communicate with the main server. Our idea is thus to inject two gradient correction terms: client-group correction and group-global correction. Specifically, the desired iteration to obtain the updated model $\boldsymbol{x}_{\text{new}}$ at the optimal point $\boldsymbol{x}^*$ can be written as

$$\boldsymbol{x}_{\text{new}} = \boldsymbol{x}^* - \gamma\Big\{\nabla F_i(\boldsymbol{x}^*) + \underbrace{(\nabla f_j(\boldsymbol{x}^*) - \nabla F_i(\boldsymbol{x}^*))}_{\text{client-group correction}} + \underbrace{(\nabla f(\boldsymbol{x}^*) - \nabla f_j(\boldsymbol{x}^*))}_{\text{group-global correction}}\Big\}, \quad (3)$$

where $\nabla f_j(\boldsymbol{x}^*) - \nabla F_i(\boldsymbol{x}^*)$ and $\nabla f(\boldsymbol{x}^*) - \nabla f_j(\boldsymbol{x}^*)$ represent client-group and group-global correction terms, respectively. Since $\nabla f(\boldsymbol{x}^*) = \boldsymbol{0}$, the two correction terms will enable the model to remain at the optimal point. Given this intuition, the ideal local iteration at client $i \in \mathcal{C}_j$ can be written as

$$\boldsymbol{x}_{i,h+1}^{t,e} = \boldsymbol{x}_{i,h}^{t,e} - \gamma\Big\{\nabla F_i(\boldsymbol{x}_{i,h}^{t,e}) + \Big(\nabla f_j(\bar{\boldsymbol{x}}_{j,h}^{t,e}) - \nabla F_i(\boldsymbol{x}_{i,h}^{t,e})\Big) + \Big(\nabla f(\bar{\boldsymbol{x}}_h^{t,e}) - \nabla f_j(\bar{\boldsymbol{x}}_{j,h}^{t,e})\Big)\Big\}, \quad (4)$$

where $t$, $e$, and $h$ represent global communication rounds, group communication rounds, and client local iterations, respectively, $\bar{\boldsymbol{x}}_{j,h}^{t,e} = \frac{1}{n_j}\sum_{i\in\mathcal{C}_j}\boldsymbol{x}_{i,h}^{t,e}$ is the averaged model within group $j$, and $\bar{\boldsymbol{x}}_h^{t,e} = \frac{1}{N}\sum_{j=1}^N \frac{1}{n_j}\sum_{i\in\mathcal{C}_j}\boldsymbol{x}_{i,h}^{t,e}$ is the averaged model across the system. Based on (4), we expect to bring each client model closer to the global optima during local updates, as illustrated in Fig. 2(c).

**Challenge encountered in HFL.** However, it is important to note that the update process in (4) still cannot be directly used in HFL. This is because client-group communication and group-global communication do not occur at every iteration of HFL training; instead, they happen at different timescales, and clients are not able to obtain the current group information $\nabla f_j(\bar{\boldsymbol{x}}_{i,h}^{t,e})$ and global information $\nabla f(\bar{\boldsymbol{x}}_{i,h}^{t,e})$ at every local iteration. We next propose a strategy that mimics the gradient correction described above while ensuring theoretical convergence.

### 3.2 Multi-Timescale Gradient Correction (MTGC)

**Tackling multi-timescale model drifts.** To approximate (4) during HFL training, we introduce two control variables $\boldsymbol{z}$ and $\boldsymbol{y}$ that track/approximate $\nabla f_j - \nabla F_i$ and $\nabla f - \nabla f_j$, respectively. The variables $\boldsymbol{z}$ and $\boldsymbol{y}$ are then employed to correct the local gradients to prevent model drifts. The challenge here is to keep updating $\boldsymbol{z}$ and $\boldsymbol{y}$ appropriately in the multi-timescale communication scenario, given that communications between the clients and group aggregator, and between the group aggregators and global aggregator, are not always feasible. We propose a strategy to update $\boldsymbol{z}$ after every $H$ local iterations, i.e., whenever each client is able to communicate with the group aggregator, allowing the group information to be updated and shared among the clients within the same group. Similarly, we propose a strategy to update $\boldsymbol{y}$ after every $E$ group aggregations, i.e.,

whenever the group aggregators are able to communicate with the global aggregator, enabling the global information to be refreshed and shared across all clients in the system. We name our strategy multi-timescale gradient correction (MTGC) due to the updates of $z$ and $y$ occurring in different timescales, to tackle the issue of multi-level model drift coupled across the hierarchy in HFL.

In Fig. 2(c), we illustrate MTGC during client-side model updates. In particular, at each local iteration $h$ of group round $e$ of global round $t$, each client $i \in C_j$ updates its local model as follows:

$$\boldsymbol{x}_{i,h+1}^{t,e} = \boldsymbol{x}_{i,h}^{t,e} - \gamma \left( \nabla F_i(\boldsymbol{x}_{i,h}^{t,e}, \xi_{i,h}^{t,e}) + \boldsymbol{z}_i^{t,e} + \boldsymbol{y}_j^t \right). \tag{5}$$

**(i) Client-group correction term.** In (5), $\boldsymbol{z}_i^{t,e}$ is responsible for correcting the gradient of client $i \in C_j$ towards the gradient of group $j$ at the $e$-th group aggregation of global round $t$. After every group aggregation $e$ at global round $t$, this term is updated at each client $i$ as follows:

$$\boldsymbol{z}_i^{t,e+1} = \frac{1}{H} \sum_{h=0}^{H-1} \left( \left( \frac{1}{n_j} \sum_{i \in C_j} \nabla F_i(\boldsymbol{x}_{i,h}^{t,e}, \xi_{i,h}^{t,e}) \right) - \nabla F_i(\boldsymbol{x}_{i,h}^{t,e}, \xi_{i,h}^{t,e}) \right). \tag{6}$$

**(ii) Group-global correction term.** $\boldsymbol{y}_j^t$ in (5) aims to correct the gradient of group $j$ towards the global gradient. At the end of global round $t$, this term is updated at group aggregator $j$ as follows:

$$\boldsymbol{y}_j^{t+1} = \frac{1}{HE} \sum_{e=0}^{E-1} \sum_{h=0}^{H-1} \left( \left( \frac{1}{N} \sum_{j=1}^{N} \frac{1}{n_j} \sum_{i \in C_j} \nabla F_i(\boldsymbol{x}_{i,h}^{t,e}, \xi_{i,h}^{t,e}) \right) - \frac{1}{n_j} \sum_{i \in C_j} \nabla F_i(\boldsymbol{x}_{i,h}^{t,e}, \xi_{i,h}^{t,e}) \right). \tag{7}$$

**Key remarks.** The updating policies for $\boldsymbol{z}_i^{t,e}$ and $\boldsymbol{y}_j^t$ follow similar patterns to the ideal corrections outlined in (4). Here, we observe that $\sum_{i \in C_j} \boldsymbol{z}_i^{t,e} = \boldsymbol{0}, \forall j$ and $\sum_{j=1}^{N} \boldsymbol{y}_j^t = \boldsymbol{0}$, indicating that the correction terms do not have an impact on the per-iteration model averages. Instead, the introduction of $\boldsymbol{z}_i^{t,e}$ and $\boldsymbol{y}_j^t$ eliminates model drifts of clients and groups, respectively, during local iterations. Intuitively, as the iteration approaches the global optimal point, we expect $\boldsymbol{z}_i^{t,e} \to \nabla f_j(\boldsymbol{x}^*) - \nabla F_i(\boldsymbol{x}^*)$ and $\boldsymbol{y}_j^t \to \nabla f(\boldsymbol{x}^*) - \nabla f_j(\boldsymbol{x}^*)$ so that the update in (5) stabilizes at the global optimal point. We also see that $\boldsymbol{z}_i^{t,e}$ and $\boldsymbol{y}_j^t$ are coupled (5), i.e., the update of one of the terms affects $\boldsymbol{x}$ which in turn affects the other one, raising challenges for theoretical analysis. In Section 4, we will guarantee convergence of MTGC in general non-convex settings without relying on bounded data heterogeneity assumptions.

**MTGC algorithm.** The overall procedure of our training strategy is summarized in Algorithm 1, where we rewrite the updates of $\boldsymbol{z}_i^{t,e}$ and $\boldsymbol{y}_j^t$ in a different but equivalent manner to facilitate practical implementation of MTGC. Compared to hierarchical FedAvg, which does not consider any correction terms, we see that no additional communication is required for MTGC within each group round $e$. Additional communication is introduced only after $E$ group aggregations for initializing $\boldsymbol{z}_i^{t,0}$ (Line 4)[2] and broadcasting $\boldsymbol{y}_j^{t+1}$ (obtained in Line 14) to the clients in $C_j$. We will see in Section 5 that these marginal additional costs lead to significant performance enhancements for HFL settings.

**Generalization to arbitrary number of levels.** The proposed MTGC algorithm can be extended to an HFL system architecture with an arbitrary number of levels. Further discussions and experimental results for a three-level case are provided in Appendix E.

### 3.3 Connection with SCAFFOLD

When the number of groups reduces to $N = 1$ with $E = 1$, we have $\boldsymbol{y}_j^t = 0$ (no group-global correction), and thus MTGC reduces to SCAFFOLD [18]. In SCAFFOLD, at each round $t$, clients perform local updates according to $\boldsymbol{x}_{i,h+1}^t = \boldsymbol{x}_{i,h}^t - \gamma \left( \nabla F_i(\boldsymbol{x}_{i,h}^t, \xi_{i,h}^t) - \boldsymbol{c}_i^t + \boldsymbol{c}^t \right), h = 0, 1, \ldots, H-1,$ where $\boldsymbol{c}_i^{t+1} = \boldsymbol{c}_i^t - \boldsymbol{c}^t + \frac{1}{H\gamma} \left( \bar{\boldsymbol{x}}^t - \boldsymbol{x}_{i,H}^t \right)$, and the server aggregates local models and controlling variables as $\bar{\boldsymbol{x}}^{t+1} = \frac{1}{N} \sum_{i=1}^{N} \boldsymbol{x}_{i,H}^t$ and $\boldsymbol{c}^{t+1} = \frac{1}{N} \sum_{i=1}^{N} \boldsymbol{c}_i^{t+1}$. We can show that $\boldsymbol{c}_i^t - \boldsymbol{c}^t$ in SCAFFOLD plays the same role as $\boldsymbol{z}_i^{t,e}$ in MTGC. However, the additional term $\boldsymbol{y}_j^t$ introduced in MTGC for the multi-level setting makes the convergence guarantee more challenging, as $\boldsymbol{y}_j^t$ is coupled with $\boldsymbol{z}_i^{t,e}$ and both are updated at different time scales. These aspects will be thoroughly examined next.

## 4 Convergence Analysis

In this section, we establish a convergence guarantee for the proposed MTGC algorithm. Our theoretical analysis relies on the following standard assumptions commonly used in the literature on stochastic optimization and FL under non-convex settings [18, 44, 4].

---

[2]This is only required for theoretical analysis. In the experiments, we initialize $\boldsymbol{z}_i^{t,0} = \boldsymbol{0}, \forall i$.

**Assumption 1.** *Each local loss function $F_i$ is differentiable and $L$-smooth, i.e., there exists a positive constant $L$ such that for any $\boldsymbol{x}$ and $\boldsymbol{y}$, $\|\nabla F_i(\boldsymbol{x}) - \nabla F_i(\boldsymbol{y})\| \leq L\|\boldsymbol{y} - \boldsymbol{x}\|, \forall i$.*

**Assumption 2.** *The stochastic gradient $\nabla F_i(\boldsymbol{x}, \xi_i)$ is an unbiased estimate of the true gradient, i.e., $\mathbb{E}_{\xi_i \sim \mathcal{D}_i}[\nabla F_i(\boldsymbol{x}, \xi_i)] = \nabla F_i(\boldsymbol{x}), \forall \boldsymbol{x}$ and the variance of the stochastic gradient $\nabla F_i(\boldsymbol{x}, \xi_i)$ is uniformly bounded as $\mathbb{E}_{\xi_i \sim \mathcal{D}_i}\|\nabla F_i(\boldsymbol{x}, \xi_i) - \nabla F_i(\boldsymbol{x})\|^2 \leq \sigma^2, \forall \boldsymbol{x}$.*

Note that (i) global aggregation, (ii) the update of upper-level correction variable $\boldsymbol{y}$ and local aggregation, and (iii) the update of lower-level correction variable $\boldsymbol{z}$ are performed at different timescales in MTGC. If we directly consider $\{\nabla f(\bar{\boldsymbol{x}}^t)\}$ as in SCAFFOLD, it is difficult to capture the effects of group aggregation and correction variable $\boldsymbol{z}$. Moreover, it is hard to establish a tight connection between $\nabla f(\bar{\boldsymbol{x}}^t)$ and $\boldsymbol{x}_{i,h}^{t,e}$, $\forall i, h, \tau$ since there is a large lag between $\boldsymbol{x}_{i,h}^{t,e}$ and $\bar{\boldsymbol{x}}^t$. To tackle this, we introduce a new metric, which is the gradient $\nabla f(\hat{\boldsymbol{x}}^{t,e})$ at virtual sequence $\{\hat{\boldsymbol{x}}^{t,e} = \frac{1}{N}\sum_{j=1}^N \bar{\boldsymbol{x}}_j^{t,e}\}$, to characterize the convergence of MTGC.

We next state our main theoretical results. All the proofs are provided in Appendix F:

**Theorem 4.1.** *Suppose Assumptions 1 and 2 hold and the learning rate satisfies $\gamma \leq \frac{1}{40EHL}$. Then the iterates $\{\hat{\boldsymbol{x}}^{t,e}\}$ obtained by the MTGC algorithm satisfy*

$$\frac{1}{TE}\sum_{t=0}^{T-1}\sum_{e=0}^{E-1}\mathbb{E}\left\|\nabla f(\hat{\boldsymbol{x}}^{t,e})\right\|^2 = \mathcal{O}\left(\frac{f(\bar{\boldsymbol{x}}^0) - f^*}{\gamma TEH} + \frac{\gamma}{\tilde{N}}L\sigma^2 + \gamma^2 E^2 H^2 L^2 \sigma^2\right), \qquad (8)$$

*where $\tilde{N} = \left(\frac{1}{N^2}\sum_{j=1}^N \frac{1}{n_j}\right)^{-1}$, and $f^*$ is the lower bound of $f(\boldsymbol{x})$, i.e., $f(\boldsymbol{x}) \geq f^*$.*

There are two key steps in our proof. The first is the characterization of the evolution of $\left\|\boldsymbol{z}_i^{t,e} + \nabla F_i\left(\bar{\boldsymbol{x}}_j^{t,e}\right) - \nabla f_j\left(\bar{\boldsymbol{x}}_j^{t,e}\right)\right\|^2$ and $\left\|\boldsymbol{y}_j^t + \nabla f_j\left(\hat{\boldsymbol{x}}^{t,e}\right) - \nabla f\left(\hat{\boldsymbol{x}}^{t,e}\right)\right\|^2$. By bounding these values that capture the error between each control variable and the ideal correction, we are able to establish a connection between the local updating direction and the global gradient without relying on the bounded gradient dissimilarity assumption, laying the foundation for the whole proof. The second is that we extracted a recursive relationship for the accumulation of group-level and client-level model drifts, and designed a novel Lyapunov function to mitigate the interplay impact between these drifts. Further details are provided in the appendix.

Applying an appropriate learning rate $\gamma$ to Algorithm 1 yields the following corollary:

**Corollary 4.1.** *Under the assumptions of Theorem 4.1, let $\mathcal{F}_0 = f(\bar{\boldsymbol{x}}^0) - f^*$. Then there exists a learning rate $\gamma \leq \frac{1}{40EHL}$ such that the iterates $\{\hat{\boldsymbol{x}}^{t,e}\}$ satisfy*

$$\frac{1}{TE}\sum_{t=0}^{T-1}\sum_{e=0}^{E-1}\mathbb{E}\left\|\nabla f(\hat{\boldsymbol{x}}^{t,e})\right\|^2 \leq \mathcal{O}\left(\sqrt{\frac{\mathcal{F}_0 L\sigma^2}{\tilde{N}TEH}} + \left(\frac{\mathcal{F}_0 L\sigma}{T}\right)^{\frac{2}{3}} + \frac{L\mathcal{F}_0}{T}\right). \qquad (9)$$

**Discussions.** Corollary 4.1 provides the convergence upper bound of the MTGC algorithm. It shows that the error approaches zero as $T \to \infty$. If $\sigma \neq 0$, the upper bound is dominated by the first term in the right-hand side of (9), which characterizes the speed of convergence of MTGC to a stationary point in the stochastic case. This reveals MTGC achieves linear speedup in the number of group aggregations $E$ and local updates $H$. In other words, we can attain the same level of performance with less global communication rounds, i.e., a smaller value of $T$, by increasing the number of local iterations, i.e., $H$, and group aggregations, i.e., $E$. When considering the special case $n_{j'} = n$, $\forall j' \in \{1, 2, \ldots, N\}$ with uniform client numbers, the rate becomes $\mathcal{O}(\sqrt{\mathcal{F}_0 L\sigma^2}/\sqrt{NnTEH})$. This implies that MTGC attains linear speedup in the number of clients as well.

Moreover, we also see that our convergence rate recovers the results of SCAFFOLD when the number of groups reduces to $N = 1$ and the number of group aggregations reduces to $E = 1$ (see Appendix G for more discussions). We also highlight that, different from prior works on HFL where the convergence bound becomes worse as the extent of data heterogeneity increases, our bound is stable against multi-level non-i.i.d. data due to the multi-timescale gradient correction approach.

## 5 Experimental Results

### 5.1 Setup

**Dataset, model, hyperparameters, and compute setting.** In our experiments, we consider four widely used datasets: EMNIST-Letters (EMNIST-L) [7], Fashion-MNIST [53], CIFAR-10 [23], and

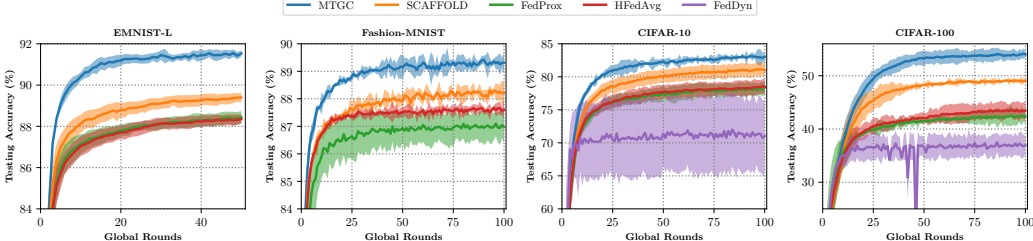

Figure 3: **Comparison with FL baselines.** In this experiment, popular FL algorithms are extended to the HFL setup for comparison with MTGC. We consider four datasets in the group non-i.i.d. & client non-i.i.d. setting. Experiments are conducted over 3 random trials. We see that MTGC obtains the best testing accuracy in each case, validating our multi-level approach for correcting multi-timescale model drifts.

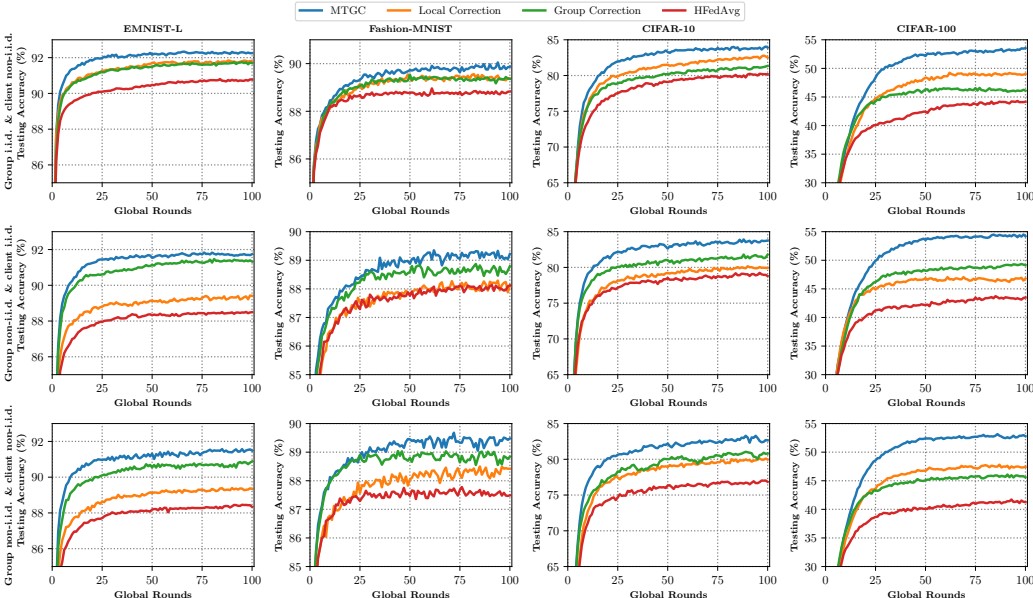

Figure 4: **Comparison with gradient correction baselines.** Three different data distribution scenarios are considered. We see that the local correction method is effective for handling client non-i.i.d. within each group (top row), while the group correction method is effective for handling non-i.i.d. across groups (middle row). MTGC obtains the most stable performance (all rows) by combining multiple correction levels.

CIFAR-100 [23]. The former two are processed through a multi-layer perceptron (MLP) model, featuring two hidden layers, each comprising 200 neurons, and ending with a softmax layer. For the CIFAR-10 classification task, we employ a convolutional neural network (CNN) following the architecture outlined in seminal work [35]. For CIFAR-100, we adopt a ResNet-18 model with batch normalization layers substituted by group normalization layers. Across all algorithms considered, we maintain a consistent learning rate $\eta = 0.1$ and batch size 50. We conduct the experiments based on a cluster of 3 NVIDIA A100 GPUs with 40 GB memory. Our code is based on the framework of [1].

**FL data distribution.** We set the total number of clients as 100, evenly distributed over $N = 10$ groups. We also study the effect of $N$ in Appendix B. We consider three different data distribution settings: (i) group i.i.d. & client non-i.i.d., (ii) group non-i.i.d. & client i.i.d., and (iii) group non-i.i.d. & client non-i.i.d. scenarios. First, in the group i.i.d. & client non-i.i.d. case, the training dataset is initially divided uniformly and randomly into $N$ segments corresponding to $N$ groups. Subsequently, each segment is further divided into $100/N$ partitions for the clients using a Dirichlet distribution [1]. Second, in the group non-i.i.d. & client i.i.d. case, the dataset is first segmented into $N$ partitions for the groups using a Dirichlet distribution, followed by a uniform random distribution of each segment to $100/N$ clients. Finally, when both groups and clients are non-i.i.d., the dataset is split into $N$ segments for the groups using a Dirichlet distribution, and then, each group's segment is distributed among $100/N$ clients through a Dirichlet distribution. The Dirichlet parameter is set to 0.1.

Table 5.1: The *number of global rounds* required by different algorithms to attain the testing accuracy of $80\%$ for CIFAR-10 under different settings. Taking HFedAvg as the benchmark, we show the speedup achieved by MTGC and other baselines as we vary aggregation periods $E$ and $H$. MTGC consistently outperforms baselines, and the speedup gets more significant as $E$ and $H$ increase. Standard deviation is based on 3 random trials.

| Data distribution | Params $(E, H)$ | HFedAvg | Local Correction | Group Correction | MTGC |
|---|---|---|---|---|---|
| Group i.i.d. & client non-i.i.d. | (10, 20) | $144.3\pm3.4$ $(1\times)$ | $57.0\pm0.8$ $(2.5\times)$ | $72.0\pm1.6$ $(2.0\times)$ | $\mathbf{45.7}\pm0.9$ $(3.2\times)$ |
| | (10, 30) | $169.0\pm2.4$ $(1\times)$ | $51.0\pm1.4$ $(3.3\times)$ | $81.3\pm1.2$ $(2.1\times)$ | $\mathbf{37.3}\pm1.9$ $(4.5\times)$ |
| | (10, 40) | $214.0\pm5.9$ $(1\times)$ | $45.3\pm1.2$ $(4.7\times)$ | $85.7\pm1.2$ $(2.5\times)$ | $\mathbf{32.0}\pm0.8$ $(6.7\times)$ |
| | (10, 20) | $144.3\pm3.4$ $(1\times)$ | $57.0\pm0.8$ $(2.5\times)$ | $72.0\pm1.6$ $(2.0\times)$ | $\mathbf{45.7}\pm0.9$ $(3.2\times)$ |
| | (20, 20) | $105.0\pm4.1$ $(1\times)$ | $34.0\pm0.8$ $(3.0\times)$ | $53.0\pm0.8$ $(2.0\times)$ | $\mathbf{22.3}\pm0.9$ $(4.7\times)$ |
| | (30, 20) | $82.7\pm2.1$ $(1\times)$ | $25.7\pm0.9$ $(3.2\times)$ | $44.7\pm0.5$ $(1.8\times)$ | $\mathbf{16.3}\pm1.2$ $(5.1\times)$ |
| Group non-i.i.d. & client i.i.d. | (10, 20) | $246.0\pm3.7$ $(1\times)$ | $92.3\pm1.7$ $(2.7\times)$ | $53.7\pm1.2$ $(4.6\times)$ | $\mathbf{37.7}\pm0.5$ $(6.5\times)$ |
| | (10, 30) | $302.7\pm4.9$ $(1\times)$ | $88.0\pm1.6$ $(3.4\times)$ | $44.3\pm1.2$ $(6.8\times)$ | $\mathbf{27.3}\pm1.2$ $(11.1\times)$ |
| | (10, 40) | $320.0\pm2.4$ $(1\times)$ | $94.7\pm1.2$ $(3.5\times)$ | $43.0\pm1.6$ $(7.2\times)$ | $\mathbf{21.3}\pm1.2$ $(15.4\times)$ |
| | (10, 20) | $246.0\pm3.7$ $(1\times)$ | $92.3\pm1.7$ $(2.7\times)$ | $53.7\pm1.2$ $(4.6\times)$ | $\mathbf{37.7}\pm0.5$ $(6.5\times)$ |
| | (20, 20) | $308.7\pm3.3$ $(1\times)$ | $74.3\pm1.7$ $(4.2\times)$ | $31.0\pm0.8$ $(10.0\times)$ | $\mathbf{18.7}\pm1.7$ $(16.5\times)$ |
| | (30, 20) | $344.7\pm4.6$ $(1\times)$ | $85.7\pm2.1$ $(4.0\times)$ | $25.7\pm1.7$ $(13.4\times)$ | $\mathbf{13.0}\pm0.8$ $(26.5\times)$ |
| Group non-i.i.d. & client non-i.i.d. | (10, 20) | $363.0\pm7.3$ $(1\times)$ | $141.7\pm2.9$ $(2.6\times)$ | $83.7\pm1.2$ $(4.3\times)$ | $\mathbf{52.7}\pm1.2$ $(6.7\times)$ |
| | (10, 30) | $>500$ $(1\times)$ | $127.7\pm2.9$ $(>3.9\times)$ | $79.7\pm1.7$ $(>6.3\times)$ | $\mathbf{38.7}\pm0.9$ $(>12.9\times)$ |
| | (10, 40) | $>500$ $(1\times)$ | $169.7\pm5.2$ $(>2.9\times)$ | $106.3\pm1.9$ $(>4.7\times)$ | $\mathbf{31.7}\pm0.5$ $(>15.8\times)$ |
| | (10, 20) | $363.0\pm7.3$ $(1\times)$ | $141.7\pm2.9$ $(2.6\times)$ | $83.7\pm1.2$ $(4.3\times)$ | $\mathbf{52.7}\pm1.2$ $(6.7\times)$ |
| | (20, 20) | $>500$ $(1\times)$ | $113.3\pm3.4$ $(>4.4\times)$ | $45.3\pm1.2$ $(>11.0\times)$ | $\mathbf{25.0}\pm1.6$ $(>20.0\times)$ |
| | (30, 20) | $>500$ $(1\times)$ | $86.7\pm2.4$ $(>5.8\times)$ | $50.7\pm2.4$ $(>9.9\times)$ | $\mathbf{19.6}\pm0.5$ $(>25.5\times)$ |

## 5.2    Results and Discussion

**Comparison with conventional FL algorithms.** For comparison, we first apply the well-known FL methods, FedProx [27], SCAFFOLD [18], and FedDyn [1], to HFL, by running their training algorithms within each group of the hierarchical system. We also consider HFedAvg [47] as a baseline. Fig. 3 compares MTGC with these baselines in the group non-i.i.d. & client non-i.i.d. case. We observe that MTGC outperforms all the considered conventional algorithms, achieving the highest testing accuracy, especially for the complicated CIFAR-100 dataset. FedDyn achieves the lowest performance, demonstrating significant variance and instability. The significant performance gap between MTGC and FedDyn, in particular, can be attributed to the hierarchical setup disrupting the special structure of FedDyn. This result reveals that some algorithms designed for the conventional star-topology FL may be non-trivial to be extended to hierarchical setups. The overall results confirm the effectiveness of our approach that effectively tackles the multi-timescale drift problem in HFL.

**Comparison with gradient correction baselines.** In Fig. 4, we compare MTGC with the gradient correction baselines. Specifically, we apply local correction $(z_i^{t,e})$ to HFedAvg, and group correction $(y_j^t)$ to HFedAvg. These baselines can be viewed as schemes applying SCAFFOLD [18] within each group and across groups, respectively. We also report the results of the original HFedAvg to see the effects of gradient correction clearly. We make the following key observations. First, the testing accuracy achieved by HFedAvg decreases as the extent of data heterogeneity increases, e.g., from the first or second row to the third row in Fig. 4. This shows that data heterogeneity hinders the convergence of HFedAvg. Second, with the assistance of local or group correction, the algorithm attains a higher accuracy. In the case of group i.i.d. & client non-i.i.d., HFedAvg augmented with client local correction performs better than the variant with group correction. Conversely, in the scenario where groups are non-i.i.d. and clients are i.i.d., the opposite holds. This can be explained by the dominance of data heterogeneity in each case. In the former scenario, because the heterogeneity is primarily at the client-level, local client correction becomes more beneficial. On the other hand, in the latter scenario, where the heterogeneity shifts to the group level, group correction becomes more advantageous. Finally, we see that MTGC consistently outperforms baselines under all settings, where the performance gains brought by the multi-timescale gradient correction become more significant when it comes to the group non-i.i.d.& client non-i.i.d. case.

**Speedup in $H$ and $E$.** In Table 5.1, we investigate the effects $H$ and $E$, which determine the periods of group aggregation and global aggregation in HFL. We report the number of global rounds required to attain the desired testing accuracy of $80\%$ for CIFAR-10 under different settings. We have the following observations: As $E$ or $H$ increases, the required number of global rounds of MTGC for achieving the desired accuracy decreases. This demonstrates the speedup of the proposed algorithm in the number of local iterations and group aggregations, which fits well with our theory discussed in Section 4. In addition, the speedup achieved by MTGC compared to HFedAvg gets more significant

as $E$ or $H$ increases. For instance, in the group i.i.d. & client non-i.i.d. case, MTGC attains $3.3\times$ speedup when $E = 10$ , $H = 20$, which increases to $4.7\times$ when $E = 10$ , $H = 40$. This reveals that MTGC utilizes local iterations better compared with the baselines.

**Impact of data heterogeneity.** Consistent with the results in Fig. 4, we see from Table 5.1 that the required number of global rounds of HFedAvg increases as data heterogeneity increases, while MTGC is more stable against non-i.i.d. data. The gain of MTGC over HFedAvg becomes evident as data heterogeneity increases, confirming the effectiveness of our multi-timescale gradient correction approach for addressing the unique challenges of HFL.

**Further experiments.** Additional experimental results including the impacts of hierarchical system parameters and the performance in 3-level HFL are provided in Appendices B and E.

## 6    Conclusion and Limitation

We have proposed MTGC, a multi-timescale gradient correction approach for HFL. Embedded with control variables updated in different timescales, MTGC effectively corrects gradient biases and alleviates both client model drift and group model drift in hierarchical setups. We established the convergence bound of MTGC in the non-convex setup and showed its stability against multi-level data heterogeneity. Finally, we confirmed the advantage of our MTGC through extensive experiments in different non-i.i.d. HFL settings. A limitation of our work is that despite providing experiments for HFL systems with more than two levels (in Appendix E), our convergence analysis focused on the two-level case, which provides an interesting future direction of investigation.

## Acknowledgments

This work was supported by the National Science Foundation (NSF) under grants CNS-2146171 and CPS-2313109, and by the Air Force Office of Scientific Research (AFOSR) under grant FA9550-24-1-0083.

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

# Appendix

## A  Connection Between HFL and Cluster FL

Our work focuses on HFL, employing a multi-layered structure consisting of local nodes, local aggregators, and a central server. Both clustered FL and HFL aim to improve FL learning efficiency by leveraging structured client groupings. The difference between them lies in the grouping criteria. HFL focuses on collaborative training over a given network topology, where clients are generally grouped based on their geographical location or network connection status, and aims to build a single global model under this setting. CFL groups clients to optimize model training, with different global models constructed depending on the group. [33] demonstrates how dynamic clustering based on data distributions can enhance model performance. [3] explores alleviating negative transfer from collaboration by clustering clients into non-overlapping coalitions based on their distribution distances and data quantities.

## B  Additional Experiments on CIFAR-10

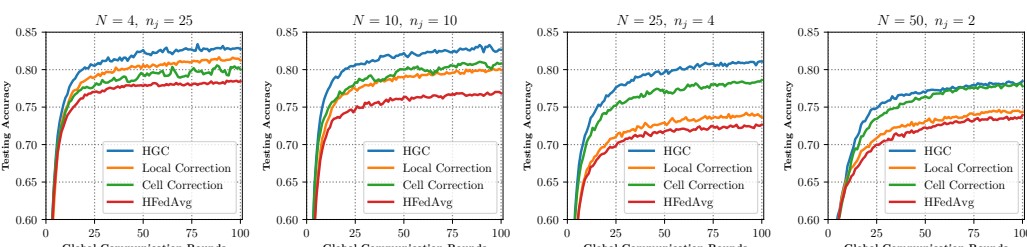

Figure 5: Comparison of testing accuracy versus global communication round across different system parameters under both group non-i.i.d. and client non-i.i.d. setup. $E$ and $H$ are set to 30 and 20, respectively.

**The impact of system parameters.** Fig. 5 shows how the performance of MTGC changes with different numbers of groups and clients in each group. From this figure, we observe that as the number of clients in each group, i.e., $n_j$ increases, client correction becomes more important. On the other hand, as the number of groups increases, the algorithm with group correction performs better than that with client correction.

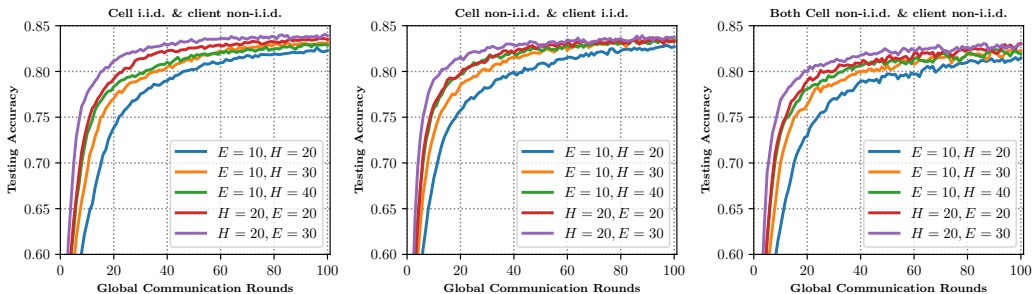

Figure 6: Performance of MTGC under a different number of local iterations, i.e., $H$, and a different number of group aggregations, i.e., $E$. The number of groups and clients in each group is set to $N = 10$ and $n_j = 10$, respectively.

**The impact of local iteration and group aggregation:** Fig. 6 depicts the performance of MTGC under a different number of local iterations, i.e., $H$, and a different number of group aggregations, i.e., $E$. It's clear that MTGC achieves speedup in the number of local iterations and group aggregations.

**Communication cost comparison.** Compared to HFedAvg, MTGC requires initializing the correction variables at the start of each global round, which adds additionaly communication overhead. Specifically, for every $E$ steps of group aggregation, MTGC incurs an additional communication cost equivalent to one transmission of the model parameters. In other words, the per-aggregation communication complexity of MTGC is $\frac{E+1}{E}$ times that of HFedAvg. To show this impact, we have added experiments comparing the communication cost and testing accuracy at the client side. This

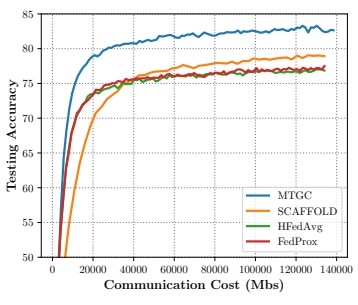
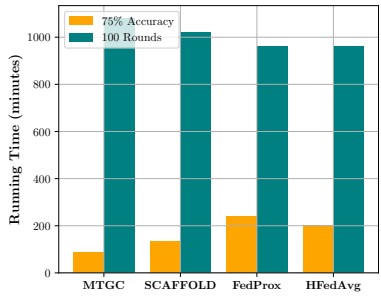

(a) Testing accuracy versus communication cost

(b) Running time comparison

Figure 7: Comparison of communication cost (a); and running time for attaining $75\%$ testing accuracy and finishing 100 global rounds on CIFAR-10 (b)

experiment was conducted on CIFAR-10 dataset with $E = 30$ and $H = 20$ under both client and group non-i.i.d. setup. The model and other parameters are the same as in the original manuscript. The results are shown in Fig. 7a. The results demonstrate that MTGC achieves higher testing accuracy for a given communication cost, highlighting the efficiency and effectiveness of our approach.

**Running time comparison.** We compared the computation time of our MTGC algorithm with the baselines. Using NVIDIA A100 GPUs with 40 GB memory, we conducted experiments on the CIFAR-10 dataset with $E = 30$ and $H = 20$ under both client and group non-i.i.d. setup. The model and other parameters are the same as in the original manuscript. We report the required time for attaining a preset accuracy of $75\%$ and for running 100 global rounds in Fig. 7b of the attached pdf. The speedup in the convergence makes up for the introduced computation cost per iteration due to the extra operation induced by the correction variables. Actually, the computation cost incurred by the correction variable is relatively small compared to computing gradients in a neural network using backpropagation.

## C  Experiments on Distribution Shift Datasets

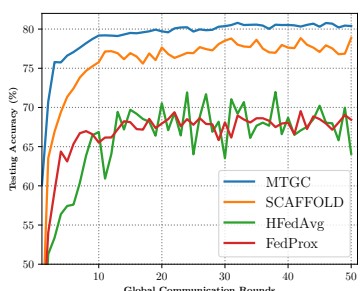
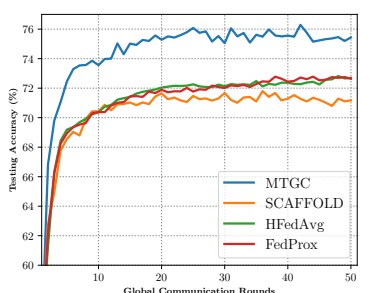

(a) Performance comparisons under label shift on Fashion-MNIST

(b) Performance comparisons under feature shift on Fashion-MNIST

Figure 8: Performance comparisons on Fashion-MNIST under label shift and Fashion-MNIST under feature shift

To further show the robustness, we studied the performance of MTGC under another two different non-i.i.d. scenarios: label shift and feature shift, as referenced in [3, 33]. These experiments were performed using the Fashion-MNIST dataset.

For label shift [3, 33], we randomly assign 3 classes out of 10 classes to each group with a relatively balanced number of instances per class, and then assign 2 classes to each client. As discussed in [3], label shift adds more heterogeneity to this system. According to the results shown in Fig. 8a, it is clear that the proposed algorithm is more robust against data heterogeneity. Specifically, there is less oscillation in MTGC compared with HFedAvg and the attained accuracy of MTGC in the given communication round is higher than all baselines.

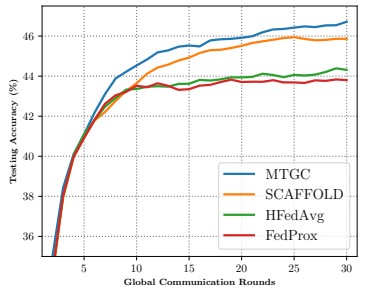 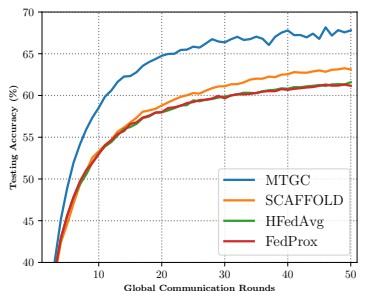

(a) Performance comparison on the Shakespeare dataset based on LSTM model

(b) Performance comparison under CINIC-10 dataset

Figure 9: Performance evaluation on Shakespeare and CINIC-10

For feature shift [3], we first partition data following the group non-i.i.d. & client non-i.i.d. case as in our original manuscript, and then let clients at different groups rotate images for different angles. Concretely, for the clients at the $i$-th group, the angle is $-50 + 10 \times i$. Note that this rotation is only applied to the training set. The feature shift increases the diversity between the training set and the testing set, which thus adds difficulty to this classification task. In Fig. 8b, we see that MTGC attains the best performance among these baselines.

## D Additional Experiments on CINIC-10 and Shakespear Datasets

We conducted additional experiments on the larger Shakespeare and CINIC-10 datasets. For the **Shakespeare dataset**, we randomly pick 100 characters (people) in Shakespeare's plays. We let each client have 1,500 samples, where each sample is a sequence of 80 characters (words). Considering that there are 100 clients in the system, there are 150,000 train samples in total. This means that the number of samples is 3 times that of CIFAR-10 (or CIFAR-100), which has 50,000 train samples. The performance comparison is presented in Fig. 9a, where we use the LSTM model, the same as [1], and set the learning rate 0.5, $H = 75$, and $E = 30$. It is seen that MTGC consistently outperforms the baseline methods in larger datasets.

The **CINIC-10 dataset** contains 90,000 training images, 90,000 validation images, and 90,000 test images, significantly larger than CIFAR-10 and CIFAR-100 with 60,000 images. It includes images from both CIFAR-10 and ImageNet, enhancing diversity. We believe that the larger size and diversity of CINIC-10 further confirm the validity of our experiments. The model and hyperparameters used for the CINIC-10 dataset are the same as those of the CIFAR-10 task shown in the original manuscript. As illustrated in Fig. 9b, MTGC maintains its superior performance on the CINIC-10 dataset, consistent with its performance on other tasks.

## E Extension to the HFL System with Arbitrary Number of Levels

We extend MTGC for the HFL system with $M$ levels in this subsection. For presentation ease, we adopt different notations than those used in the main text. Specifically, we denote the number of total iterations at clients as $r$. The aggregation periods for level $m$ are denoted as $P_m$. This means that the $m$-th level aggregator aggregates the model from the clients within its coverage after every $P_m$ local iterations. The global server is treated as the first level aggregator. Note that $P_m > P_{m+1}$ and $P_{m+1} \mid P_m, \forall m = 1, \ldots, M - 1$. We denote the model maintained at the nodes connected to the $m$-th level aggregator $(k_1, k_2, \ldots, k_{m-1})$ as $\boldsymbol{x}_{k_1,\ldots,k_m}^r$, where $k_m \in \{1, \ldots, N_m\}$. The gradient correction term between nodes $(k_1, k_2, \ldots, k_{m-1})$ and $(k_1, k_2, \ldots, k_m)$ is denoted as $\boldsymbol{\nu}_{k_1,k_2,\ldots,k_m}^r$. The overall procedures are summarized in Algorithm 2.

**Algorithm 2:** MTGC for the HFL System with Arbitrary Number of Levels

**Input:** $\gamma, \{P_i : i \in \{1, 2, \ldots, M\}\}$

1 **Initialize:** $\boldsymbol{\nu}_{k_1}^r, \boldsymbol{\nu}_{k_1,k_2}^r, \ldots, \boldsymbol{\nu}_{k_1,k_2,\ldots,k_M}^r, \forall k_1, k_2, \ldots, k_M$

2    $r = 0, 1, \ldots, R - 1$ **do**

3      **each client** $(k_1, \ldots, k_M) \in \mathcal{V}$ **in parallel do**

4        Compute stochastic gradient: $\boldsymbol{g}_{k_1,\ldots,k_M}^r$

5        $\boldsymbol{x}_{k_1,\ldots,k_M}^{r+1} = \boldsymbol{x}_{k_1,\ldots,k_M}^r - \gamma\left(\boldsymbol{g}_{k_1,\ldots,k_M}^r + \boldsymbol{\nu}_{k_1}^r + \boldsymbol{\nu}_{k_1,k_2}^r + \ldots + \boldsymbol{\nu}_{k_1,k_2,\ldots,k_M}^r\right)$

6      **level** $i = M, \ldots, 1$ **do**

7        **if** $P_i \mid r + 1$ **then**

8          $i$-th level model aggregate:

           $\boldsymbol{x}_{k_1,\ldots,k_i}^{r+1} = \frac{1}{N_i \ldots N_M} \sum_{k_i=1}^{N_i} \cdots \sum_{k_M=1}^{N_M} \boldsymbol{x}_{k_i,\ldots,k_M}^{r+1}, \forall k_1, \ldots, k_i$

9          $i$-th level correction term update:

           $\boldsymbol{\nu}_{k_1,k_2,\ldots,k_i}^{r+1} = \boldsymbol{\nu}_{k_1,k_2,\ldots,k_i}^r + \frac{1}{\gamma P_i}\left(\boldsymbol{x}_{k_1,\ldots,k_i}^{r+1} - \boldsymbol{x}_{k_1,\ldots,k_i}^t\right)$

10          Model dissemination: $\boldsymbol{x}_{k_1,\ldots,k_M}^{r+1} \leftarrow \boldsymbol{x}_{k_1,\ldots,k_i}^{r+1}, \forall k_1, \ldots, k_M$

11          Initialize $\boldsymbol{\nu}_{k_1,k_2,\ldots,k_{i+1}}^r, \ldots, \boldsymbol{\nu}_{k_1,k_2,\ldots,k_M}^r, \forall k_1, k_2, \ldots, k_M$

12        **else**

13          $\boldsymbol{\nu}_{k_1}^{r+1} = \boldsymbol{\nu}_{k_1}^r, \boldsymbol{\nu}_{k_1,k_2}^{r+1} = \boldsymbol{\nu}_{k_1,k_2}^r, \ldots, \boldsymbol{\nu}_{k_1,k_2,\ldots,k_i}^{r+1} = \boldsymbol{\nu}_{k_1,k_2,\ldots,k_i}^r, \forall k_1, k_2, \ldots, k_i$

14          **break**

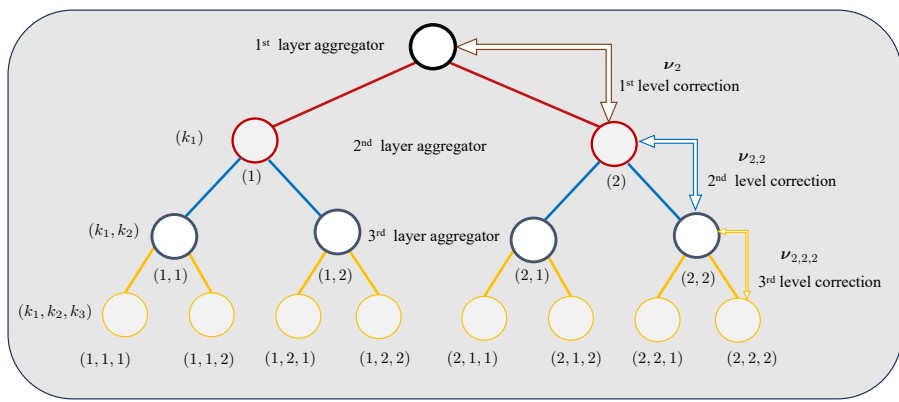

Figure 10: HFL system with 3-level topology

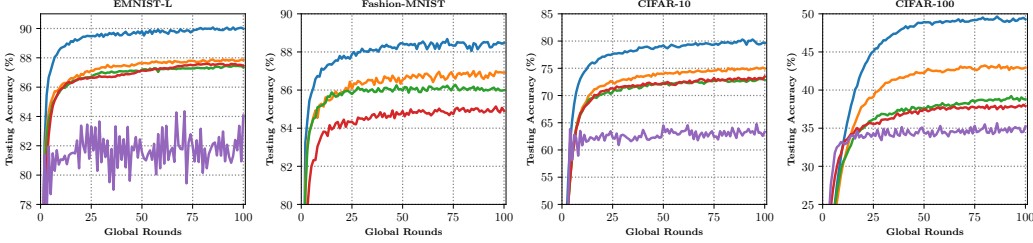

Figure 11: Performance of MTGC for three-level HFL with data non-i.i.d. across each level. Parameters are set to $N_1 = 4$, $N_2 = 5$, $N_3 = 5$, $P_1 = 500$, $P_2 = 100$, $P_3 = 10$.

We numerically validate the performance of MTGC by conducting experiments in the three-level case as shown in Fig. 10. The results are shown in Fig. 11. The total number of clients is set to be 100 while $N_1 = 4$, $N_2 = 5$, $N_3 = 5$. Additionally, the aggregation periods are set to be $P_1 = 500$, $P_2 = 100$, $P_3 = 10$. The data is non-i.i.d. distributed across each level.

# F  Proof of the Main Results

## F.1  Preliminaries

Before proceeding to the proof of the main theorem. We introduce some basic inequalities in this subsection that will be frequently used in our proof.

**Lemma F.1.1.** *For any set of $K$ vectors $\{\boldsymbol{p}_k\}_{k=1}^K$, $\left\|\frac{1}{K}\sum_{k=1}^K \boldsymbol{p}_k\right\|^2 \leq \frac{1}{K}\sum_{k=1}^K \|\boldsymbol{p}_k\|^2$, $\left\|\sum_{k=1}^K \boldsymbol{p}_k\right\|^2 \leq K\sum_{k=1}^K \|\boldsymbol{p}_k\|^2$, and*

$$\frac{1}{K}\sum_{k=1}^K \left\|\boldsymbol{p}_k - \frac{1}{K}\sum_{k=1}^K \boldsymbol{p}_k\right\|^2 = \frac{1}{K}\sum_{k=1}^K \|\boldsymbol{p}_k\|^2 - \left\|\frac{1}{K}\sum_{k=1}^K \boldsymbol{p}_k\right\|^2.$$

**Lemma F.1.2.** *For any two vectors $\boldsymbol{p}, \boldsymbol{q} \in \mathbb{R}^d$, $\|\boldsymbol{p}+\boldsymbol{q}\|^2 \leq (1+\alpha)\|\boldsymbol{p}\|^2 + \left(1+\frac{1}{\alpha}\right)\|\boldsymbol{q}\|^2$.*

**Lemma F.1.3.** *Suppose a sequence of random vectors $\{\boldsymbol{p}_k\}_{k=1}^K$ satisfy $\mathbb{E}[\boldsymbol{p}_k] = \boldsymbol{0}, \forall k$. Then,*

$$\mathbb{E}\left\|\frac{1}{K}\sum_{k=1}^K \boldsymbol{p}_k\right\|^2 = \frac{1}{K^2}\sum_{k=1}^K \mathbb{E}\|\boldsymbol{p}_k\|^2.$$

**Lemma F.1.4.** *[46, Lemma 2] Suppose a sequence of random vectors $\{\boldsymbol{p}_k\}_{k=1}^K$ satisfy $\mathbb{E}[\boldsymbol{p}_k \mid \boldsymbol{p}_{k-1}, \boldsymbol{p}_{k-2}, \dots, \boldsymbol{p}_1] = \boldsymbol{0}, \forall k$. Then,*

$$\mathbb{E}\left[\left\|\sum_{k=1}^K \boldsymbol{p}_k\right\|^2\right] = \sum_{k=1}^K \mathbb{E}\left[\|\boldsymbol{p}_k\|^2\right].$$

**Lemma F.1.5.** *[21, Lemma 17] For any $a_0 \geq 0, b \geq 0, c \geq 0, d > 0$, there exist a constant $\eta \leq \frac{1}{d}$ such that*

$$\frac{a_0}{T\eta} + b\eta + c\eta^2 \leq 2\left(\frac{a_0 b}{T}\right)^{\frac{1}{2}} + 2c^{\frac{1}{3}}\left(\frac{a_0}{T}\right)^{\frac{2}{3}} + \frac{da_0}{T}. \tag{10}$$

## F.2  Proofs of Theorem 4.1 and Corollary 4.1

For the convenience of presentation, we introduce the following notations

$$\sum_{e=0}^{E-1}\frac{1}{N}\sum_{j=1}^N Z_j^{t,e} = \sum_{e=0}^{E-1}\frac{1}{N}\sum_{j=1}^N \frac{1}{n_j}\sum_{i\in\mathcal{C}_j}\mathbb{E}\left\|\boldsymbol{z}_i^{t,e} + \nabla F_i\left(\bar{\boldsymbol{x}}_j^{t,e}\right) - \nabla f_j\left(\bar{\boldsymbol{x}}_j^{t,e}\right)\right\|^2$$

$$\sum_{e=0}^{E-1}\frac{1}{N}\sum_{j=1}^N Y_j^{t,e} = \sum_{e=0}^{E-1}\frac{1}{N}\sum_{j=1}^N \mathbb{E}\left\|\boldsymbol{y}_j^t + \nabla f_j\left(\hat{\boldsymbol{x}}^{t,e}\right) - \nabla f\left(\hat{\boldsymbol{x}}^{t,e}\right)\right\|^2$$

$$D_t = \sum_{e=0}^{E-1}\frac{1}{N}\sum_{j=1}^N \mathbb{E}\left\|\hat{\boldsymbol{x}}^{t,e} - \bar{\boldsymbol{x}}_j^{t,e}\right\|^2 \tag{11}$$

$$Q_t = \sum_{e=0}^{E-1}\frac{1}{NH}\sum_{j=1}^N \frac{1}{n_j}\sum_{i\in\mathcal{C}_j}\sum_{h=0}^{H-1}\mathbb{E}\left\|\bar{\boldsymbol{x}}_j^{t,e} - \boldsymbol{x}_{i,h}^{t,e}\right\|^2$$

$$\sum_{e=0}^{E-1}\frac{1}{N}\sum_{j=1}^N \Theta_j^{t,e} = \sum_{e=0}^{E-1}\frac{1}{N}\sum_{j=1}^N \mathbb{E}\left\|\bar{\boldsymbol{x}}_j^{t,e+1} - \bar{\boldsymbol{x}}_j^{t,e}\right\|^2,$$

where $Z_j^{t,e}$ and $Y_j^{t,e}$ characterize the biases between client-group correction term $\boldsymbol{z}_i^{t,e}$ and $\nabla F_i\left(\bar{\boldsymbol{x}}_j^{t,e}\right) - \nabla f_j\left(\bar{\boldsymbol{x}}_j^{t,e}\right)$ and between group-global correction term $\boldsymbol{y}_j^t$ and $\nabla f_j\left(\hat{\boldsymbol{x}}^{t,e}\right) - \nabla f\left(\hat{\boldsymbol{x}}^{t,e}\right)$, respectively, $D_t$ and $Q_t$ denote the group model drift and client model drift, respectively, and $\Theta_j^{t,e}$ represents model progress for group $j$.

To prove the convergence of MTGC, we start with characterizing the evolution of the global loss, i.e., $f(\boldsymbol{x})$ through the following lemma.

**Lemma F.2.1.** *Suppose that Assumptions 1 and 2 hold and $\gamma \leq \frac{1}{2HL}$, then the iterates generated by Algorithm 1 satisfy*

$$\mathbb{E}f(\bar{\boldsymbol{x}}^{t+1}) \leq \mathbb{E}f(\bar{\boldsymbol{x}}^t) - \frac{\gamma H}{2}\sum_{e=0}^{E-1}\mathbb{E}\big\|\nabla f(\hat{\boldsymbol{x}}^{t,e})\big\|^2 + \gamma L^2 H\left(Q_t + D_t\right) + \gamma^2 LEH\frac{1}{N^2}\sum_{j=1}^{N}\frac{1}{n_j}\sigma^2. \tag{12}$$

Lemma (F.2.1) implies that we need further to study the evolution of $Q_t$ and $D_t$. In particular, we establish upper bounds for $Q_t$ and $D_t$ in Lemmas F.2.2 and F.2.3, respectively.

**Lemma F.2.2.** *Suppose that Assumptions 1 and 2 hold and $\gamma \leq \frac{1}{8HL}$, then the client model drift $Q_t$, defined in (11), can be bounded as*

$$Q_t \leq 24\gamma^2 H^2 L^2 D_t + 12\gamma^2 H^2 \sum_{e=0}^{E-1}\frac{1}{N}\sum_{j=1}^{N}Z_t^{t,e} + 12\gamma^2 H^2 \sum_{e=0}^{E-1}\frac{1}{N}\sum_{j=1}^{N}Y_j^{t,e}$$

$$+ 24\gamma^2 H^2 \sum_{e=0}^{E-1}\mathbb{E}\left\|\nabla f\left(\hat{\boldsymbol{x}}^{t,e}\right)\right\|^2 + 3EH\gamma^2\sigma^2. \tag{13}$$

**Lemma F.2.3.** *Suppose that Assumptions 1 and 2 hold and $\gamma \leq \frac{1}{10EHL}$. Then the group model drift $D_t$, defined in (11), can be bounded as*

$$D_t \leq 24\gamma^2 E^2 H^2 L^2 Q_t + 12\gamma^2 E^2 H^2 \sum_{e=0}^{E-1}\frac{1}{N}\sum_{j=1}^{N}Y_j^{t,e} + 3\gamma^2 E^3 H\frac{N-1}{N^2}\sum_{j=1}^{N}\frac{1}{n_j}\sigma^2. \tag{14}$$

The results shown in Lemmas F.2.2 and F.2.3 suggest that $Z_j^{t,e}$ and $Y_j^{t,e}$ are crucial for understanding the dynamics of MTGC. Hence, we derive upper bounds for $Z_j^{t,e}$ and $Y_j^{t,e}$, which are presented in Lemmas F.2.4 and F.2.5, respectively.

**Lemma F.2.4.** *Suppose that Assumptions 1 and 2 hold. Then the bias between client-group correction term $\boldsymbol{z}_i^{t,e}$ and $\nabla F_i\left(\bar{\boldsymbol{x}}_j^{t,e}\right) - \nabla f_j\left(\bar{\boldsymbol{x}}_j^{t,e}\right)$, i.e., $Z_j^{t,e}$ can be bounded as*

$$\sum_{e=0}^{E-1}\frac{1}{N}\sum_{j=1}^{N}Z_j^{t,e} \leq 4L^2 Q_t + 4L^2 \sum_{e=0}^{E-1}\frac{1}{N}\sum_{j=1}^{N}\Theta_j^{t,e} + 2\frac{E}{H}\sigma^2 + \sigma^2. \tag{15}$$

**Lemma F.2.5.** *Suppose that Assumptions 1 and 2 hold. Then the bias between group-global correction term $\boldsymbol{y}_j^t$ and $\nabla f_j\left(\hat{\boldsymbol{x}}^{t,e}\right) - \nabla f\left(\hat{\boldsymbol{x}}^{t,e}\right)$, i.e., $Y_j^{t,e}$ defined in (11), $t \geq 1$, can be bounded as*

$$\sum_{e=0}^{E-1}\frac{1}{N}\sum_{j=1}^{N}Y_j^{t,e} \leq \left(8L^2 + 48\gamma^2 L^4 E^2 H^2\right)(Q_{t-1} + D_{t-1}) + 48\gamma^2 L^4 E^2 H^2 (Q_t + D_t)$$

$$+ 48\gamma^2 L^2 E^2 H^2 \sum_{\tau=0}^{E-1}\left(\mathbb{E}\big\|\nabla f(\hat{\boldsymbol{x}}^{t-1,\tau})\big\|^2 + \mathbb{E}\big\|\nabla f(\hat{\boldsymbol{x}}^{t,\tau})\big\|^2\right) \tag{16}$$

$$+ 32\gamma^2 L^2 E^2 H\frac{1}{N^2}\sum_{j=1}^{N}\frac{1}{n_j}\sigma^2 + \frac{2}{H}\frac{1}{N}\sum_{j=1}^{N}\frac{1}{n_j}\sigma^2.$$

*Additionally, when $t = 0$, $Y_j^{0,e}$ can be bounded as*

$$\sum_{e=0}^{E-1}\frac{1}{N}\sum_{j=1}^{N}Y_j^{0,e} \leq E\frac{1}{N}\sum_{j=1}^{N}\frac{1}{n_j}\sigma^2 + L^2 \sum_{e=0}^{E-1}\mathbb{E}\left\|\hat{\boldsymbol{x}}^{0,e} - \bar{\boldsymbol{x}}^0\right\|^2. \tag{17}$$

In addition, the upper bound of $\Theta_j^{t,e}$ is presented in the following lemma.

**Lemma F.2.6.** *Suppose that Assumptions 1 and 2 hold. Then the group model progress at the $(t,e)$-th round, i.e., $\Theta_j^{t,e}$, defined in (11), can be bounded as*

$$\sum_{e=0}^{E-1}\frac{1}{N}\sum_{j=1}^{N}\Theta_j^{t,e} \leq 8\gamma^2 H^2 L^2 Q_t + 8\gamma^2 H^2 L^2 D_t + 8\gamma^2 H^2 \sum_{e=0}^{E-1}\frac{1}{N}\sum_{j=1}^{N}Y_j^{t,e}$$

$$+ 8\gamma^2 H^2 \sum_{e=0}^{E-1}\mathbb{E}\left\|\nabla f\left(\hat{\boldsymbol{x}}^{t,e}\right)\right\|^2 + 2\gamma^2 EH\frac{1}{N}\sum_{j=1}^{N}\frac{1}{n_j}\sigma^2. \tag{18}$$

Recalling Lemma (F.2.1), we can see that what we actually need is the evolution of $Q_t + D_t$. With Lemmas F.2.2-F.2.6, we can characterize this evolution which is formalized in Lemma F.2.7.

**Lemma F.2.7.** *Suppose that Assumptions 1 and 2 hold and $\gamma \leq \frac{1}{33EHL}$, the model deviation $\Gamma_t = Q_t + D_t$ satisfies*

$$\Gamma_t \leq \frac{1}{2}\Gamma_{t-1} + \left(1152\gamma^4 H^4 L^2 + 72\gamma^2 E^2 H^2\right) \sum_{\tau=0}^{E-1} \left(\mathbb{E}\|\nabla f(\hat{\boldsymbol{x}}^{t-1,\tau})\|^2 + \mathbb{E}\|\nabla f(\hat{\boldsymbol{x}}^{t,\tau})\|^2\right) + 294\gamma^2 E^3 H\sigma^2,$$

$$\Gamma_0 \leq \left(648\gamma^4 H^4 L^2 + 42\gamma^2 E^2 H^2\right) \sum_{e=0}^{E-1} \mathbb{E}\|\nabla f\left(\hat{\boldsymbol{x}}^{0,e}\right)\|^2 + 146\gamma^2 E^3 H^2 \sigma^2.$$

The proofs of Lemmas F.2.1-F.2.7 are provided in Appendix F.3. With these lemmas, we are ready to prove Theorem 4.1.

### F.2.1 Proof of Theorem 4.1

Starting with Lemma F.2.1, i.e.,

$$\mathbb{E}f(\bar{\boldsymbol{x}}^{t+1}) - \gamma L^2 H\Gamma_t \leq \mathbb{E}f(\bar{\boldsymbol{x}}^t) - \frac{\gamma H}{2} \sum_{e=0}^{E-1} \mathbb{E}\|\nabla f(\hat{\boldsymbol{x}}^{t,e})\|^2 + \gamma^2 LEH \frac{1}{N^2} \sum_{j=1}^{N} \frac{1}{n_j}\sigma^2, \tag{19}$$

where $\Gamma_t = Q_t + D_t$. Adding $2\gamma L^2 H\Gamma_t$ on both sides of the above inequality and utilizing Lemma F.2.7, we have

$$\mathbb{E}f(\bar{\boldsymbol{x}}^{t+1}) + \gamma L^2 H\Gamma_t \leq \mathbb{E}f(\bar{\boldsymbol{x}}^t) + \gamma L^2 H\Gamma_{t-1} - \frac{\gamma H}{2} \sum_{e=0}^{E-1} \mathbb{E}\|\nabla f(\hat{\boldsymbol{x}}^{t,e})\|^2 + \gamma^2 LEH \frac{1}{N^2} \sum_{j=1}^{N} \frac{1}{n_j}\sigma^2$$

$$+ 2\gamma L^2 H \times 294\gamma^2 E^3 H\sigma^2 + \left(2304\gamma^5 H^5 L^4 + 144\gamma^3 E^2 H^3 L^2\right) \sum_{\tau=0}^{E-1} \left(\mathbb{E}\|\nabla f(\hat{\boldsymbol{x}}^{t-1,\tau})\|^2 + \mathbb{E}\|\nabla f(\hat{\boldsymbol{x}}^{t,\tau})\|^2\right).$$

For notation ease, we denote $\Phi_{t+1} = \mathbb{E}f(\bar{\boldsymbol{x}}^{t+1}) - f^* + \gamma L^2 H\Gamma_t$, $\Phi_t \geq 0$, $\forall t \geq 0$. As long as $\gamma \leq \frac{E}{8HL}$ in Theorem 4.1, $2304\gamma^5 H^5 L^4 \leq 36\gamma^3 E^2 H^3 L^2$, we thus have

$$\Phi_{t+1} \leq \Phi_t - \frac{\gamma H}{2} \sum_{e=0}^{E-1} \mathbb{E}\|\nabla f(\hat{\boldsymbol{x}}^{t,e})\|^2 + 180\gamma^3 E^2 H^3 L^2 \sum_{\tau=0}^{E-1} \left(\mathbb{E}\|\nabla f(\hat{\boldsymbol{x}}^{t-1,\tau})\|^2 + \mathbb{E}\|\nabla f(\hat{\boldsymbol{x}}^{t,\tau})\|^2\right)$$

$$+ 2\gamma L^2 H \times 294\gamma^2 E^3 H\sigma^2 + \gamma^2 LEH \frac{1}{N^2} \sum_{j=1}^{N} \frac{1}{n_j}\sigma^2. \tag{20}$$

When $\gamma \leq \frac{1}{40EHL}$, we have $\frac{\gamma H}{2} - 360\gamma^3 E^2 H^3 L^2 \geq \frac{\gamma H}{4}$. Telescoping the above inequality from $t = 1$ to $T - 1$, we have

$$\Phi_T \leq \Phi_1 - \frac{\gamma H}{4} \sum_{t=1}^{T-1}\sum_{e=0}^{E-1} \mathbb{E}\|\nabla f(\hat{\boldsymbol{x}}^{t,e})\|^2 + 180\gamma^3 E^2 H^3 L^2 \sum_{\tau=0}^{E-1} \mathbb{E}\|\nabla f(\hat{\boldsymbol{x}}^{0,\tau})\|^2$$

$$+ 2\gamma(T-1)L^2 H \times 294\gamma^2 E^3 H\sigma^2 + \gamma^2(T-1)LEH \frac{1}{N^2} \sum_{j=1}^{N} \frac{1}{n_j}\sigma^2. \tag{21}$$

According to Lemma F.2.1, when $t = 0$,

$$\mathbb{E}f(\bar{\boldsymbol{x}}^1) \leq \mathbb{E}f(\bar{\boldsymbol{x}}^0) - \frac{\gamma H}{2} \sum_{e=0}^{E-1} \mathbb{E}\|\nabla f(\hat{\boldsymbol{x}}^{0,e})\|^2 + \gamma L^2 H\Gamma_0 + \gamma^2 LEH \frac{1}{N^2} \sum_{j=1}^{N} \frac{1}{n_j}\sigma^2, \tag{22}$$

it follows that

$$\mathbb{E}f(\bar{\boldsymbol{x}}^1) + \gamma L^2 H\Gamma_0 \leq \mathbb{E}f(\bar{\boldsymbol{x}}^0) - \frac{\gamma H}{2} \sum_{e=0}^{E-1} \mathbb{E}\|\nabla f(\hat{\boldsymbol{x}}^{0,e})\|^2 + 2\gamma L^2 H\Gamma_0 + \gamma^2 LEH \frac{1}{N^2} \sum_{j=1}^{N} \frac{1}{n_j}\sigma^2. \tag{23}$$

Combining (23) with (21) and plugging the upper bound of $\Gamma_0$, established in Lemma F.2.7, into the inequality, we have

$$
\begin{aligned}
\Phi_T \leq & \mathbb{E}f(\bar{\boldsymbol{x}}^0) - f^* + 2\gamma L^2 H \times 146\gamma^2 E^3 H^2 \sigma^2 + 2\gamma T L^2 H \times 292\gamma^2 E^3 H \sigma^2 + \gamma^2 T L E H \frac{1}{N^2} \sum_{j=1}^{N} \frac{1}{n_j} \sigma^2 \\
& - \left( \frac{\gamma H}{2} - 180\gamma^3 E^2 H^3 L^2 - 2\gamma L^2 H \times \left( 648\gamma^4 H^4 L^2 + 42\gamma^2 E^2 H^2 \right) \right) \sum_{e=0}^{E-1} \mathbb{E}\left\| \nabla f(\hat{\boldsymbol{x}}^{0,e}) \right\|^2 \\
& - \frac{\gamma H}{4} \sum_{t=1}^{T-1} \sum_{e=0}^{E-1} \mathbb{E}\left\| \nabla f(\hat{\boldsymbol{x}}^{t,e}) \right\|^2 .
\end{aligned}
$$

The setup of $\gamma$ presented in Theorem 4.1 is enough to guarantee $\frac{\gamma H}{2} - 180\gamma^3 E^2 H^3 L^2 - 2\gamma L^2 H \times \left( 648\gamma^4 H^4 L^2 + 42\gamma^2 E^2 H^2 \right) \geq \frac{\gamma H}{4}$. Therefore, combining the last terms and taking some basic algebra operation, we have

$$
\frac{1}{TE} \sum_{t=0}^{T-1} \sum_{e=0}^{E-1} \mathbb{E}\left\| \nabla f(\hat{\boldsymbol{x}}^{t,e}) \right\|^2 \leq 4 \frac{\mathbb{E}f(\bar{\boldsymbol{x}}^0) - f^*}{\gamma T E H} + 4\gamma L \frac{1}{N^2} \sum_{j=1}^{N} \frac{1}{n_j} \sigma^2 + \frac{1168}{T} \gamma^2 L^2 E^2 H^2 \sigma^2 + 2352\gamma^2 L^2 E^2 H \sigma^2 .
$$

The above bound can be further simplified to

$$
\frac{1}{TE} \sum_{t=0}^{T-1} \sum_{e=0}^{E-1} \mathbb{E}\left\| \nabla f(\hat{\boldsymbol{x}}^{t,e}) \right\|^2 \leq 4 \frac{\mathbb{E}f(\bar{\boldsymbol{x}}^0) - f^*}{\gamma T E H} + 4 \frac{\gamma L \sigma^2}{\tilde{N}} + 3520\gamma^2 E^2 H^2 L^2 \sigma^2 . \tag{24}
$$

This completes the proof of Theorem 4.1.

### F.2.2 Proof of Corollary 4.1

Rewriting the bound in 24 as

$$
\frac{1}{TE} \sum_{t=0}^{T-1} \sum_{e=0}^{E-1} \mathbb{E}\left\| \nabla f(\hat{\boldsymbol{x}}^{t,e}) \right\|^2 \leq \frac{4 \left( \mathbb{E}f(\bar{\boldsymbol{x}}^0) - f^* \right)}{T(\gamma E H)} + \frac{4 L \sigma^2}{\tilde{N} E H}(\gamma E H) + 3520 L^2 \sigma^2 (\gamma E H)^2 , \tag{25}
$$

and recalling Lemma F.1.5, one can claim that there exists a learning rate $(\gamma E H) \leq \frac{1}{d}$ such that

$$
\begin{aligned}
\frac{1}{TE} \sum_{t=0}^{T-1} \sum_{e=0}^{E-1} \mathbb{E}\left\| \nabla f(\hat{\boldsymbol{x}}^{t,e}) \right\|^2 \leq & 8 \sqrt{\frac{(\mathbb{E}f(\bar{\boldsymbol{x}}^0) - f^*) L \sigma^2}{\tilde{N} T E H}} + 96 \left( \frac{(\mathbb{E}f(\bar{\boldsymbol{x}}^0) - f^*) L \sigma}{T} \right)^{\frac{2}{3}} + \frac{d(\mathbb{E}f(\bar{\boldsymbol{x}}^0) - f^*)}{T} \\
& \sim \mathcal{O}\left( \sqrt{\frac{\mathcal{F}_0 L \sigma^2}{\tilde{N} T E H}} + \left( \frac{\mathcal{F}_0 L \sigma}{T} \right)^{\frac{2}{3}} + \frac{d \mathcal{F}_0}{T} \right) .
\end{aligned} \tag{26}
$$

Given that we need $\gamma \leq \frac{1}{40 E H L}$ for Theorem 4.1, we can set $d = 40L$. We thus can find a step size in the range of $(\gamma E H) \leq \frac{1}{40L}$, i.e., $\gamma \leq \frac{1}{40 E H L}$ such that

$$
\frac{1}{TE} \sum_{t=0}^{T-1} \sum_{e=0}^{E-1} \mathbb{E}\left\| \nabla f(\hat{\boldsymbol{x}}^{t,e}) \right\|^2 \leq \mathcal{O}\left( \sqrt{\frac{\mathcal{F}_0 L \sigma^2}{\tilde{N} T E H}} + \left( \frac{\mathcal{F}_0 L \sigma}{T} \right)^{\frac{2}{3}} + \frac{L \mathcal{F}_0}{T} \right) . \tag{27}
$$

This completes the proof of Corollary 4.1.

### F.3 Proofs of Lemmas F.2.1-F.2.7

### F.3.1 Proof of Lemma F.2.1

Under the framework of MTGC, the virtual global model obeys the following iteration:

$$
\hat{\boldsymbol{x}}^{t,e+1} = \hat{\boldsymbol{x}}^{t,e} - \gamma \sum_{j=1}^{N} \frac{1}{n_j} \sum_{i \in \mathcal{C}_j} \sum_{h=0}^{H-1} \left( \nabla F_i \left( \boldsymbol{x}_{i,h}^{t,e}, \xi_{i,h}^{t,e} \right) + \boldsymbol{z}_i^{t,e} + \boldsymbol{y}_j^t \right) .
$$

As $\sum_{i\in\mathcal{C}_j} \boldsymbol{z}_i^{t,e} = \boldsymbol{0}$ and $\sum_{j=1}^{N} \boldsymbol{y}_j^t = \boldsymbol{0}$, the the virtual global model iteration reduces to

$$\hat{\boldsymbol{x}}^{t,e+1} = \hat{\boldsymbol{x}}^{t,e} - \gamma \sum_{j=1}^{N} \frac{1}{n_j} \sum_{i\in\mathcal{C}_j} \sum_{h=0}^{H-1} \nabla F_i\left(\boldsymbol{x}_{i,h}^{t,e}, \xi_{i,h}^{t,e}\right). \tag{28}$$

With Assumption 1, we have

$$f(\boldsymbol{y}) \leq f(\boldsymbol{x}) + <\nabla f(\boldsymbol{x}), \boldsymbol{y} - \boldsymbol{x}> + \frac{L}{2}\|\boldsymbol{y} - \boldsymbol{x}\|^2. \tag{29}$$

Plugging (28) into (29), we have

$$\mathbb{E}f(\hat{\boldsymbol{x}}^{t,e+1}) \leq \mathbb{E}f(\hat{\boldsymbol{x}}^{t,e}) \underbrace{- \gamma\mathbb{E}\left\langle \nabla f(\hat{\boldsymbol{x}}^{t,e}), \frac{1}{N}\sum_{j=1}^{N}\frac{1}{n_j}\sum_{i\in\mathcal{C}_j}\sum_{h=0}^{H-1}\nabla F_i\left(\boldsymbol{x}_{i,h}^{t,e}, \xi_{i,h}^{t,e}\right)\right\rangle}_{T_1}$$

$$\underbrace{+ \gamma^2 L \frac{1}{2}\mathbb{E}\left\|\frac{1}{N}\sum_{j=1}^{N}\frac{1}{n_j}\sum_{i\in\mathcal{C}_j}\sum_{h=0}^{H-1}\nabla F_i\left(\boldsymbol{x}_{i,h}^{t,e}, \xi_{i,h}^{t,e}\right)\right\|^2}_{T_2}. \tag{30}$$

Utilizing $\mathbb{E}\left[\frac{1}{N}\sum_{j=1}^{N}\frac{1}{n_j}\sum_{i\in\mathcal{C}_j}\sum_{h=0}^{H-1}\nabla F_i\left(\boldsymbol{x}_{i,h}^{t,e}, \xi_{i,h}^{t,e}\right) - \frac{1}{N}\sum_{j=1}^{N}\frac{1}{n_j}\sum_{i\in\mathcal{C}_j}\sum_{h=0}^{H-1}\nabla F_i(\boldsymbol{x}_{i,h}^{t,e})\right] = 0$, we rewrite $T_1$ as follows

$$T_1 = -\gamma H\mathbb{E}\left\langle \nabla f(\hat{\boldsymbol{x}}^{t,e}), \frac{1}{NH}\sum_{j=1}^{N}\frac{1}{n_j}\sum_{i\in\mathcal{C}_j}\sum_{h=0}^{H-1}\nabla F_i(\boldsymbol{x}_{i,h}^{t,e})\right\rangle$$

$$= \frac{\gamma H}{2}\mathbb{E}\left\|\nabla f(\hat{\boldsymbol{x}}^{t,e}) \mp \frac{1}{N}\sum_{j=1}^{N}\nabla f_j(\bar{\boldsymbol{x}}_j^{t,e}) - \frac{1}{NH}\sum_{j=1}^{N}\frac{1}{n_j}\sum_{i\in\mathcal{C}_j}\sum_{h=0}^{H-1}\nabla F_i(\boldsymbol{x}_{i,h}^{t,e})\right\|^2 - \frac{\gamma H}{2}\mathbb{E}\|\nabla f(\hat{\boldsymbol{x}}^{t,e})\|^2$$

$$- \frac{\gamma H}{2}\mathbb{E}\left\|\frac{1}{NH}\sum_{j=1}^{N}\frac{1}{n_j}\sum_{i\in\mathcal{C}_j}\sum_{h=0}^{H-1}\nabla F_i(\boldsymbol{x}_{i,h}^{t,e})\right\|^2$$

$$\leq \gamma H\mathbb{E}\left\|\nabla f(\hat{\boldsymbol{x}}^{t,e}) - \frac{1}{N}\sum_{j=1}^{N}\nabla f_j(\bar{\boldsymbol{x}}_j^{t,e})\right\|^2 + \gamma H\mathbb{E}\left\|\frac{1}{N}\sum_{j=1}^{N}\nabla f_j(\bar{\boldsymbol{x}}_j^{t,e}) - \frac{1}{NH}\sum_{j=1}^{N}\frac{1}{n_j}\sum_{i\in\mathcal{C}_j}\sum_{h=0}^{H-1}\nabla F_i(\boldsymbol{x}_{i,h}^{t,e})\right\|^2$$

$$- \frac{\gamma H}{2}\mathbb{E}\|\nabla f(\hat{\boldsymbol{x}}^{t,e})\|^2 - \frac{\gamma H}{2}\mathbb{E}\left\|\frac{1}{NH}\sum_{j=1}^{N}\frac{1}{n_j}\sum_{i\in\mathcal{C}_j}\sum_{h=0}^{H-1}\nabla F_i(\boldsymbol{x}_{i,h}^{t,e})\right\|^2$$

$$\leq \gamma H\frac{1}{N}\sum_{j=1}^{N}\mathbb{E}\|\nabla f_j(\hat{\boldsymbol{x}}^{t,e}) - \nabla f_j(\bar{\boldsymbol{x}}_j^{t,e})\|^2 + \gamma H\frac{1}{NH}\sum_{j=1}^{N}\frac{1}{n_j}\sum_{i\in\mathcal{C}_j}\sum_{h=0}^{H-1}\mathbb{E}\|\nabla F_i(\bar{\boldsymbol{x}}_j^{t,e}) - \nabla F_i(\boldsymbol{x}_{i,h}^{t,e})\|^2$$

$$- \frac{\gamma H}{2}\mathbb{E}\|\nabla f(\hat{\boldsymbol{x}}^{t,e})\|^2 - \frac{\gamma H}{2}\mathbb{E}\left\|\frac{1}{NH}\sum_{j=1}^{N}\frac{1}{n_j}\sum_{i\in\mathcal{C}_j}\sum_{h=0}^{H-1}\nabla F_i(\boldsymbol{x}_{i,h}^{t,e})\right\|^2$$

$$\leq \gamma HL^2\frac{1}{N}\sum_{j=1}^{N}\mathbb{E}\|\hat{\boldsymbol{x}}^{t,e} - \bar{\boldsymbol{x}}_j^{t,e}\|^2 + \gamma HL^2\frac{1}{NH}\sum_{j=1}^{N}\frac{1}{n_j}\sum_{h=0}^{H-1}\sum_{i\in\mathcal{C}_j}\mathbb{E}\|\bar{\boldsymbol{x}}_j^{t,e} - \boldsymbol{x}_{i,h}^{t,e}\|^2$$

$$- \frac{\gamma H}{2}\mathbb{E}\|\nabla f(\hat{\boldsymbol{x}}^{t,e})\|^2 - \frac{\gamma H}{2}\mathbb{E}\left\|\frac{1}{N}\sum_{j=1}^{N}\frac{1}{n_j}\sum_{i\in\mathcal{C}_j}\frac{1}{H}\sum_{h=0}^{H-1}\nabla F_i(\boldsymbol{x}_{i,h}^{t,e})\right\|^2,$$

where the last inequality comes from Assumption 1.

On the other hand, we can bound $T_2$ as follows

$$
\begin{aligned}
T_2 \leq &\mathbb{E}\left\|\frac{1}{N}\sum_{j=1}^{N}\frac{1}{n_j}\sum_{i\in\mathcal{C}_j}\sum_{h=0}^{H-1}\left(\nabla F_i\left(\boldsymbol{x}_{i,h}^{t,e},\xi_{i,h}^{t,e}\right)-\nabla F_i(\boldsymbol{x}_{i,h}^{t,e})\right)\right\|^2 + \mathbb{E}\left\|\frac{1}{N}\sum_{j=1}^{N}\frac{1}{n_j}\sum_{i\in\mathcal{C}_j}\sum_{h=0}^{H-1}\nabla F_i(\boldsymbol{x}_{i,h}^{t,e})\right\|^2 \\
\leq &\frac{1}{N^2}\sum_{j=1}^{N}\frac{1}{n_j^2}\sum_{i\in\mathcal{C}_j}\mathbb{E}\left\|\sum_{h=0}^{H-1}\boldsymbol{g}_{i,h}^{t,e}-\sum_{h=0}^{H-1}\nabla F_i(\boldsymbol{x}_{i,h}^{t,e})\right\|^2 + \mathbb{E}\left\|\frac{1}{N}\sum_{j=1}^{N}\frac{1}{n_j}\sum_{i\in\mathcal{C}_j}\sum_{h=0}^{H-1}\nabla F_i(\boldsymbol{x}_{i,h}^{t,e})\right\|^2 \\
\leq &\frac{H\sigma^2}{N^2}\sum_{j=1}^{N}\frac{1}{n_j}+\mathbb{E}\left\|\frac{1}{N}\sum_{j=1}^{N}\frac{1}{n_j}\sum_{i\in\mathcal{C}_j}\sum_{h=0}^{H-1}\nabla F_i(\boldsymbol{x}_{i,h}^{t,e})\right\|^2,
\end{aligned}
$$

where second inequality comes from Lemma F.1.3 and the last inequality follows Assumption 2 and Lemma F.1.4.

Plugging the derived upper bounds of $T_1$ and $T_2$ into 30 and utilized $\gamma \leq \frac{1}{2HL}$, we obtain

$$
\begin{aligned}
\mathbb{E}f(\hat{\boldsymbol{x}}^{t,e+1}) \leq &\mathbb{E}f(\hat{\boldsymbol{x}}^{t,e}) - \frac{\gamma H}{2}\mathbb{E}\left\|\nabla f(\hat{\boldsymbol{x}}^{t,e})\right\|^2 + \gamma^2 L\frac{H\sigma^2}{N^2}\sum_{j=1}^{N}\frac{1}{n_j} \\
&+ \gamma HL^2\frac{1}{NH}\sum_{j=1}^{N}\sum_{h=0}^{H-1}\frac{1}{n_j}\sum_{i\in\mathcal{C}_j}\mathbb{E}\left\|\bar{\boldsymbol{x}}_j^{t,e}-\boldsymbol{x}_{i,h}^{t,e}\right\|^2 + \gamma HL^2\frac{1}{N}\sum_{j=1}^{N}\mathbb{E}\left\|\hat{\boldsymbol{x}}^{t,e}-\bar{\boldsymbol{x}}_j^{t,e}\right\|^2.
\end{aligned}
$$

Telescoping the above inequality from $e=0$ to $H-1$ gives rise to Lemma F.2.1.

### F.3.2 Proofs of Lemmas F.2.2 and F.2.6

**Part I (Lemma F.2.2):** Let $q_{j,h}^{t,e} = \frac{1}{n_j}\sum_{i\in\mathcal{C}_j}\mathbb{E}\left\|\bar{\boldsymbol{x}}_j^{t,e}-\boldsymbol{x}_{i,h}^{t,e}\right\|^2$, $q_{j,0}^{t,e}=0$. For $0\leq h\leq H-2$, we have

$$
\begin{aligned}
&n_j q_{j,h+1}^{t,e} \\
=&\sum_{i\in\mathcal{C}_j}\mathbb{E}\left\|\boldsymbol{x}_{i,h}^{t,e}-\gamma\left(\nabla F_i\left(\boldsymbol{x}_{i,h}^{t,e},\xi_{i,h}^{t,e}\right)+\boldsymbol{y}_j^t+\boldsymbol{z}_i^{t,e}\right)-\bar{\boldsymbol{x}}_j^{t,e}\right\|^2 \\
\leq&\left(1+\frac{1}{H-1}\right)\mathbb{E}\left\|\boldsymbol{x}_{i,h}^{t,e}-\bar{\boldsymbol{x}}_j^{t,e}\right\|^2+H\sum_{i\in\mathcal{C}_j}\mathbb{E}\left\|\gamma\left(\nabla F_i\left(\boldsymbol{x}_{i,h}^{t,e}\right)+\boldsymbol{y}_j^t+\boldsymbol{z}_i^{t,e}\right)\right\|^2+n_j\gamma^2\sigma^2 \\
=&\left(1+\frac{1}{H-1}\right)\mathbb{E}\left\|\boldsymbol{x}_{i,h}^{t,e}-\bar{\boldsymbol{x}}_j^{t,e}\right\|^2+\gamma^2 H\sum_{i\in\mathcal{C}_j}\mathbb{E}\left\|\nabla F_i\left(\boldsymbol{x}_{i,h}^{t,e}\right)\mp\nabla F_i\left(\bar{\boldsymbol{x}}_j^{t,e}\right)\mp\nabla f_j\left(\bar{\boldsymbol{x}}_j^{t,e}\right)\right. \\
&\left.\mp\nabla f_j\left(\hat{\boldsymbol{x}}^{t,e}\right)\mp\nabla f\left(\hat{\boldsymbol{x}}^{t,e}\right)+\boldsymbol{y}_j^t+\boldsymbol{z}_i^{t,e}\right\|^2+n_j\gamma^2\sigma^2 \\
\leq&\left(1+\frac{1}{H-1}+4\gamma^2 HL^2\right)\sum_{i\in\mathcal{C}_j}\mathbb{E}\left\|\boldsymbol{x}_{i,h}^{t,e}-\bar{\boldsymbol{x}}_j^{t,e}\right\|^2+4\gamma^2 H\sum_{i\in\mathcal{C}_j}\mathbb{E}\left\|\boldsymbol{z}_i^{t,e}+\nabla F_i\left(\bar{\boldsymbol{x}}_j^{t,e}\right)-\nabla f_j\left(\bar{\boldsymbol{x}}_j^{t,e}\right)\right\|^2 \\
&+4\gamma^2 Hn_j\mathbb{E}\left\|\boldsymbol{y}_i^t+\nabla f_j\left(\hat{\boldsymbol{x}}^{t,e}\right)-\nabla f\left(\hat{\boldsymbol{x}}^{t,e}\right)\right\|^2+8\gamma^2 HL^2 n_j\mathbb{E}\left\|\bar{\boldsymbol{x}}_j^{t,e}-\hat{\boldsymbol{x}}^{t,e}\right\| \\
&+8\gamma^2 Hn_j\mathbb{E}\left\|\nabla f\left(\hat{\boldsymbol{x}}^{t,e}\right)\right\|^2+n_j\gamma^2\sigma^2,
\end{aligned}
$$

where the first inequality comes from Lemma F.1.2 and Assumption 2 and the second inequality follows Lemma F.1.1. Let $\rho_1 = \left(1+\frac{1}{H-1}+4\gamma^2 HL^2\right)$. As $\gamma \leq \frac{1}{8HL}$, we have $\rho_1^h \leq \rho_1^{H-1} \leq \left(1+\frac{1}{H-1}+\frac{1}{16(H-1)}\right)^{H-1} \leq e_0^{\frac{17}{16}} < 3$ and $\sum_{h=0}^{H-1}\rho_1^h \leq 3H$, where $e_0$ denotes Euler's number. We

thus have

$$n_j q_{j,h+1}^{t,e}$$

$$\leq \left(\sum_{\tau=0}^{h} \rho_1^{\tau}\right) \left(4\gamma^2 H \sum_{i\in\mathcal{C}_j} \mathbb{E}\left\|\boldsymbol{z}_i^{t,e} + \nabla F_i\left(\bar{\boldsymbol{x}}_j^{t,e}\right) - \nabla f_j\left(\bar{\boldsymbol{x}}_j^{t,e}\right)\right\|^2 + 8\gamma^2 H L^2 n_j \mathbb{E}\left\|\bar{\boldsymbol{x}}_j^{t,e} - \hat{\boldsymbol{x}}^{t,e}\right\|\right.$$

$$\left. + 4\gamma^2 H n_j \mathbb{E}\left\|\boldsymbol{y}_j^t + \nabla f_j\left(\bar{\boldsymbol{x}}_j^{t,e}\right) - \nabla f\left(\bar{\boldsymbol{x}}^t\right)\right\|^2 + 8\gamma^2 H n_j \mathbb{E}\left\|\nabla f\left(\hat{\boldsymbol{x}}^{t,e}\right)\right\|^2 + n_j \gamma^2 \sigma^2\right)$$

$$\leq 12\gamma^2 H^2 \sum_{i\in\mathcal{C}_j} \mathbb{E}\left\|\boldsymbol{z}_i^{t,e} + \nabla F_i\left(\bar{\boldsymbol{x}}_j^{t,e}\right) - \nabla f_j\left(\bar{\boldsymbol{x}}_j^{t,e}\right)\right\|^2 + 12\gamma^2 H^2 n_j \mathbb{E}\left\|\boldsymbol{y}_j^t + \nabla f_j\left(\hat{\boldsymbol{x}}^{t,e}\right) - \nabla f\left(\hat{\boldsymbol{x}}^{t,e}\right)\right\|^2$$

$$+ 24\gamma^2 H^2 L^2 n_j \mathbb{E}\left\|\bar{\boldsymbol{x}}_j^{t,e} - \hat{\boldsymbol{x}}^{t,e}\right\| + 24\gamma^2 H^2 n_j \mathbb{E}\left\|\nabla f\left(\hat{\boldsymbol{x}}^{t,e}\right)\right\|^2 + 3H n_j \gamma^2 \sigma^2,$$

where the last inequality follows Assumption 1.

Plugging the derived upper bound of $n_j q_{j,h+1}^{t,e}$ into $Q_t = \sum_{e=0}^{E-1} \frac{1}{N} \sum_{j=1}^{N} \left(\frac{1}{H} \sum_{h=0}^{H-1} q_{j,h}^{t,e}\right)$ gives rise to Lemma F.2.2.

**Part II (Lemma F.2.6):** First, $\bar{\boldsymbol{x}}_j^{t,e+1} = \bar{\boldsymbol{x}}_j^{t,e} + \gamma \sum_{h=0}^{H-1} \frac{1}{n_j} \sum_{i\in\mathcal{C}_j} \left(\nabla F_i\left(\boldsymbol{x}_{i,h}^{t,e}, \xi_{i,h}^{t,e}\right) + \boldsymbol{z}_i^{t,e} + \boldsymbol{y}_j^t\right)$. As $\sum_{i\in\mathcal{C}_j} \boldsymbol{z}_i^{t,e} = \boldsymbol{0}$, we can rewrite $\Theta_j^{t,e}$ as

$$\Theta_j^{t,e} = \mathbb{E}\left\|\bar{\boldsymbol{x}}_j^{t,e+1} - \bar{\boldsymbol{x}}_j^{t,e}\right\|^2 = \mathbb{E}\left\|\gamma \sum_{h=0}^{H-1} \frac{1}{n_j} \sum_{i\in\mathcal{C}_j} \left(\nabla F_i\left(\boldsymbol{x}_{i,h}^{t,e}, \xi_{i,h}^{t,e}\right) + \boldsymbol{y}_j^t\right)\right\|^2.$$

Next, we establish an upper bound for $\Theta_j^{t,e}$ as follows

$$\Theta_j^{t,e} \leq 2\gamma^2 \mathbb{E}\left\|\sum_{h=0}^{H-1} \frac{1}{n_j} \sum_{i\in\mathcal{C}_j} \left(\nabla F_i\left(\boldsymbol{x}_{i,h}^{t,e}\right) + \boldsymbol{y}_j^t\right)\right\|^2 + 2\gamma^2 \mathbb{E}\left\|\sum_{h=0}^{H-1} \frac{1}{n_j} \sum_{i\in\mathcal{C}_j} \left(\nabla F_i\left(\boldsymbol{x}_{i,h}^{t,e}, \xi_{i,h}^{t,e}\right) - \nabla F_i\left(\boldsymbol{x}_{i,h}^{t,e}\right)\right)\right\|^2$$

$$\leq 2\gamma^2 H \sum_{h=0}^{H-1} \mathbb{E}\left\|\frac{1}{n_j} \sum_{i\in\mathcal{C}_j} \nabla F_i\left(\boldsymbol{x}_{i,h}^{t,e}\right) + \boldsymbol{y}_j^t\right\|^2 + 2\frac{1}{n_j^2} \sum_{i\in\mathcal{C}_j} \gamma^2 \mathbb{E}\left\|\sum_{h=0}^{H-1} \left(\nabla F_i\left(\boldsymbol{x}_{i,h}^{t,e}, \xi_{i,h}^{t,e}\right) - \nabla F_i\left(\boldsymbol{x}_{i,h}^{t,e}\right)\right)\right\|^2$$

$$\leq 2\gamma^2 H \sum_{h=0}^{H-1} \mathbb{E}\left\|\frac{1}{n_j} \sum_{i\in\mathcal{C}_j} \nabla F_i\left(\boldsymbol{x}_{i,h}^{t,e}\right) \mp \nabla f_j\left(\bar{\boldsymbol{x}}_j^{t,e}\right) \mp \nabla f_j\left(\hat{\boldsymbol{x}}^{t,e}\right) \mp \nabla f\left(\hat{\boldsymbol{x}}^{t,e}\right) + \boldsymbol{y}_j^t\right\|^2 + 2\gamma^2 \frac{H\sigma^2}{n_j}$$

$$\leq 8\gamma^2 H^2 L^2 \frac{1}{H} \sum_{h=0}^{H-1} \frac{1}{n_j} \sum_{i\in\mathcal{C}_j} \mathbb{E}\left\|\boldsymbol{x}_{i,h}^{t,e} - \bar{\boldsymbol{x}}_j^{t,e}\right\|^2 + 8\gamma^2 H^2 \mathbb{E}\left\|\boldsymbol{y}_j^t + \nabla f_j\left(\hat{\boldsymbol{x}}^{t,e}\right) - \nabla f\left(\hat{\boldsymbol{x}}^{t,e}\right)\right\|^2$$

$$+ 8\gamma^2 H^2 L^2 \mathbb{E}\left\|\bar{\boldsymbol{x}}_j^{t,e} - \hat{\boldsymbol{x}}^{t,e}\right\| + 8\gamma^2 H^2 \mathbb{E}\left\|\nabla f\left(\hat{\boldsymbol{x}}^{t,e}\right)\right\|^2 + 2\gamma^2 \frac{H\sigma^2}{n_j},$$

where the second inequality holds due to Lemmas F.1.1 and F.1.3, the second third inequality comes from Lemmas F.1.4 and Assumption 2, and the last inequality follows Assumption 1. Plugging this upper bound into $\sum_{e=0}^{E-1} \frac{1}{N} \sum_{j=1}^{N} \Theta_j^{t,e}$ gives rise to Lemma F.2.6.

### F.3.3 Proof of Lemma F.2.3

To bound $D_t = \sum_{e=0}^{E-1} \frac{1}{N} \sum_{j=1}^{N} \mathbb{E}_t^e \left\|\bar{\boldsymbol{x}}_j^{t,e} - \hat{\boldsymbol{x}}^{t,e}\right\|^2$, we first rewrite $\mathbb{E}\left\|\bar{\boldsymbol{x}}_j^{t,e+1} - \hat{\boldsymbol{x}}^{t,e+1}\right\|^2$ as follows

$$\mathbb{E}\left\|\bar{\boldsymbol{x}}_j^{t,e+1} - \hat{\boldsymbol{x}}^{t,e+1}\right\|^2$$

$$= \mathbb{E}\left\|\bar{\boldsymbol{x}}_j^{t,e} - \frac{1}{n_j} \sum_{i\in\mathcal{C}_j} \sum_{h=0}^{H-1} \gamma\left(\nabla F_i\left(\boldsymbol{x}_{i,h}^{t,e}, \xi_{i,h}^{t,e}\right) + \boldsymbol{y}_j^t + \boldsymbol{z}_i^{t,e}\right)\right.$$

$$\left. - \hat{\boldsymbol{x}}^{t,e} + \frac{1}{N} \sum_{j=1}^{N} \frac{1}{n_j} \sum_{i\in\mathcal{C}_j} \sum_{h=0}^{H-1} \gamma\left(\nabla F_i\left(\boldsymbol{x}_{i,h}^{t,e}, \xi_{i,h}^{t,e}\right) + \boldsymbol{y}_j^t + \boldsymbol{z}_i^{t,e}\right)\right\|^2$$

$$= \mathbb{E}\left\|\bar{\boldsymbol{x}}_j^{t,e} - \hat{\boldsymbol{x}}^{t,e} - \frac{1}{n_j} \sum_{i\in\mathcal{C}_j} \sum_{h=0}^{H-1} \gamma\left(\boldsymbol{y}_j^t + \nabla F_i\left(\boldsymbol{x}_{i,h}^{t,e}, \xi_{i,h}^{t,e}\right)\right) + \frac{1}{N} \sum_{j=1}^{N} \frac{1}{n_j} \sum_{i\in\mathcal{C}_j} \sum_{h=0}^{H-1} \gamma \nabla F_i\left(\boldsymbol{x}_{i,h}^{t,e}, \xi_{i,h}^{t,e}\right)\right\|^2,$$

where we utilize $\frac{1}{N}\sum_{j=1}^N \boldsymbol{y}_j^t = \boldsymbol{0}$ and $\frac{1}{n_j}\sum_{i\in\mathcal{C}_j}\boldsymbol{z}_i^{t,e} = \boldsymbol{0}$. Next, we bound $\mathbb{E}\big\|\bar{\boldsymbol{x}}_j^{t,e+1} - \hat{\boldsymbol{x}}^{t,e+1}\big\|^2$ as follows

$$
\mathbb{E}\big\|\bar{\boldsymbol{x}}_j^{t,e+1} - \hat{\boldsymbol{x}}^{t,e+1}\big\|^2
$$

$$
\leq \left(1+\frac{1}{E-1}\right)\mathbb{E}\big\|\bar{\boldsymbol{x}}_j^{t,e} - \hat{\boldsymbol{x}}^{t,e}\big\|^2 + \gamma^2 E\mathbb{E}\Bigg\|\sum_{h=0}^{H-1}\boldsymbol{y}_j^t + \sum_{h=0}^{H-1}\frac{1}{n_j}\sum_{i\in\mathcal{C}_j}\left(\nabla F_i\left(\boldsymbol{x}_{i,h}^{t,e},\xi_{i,h}^{t,e}\right)\right)
$$

$$
\mp\sum_{h=0}^{H-1}\frac{1}{n_j}\sum_{i\in\mathcal{C}_j}\nabla F_i\left(\boldsymbol{x}_{i,h}^{t,e}\right) - \sum_{h=0}^{H-1}\frac{1}{N}\sum_{j=1}^N\frac{1}{n_j}\sum_{i\in\mathcal{C}_j}\nabla F_i\left(\boldsymbol{x}_{i,h}^{t,e},\xi_{i,h}^{t,e}\right) \mp \sum_{h=0}^{H-1}\frac{1}{N}\sum_{j=1}^N\frac{1}{n_j}\sum_{i\in\mathcal{C}_j}\nabla F_i\left(\boldsymbol{x}_{i,h}^{t,e}\right)\Bigg\|^2
$$

$$
\leq \left(1+\frac{1}{E-1}\right)\mathbb{E}\big\|\bar{\boldsymbol{x}}_j^{t,e} - \hat{\boldsymbol{x}}^{t,e}\big\|^2
$$

$$
+ 2\gamma^2 E\mathbb{E}\Bigg\|\sum_{h=0}^{H-1}\boldsymbol{y}_j^t + \sum_{h=0}^{H-1}\frac{1}{n_j}\sum_{i\in\mathcal{C}_j}\nabla F_i\left(\boldsymbol{x}_{i,h}^{t,e}\right) - \sum_{h=0}^{H-1}\frac{1}{N}\sum_{j=1}^N\frac{1}{n_j}\sum_{i\in\mathcal{C}_j}\nabla F_i\left(\boldsymbol{x}_{i,h}^{t,e}\right)\Bigg\|^2
$$

$$
+ 2\gamma^2 E\mathbb{E}\Bigg\|\sum_{h=0}^{H-1}\frac{1}{n_j}\sum_{i\in\mathcal{C}_j}\nabla F_i\left(\boldsymbol{x}_{i,h}^{t,e},\xi_{i,h}^{t,e}\right) - \sum_{h=0}^{H-1}\frac{1}{n_j}\sum_{i\in\mathcal{C}_j}\nabla F_i\left(\boldsymbol{x}_{i,h}^{t,e}\right)
$$

$$
- \sum_{h=0}^{H-1}\frac{1}{N}\sum_{j=1}^N\frac{1}{n_j}\sum_{i\in\mathcal{C}_j}\nabla F_i\left(\boldsymbol{x}_{i,h}^{t,e},\xi_{i,h}^{t,e}\right) + \sum_{h=0}^{H-1}\frac{1}{N}\sum_{j=1}^N\frac{1}{n_j}\sum_{i\in\mathcal{C}_j}\nabla F_i\left(\boldsymbol{x}_{i,h}^{t,e}\right)\Bigg\|^2,
$$

where the first inequality comes from Lemma F.1.2. We thus have

$$
\frac{1}{N}\sum_{j=1}^N\mathbb{E}\big\|\bar{\boldsymbol{x}}_j^{t,e+1} - \hat{\boldsymbol{x}}^{t,e+1}\big\|^2
$$

$$
\leq \left(1+\frac{1}{E-1}\right)\frac{1}{N}\sum_{j=1}^N\mathbb{E}\big\|\bar{\boldsymbol{x}}_j^{t,e} - \hat{\boldsymbol{x}}^{t,e}\big\|^2
$$

$$
+ 2\gamma^2 EH^2\frac{1}{N}\sum_{j=1}^N\frac{1}{H}\sum_{h=0}^{H-1}\mathbb{E}\Bigg\|\boldsymbol{y}_j^t + \frac{1}{n_j}\sum_{i\in\mathcal{C}_j}\nabla F_i\left(\boldsymbol{x}_{i,h}^{t,e}\right) - \frac{1}{N}\sum_{j=1}^N\frac{1}{n_j}\sum_{i\in\mathcal{C}_j}\nabla F_i\left(\boldsymbol{x}_{i,h}^{t,e}\right)\Bigg\|^2
$$

$$
+ 2\gamma^2 E\frac{N-1}{N^2}\sum_{j=1}^N\mathbb{E}\Bigg\|\sum_{h=0}^{H-1}\frac{1}{n_j}\sum_{i\in\mathcal{C}_j}\nabla F_i\left(\boldsymbol{x}_{i,h}^{t,e},\xi_{i,h}^{t,e}\right) - \sum_{h=0}^{H-1}\frac{1}{n_j}\sum_{i\in\mathcal{C}_j}\nabla F_i\left(\boldsymbol{x}_{i,h}^{t,e}\right)\Bigg\|^2 \qquad (31)
$$

$$
\leq \left(1+\frac{1}{E-1}\right)\frac{1}{N}\sum_{j=1}^N\mathbb{E}\big\|\bar{\boldsymbol{x}}_j^{t,e} - \hat{\boldsymbol{x}}^{t,e}\big\|^2 + 2\gamma^2 EH\frac{N-1}{N^2}\sum_{j=1}^N\frac{1}{n_j}\sigma^2
$$

$$
+ \underbrace{2\gamma^2 EH^2\frac{1}{N}\sum_{j=1}^N\frac{1}{H}\sum_{h=0}^{H-1}\mathbb{E}\Bigg\|\boldsymbol{y}_j^t + \frac{1}{n_j}\sum_{i\in\mathcal{C}_j}\nabla F_i\left(\boldsymbol{x}_{i,h}^{t,e}\right) - \frac{1}{N}\sum_{j=1}^N\frac{1}{n_j}\sum_{i\in\mathcal{C}_j}\nabla F_i\left(\boldsymbol{x}_{i,h}^{t,e}\right)\Bigg\|^2}_{T_3},
$$

where the first equality comes from Lemmas F.1.1 and the second inequality follows Lemmas F.1.3 and F.1.4. Additionally, we bound $T_3$ as

$$
T_3 = \frac{1}{H} \sum_{h=0}^{H-1} \frac{1}{N} \sum_{j=1}^{N} \mathbb{E} \left\| \boldsymbol{y}_j^t \mp \nabla f_j \left(\hat{\boldsymbol{x}}^{t,e}\right) \mp \nabla f \left(\hat{\boldsymbol{x}}^{t,e}\right) + \frac{1}{n_j} \sum_{i \in \mathcal{C}_j} \nabla F_i \left(\boldsymbol{x}_{i,h}^{t,e}\right) - \frac{1}{N} \sum_{j=1}^{N} \frac{1}{n_j} \sum_{i \in \mathcal{C}_j} \nabla F_i \left(\boldsymbol{x}_{i,h}^{t,e}\right) \right\|^2
$$

$$
\leq 2 \frac{1}{N} \sum_{j=1}^{N} \mathbb{E} \left\| \boldsymbol{y}_j^t + \nabla f_j \left(\hat{\boldsymbol{x}}^{t,e}\right) - \nabla f \left(\hat{\boldsymbol{x}}^{t,e}\right) \right\|^2 + 2 \frac{1}{H} \sum_{h=0}^{H-1} \frac{1}{N} \sum_{j=1}^{N} \mathbb{E} \left\| \frac{1}{n_j} \sum_{i \in \mathcal{C}_j} \nabla F_i \left(\boldsymbol{x}_{i,h}^{t,e}\right) - \nabla f_j \left(\hat{\boldsymbol{x}}^{t,e}\right) \right.
$$

$$
\left. - \left( \frac{1}{N} \sum_{j=1}^{N} \frac{1}{n_j} \sum_{i \in \mathcal{C}_j} \nabla F_i \left(\boldsymbol{x}_{i,h}^{t,e}\right) - \nabla f \left(\hat{\boldsymbol{x}}^{t,e}\right) \right) \right\|^2
$$

$$
\leq 2 \frac{1}{N} \sum_{j=1}^{N} \mathbb{E} \left\| \boldsymbol{y}_j^t + \nabla f_j \left(\hat{\boldsymbol{x}}^{t,e}\right) - \nabla f \left(\hat{\boldsymbol{x}}^{t,e}\right) \right\|^2 + 2 \frac{1}{H} \sum_{h=0}^{H-1} \frac{1}{N} \sum_{j=1}^{N} \mathbb{E} \left\| \frac{1}{n_j} \sum_{i \in \mathcal{C}_j} \nabla F_i \left(\boldsymbol{x}_{i,h}^{t,e}\right) - \nabla f_j \left(\hat{\boldsymbol{x}}^{t,e}\right) \right\|^2
$$

$$
\leq 4L^2 \frac{1}{N} \sum_{j=1}^{N} \frac{1}{n_j} \sum_{i \in \mathcal{C}_j} \frac{1}{H} \sum_{h=0}^{H-1} \mathbb{E} \left\| \boldsymbol{x}_{i,h}^{t,e} - \bar{\boldsymbol{x}}_j^{t,e} \right\|^2 + 4L^2 \frac{1}{N} \sum_{j=1}^{N} \mathbb{E} \left\| \bar{\boldsymbol{x}}_j^{t,e} - \hat{\boldsymbol{x}}^{t,e} \right\|^2
$$

$$
+ 2 \frac{1}{N} \sum_{j=1}^{N} \mathbb{E} \left\| \boldsymbol{y}_j^t + \nabla f_j \left(\hat{\boldsymbol{x}}^{t,e}\right) - \nabla f \left(\hat{\boldsymbol{x}}^{t,e}\right) \right\|^2,
$$

where the second inequality follows Lemma F.1.1 and the last inequality follows Assumption 1.

Plugging the derived upper bound of $T_3$ into (31) gives rise to

$$
\frac{1}{N} \sum_{j=1}^{N} \mathbb{E} \left\| \bar{\boldsymbol{x}}_j^{t,e+1} - \hat{\boldsymbol{x}}^{t,e+1} \right\|^2 \leq \left( 1 + \frac{1}{E-1} + 8\gamma^2 E H^2 L^2 \right) \frac{1}{N} \sum_{j=1}^{N} \mathbb{E} \left\| \bar{\boldsymbol{x}}_j^{t,e} - \hat{\boldsymbol{x}}^{t,e} \right\|^2 + 2\gamma^2 E H \frac{N-1}{N^2} \sum_{j=1}^{N} \frac{1}{n_j} \sigma^2
$$

$$
+ 4\gamma^2 E H^2 \frac{1}{N} \sum_{j=1}^{N} \mathbb{E} \left\| \boldsymbol{y}_j^t + \nabla f_j \left(\hat{\boldsymbol{x}}^{t,e}\right) - \nabla f \left(\hat{\boldsymbol{x}}^{t,e}\right) \right\|^2 + 8\gamma^2 E H^2 L^2 \frac{1}{N} \sum_{j=1}^{N} \frac{1}{n_j} \sum_{i \in \mathcal{C}_j} \frac{1}{H} \sum_{h=0}^{H-1} \mathbb{E} \left\| \boldsymbol{x}_{i,h}^{t,e} - \bar{\boldsymbol{x}}_j^{t,e} \right\|^2.
$$

Let $\rho_2 = 1 + \frac{1}{E-1} + 8\gamma^2 E H^2 L^2$. As $\gamma \leq \frac{1}{10 E H L}$, we have $\rho_2 \leq \rho_2^{E-1} \leq \left( 1 + \frac{1}{E-1} + \frac{1}{12(E-1)} \right)^{E-1} \leq e_0^{\frac{13}{12}} < 3$, where $e_0$ denotes Euler's number. Therefore, we have

$$
\frac{1}{N} \sum_{j=1}^{N} \mathbb{E} \left\| \bar{\boldsymbol{x}}_j^{t,e+1} - \hat{\boldsymbol{x}}^{t,e+1} \right\|^2
$$

$$
\leq \left( \sum_{\nu=0}^{e} \rho_2^{\nu} \right) 2\gamma^2 E H \frac{N-1}{N^2} \sum_{j=1}^{N} \frac{1}{n_j} \sigma^2 + 4 \max\{\rho_2^{\nu}\} \gamma^2 E H^2 \sum_{\nu=0}^{e} \frac{1}{N} \sum_{j=1}^{N} \mathbb{E} \left\| \boldsymbol{y}_j^t + \nabla f_j \left(\hat{\boldsymbol{x}}^{t,\nu}\right) - \nabla f \left(\hat{\boldsymbol{x}}^{t,\nu}\right) \right\|^2
$$

$$
+ 8 \max\{\rho_2^{\nu}\} \gamma^2 E H^2 L^2 \sum_{\nu=0}^{e} \frac{1}{N} \sum_{j=1}^{N} \frac{1}{n_j} \sum_{i \in \mathcal{C}_j} \frac{1}{H} \sum_{h=0}^{H-1} \mathbb{E} \left\| \boldsymbol{x}_{i,h}^{t,\nu} - \bar{\boldsymbol{x}}_j^{t,\nu} \right\|^2
$$

$$
\leq 6\gamma^2 (e+1) E H \frac{N-1}{N^2} \sum_{j=1}^{N} \frac{1}{n_j} \sigma^2 + 12\gamma^2 E H^2 \sum_{\nu=0}^{e} \frac{1}{N} \sum_{j=1}^{N} \mathbb{E} \left\| \boldsymbol{y}_j^t + \nabla f_j \left(\hat{\boldsymbol{x}}^{t,\nu}\right) - \nabla f \left(\hat{\boldsymbol{x}}^{t,\nu}\right) \right\|^2
$$

$$
+ 24\gamma^2 E H^2 L^2 \sum_{\nu=0}^{e} \frac{1}{N} \sum_{j=1}^{N} \frac{1}{n_j} \sum_{i \in \mathcal{C}_j} \frac{1}{H} \sum_{h=0}^{H-1} \mathbb{E} \left\| \boldsymbol{x}_{i,h}^{t,\nu} - \bar{\boldsymbol{x}}_j^{t,\nu} \right\|^2.
$$

(32)

Hence, for $D_t = \sum_{e=0}^{E-1} \frac{1}{N} \sum_{j=1}^{N} \mathbb{E}_t^e \left\| \bar{\boldsymbol{x}}_j^{t,e} - \hat{\boldsymbol{x}}^{t,e} \right\|^2$, we have

$$
D_t \leq 3\gamma^2 E^3 H \frac{N-1}{N^2} \sum_{j=1}^{N} \frac{1}{n_j} \sigma^2 + 12\gamma^2 E^2 H^2 \sum_{e=0}^{E-1} \frac{1}{N} \sum_{j=1}^{N} \mathbb{E} \left\| \boldsymbol{y}_j^t + \nabla f_j \left(\hat{\boldsymbol{x}}^{t,e}\right) - \nabla f \left(\hat{\boldsymbol{x}}^{t,e}\right) \right\|^2
$$

$$
+ 24\gamma^2 E^2 H^2 L^2 \sum_{e=0}^{E-1} \frac{1}{N} \sum_{j=1}^{N} \frac{1}{n_j} \sum_{i \in \mathcal{C}_j} \frac{1}{H} \sum_{h=0}^{H-1} \mathbb{E} \left\| \boldsymbol{x}_{i,h}^{t,e} - \bar{\boldsymbol{x}}_j^{t,e} \right\|^2.
$$

(33)

This completes the proof of Lemma F.2.3.

### F.3.4 Proof of Lemma F.2.4

For $e = 0$, $\boldsymbol{z}_i^{t,0} = -\nabla F_i\left(\boldsymbol{x}_{i,0}^{t,0}, \xi_{i,0}^{t,0}\right) + \frac{1}{n_j}\sum_{i \in \mathcal{C}_j}\nabla F_i\left(\boldsymbol{x}_{i,0}^{t,0}, \xi_{i,0}^{t,0}\right)$ where $\boldsymbol{x}_{i,0}^{t,0} = \bar{\boldsymbol{x}}_j^{t,0}$, we have

$$
\begin{aligned}
Z_j^{t,0} =& \frac{1}{n_j}\sum_{i \in \mathcal{C}_j}\mathbb{E}\left\|-\nabla F_i\left(\bar{\boldsymbol{x}}_j^{t,0}, \xi_{i,0}^{t,0}\right) + \frac{1}{n_j}\sum_{i \in \mathcal{C}_j}\nabla F_i\left(\bar{\boldsymbol{x}}_j^{t,0}, \xi_{i,0}^{t,0}\right) + \nabla F_i\left(\bar{\boldsymbol{x}}_j^{t,0}\right) - \nabla f_j\left(\bar{\boldsymbol{x}}_j^{t,0}\right)\right\|^2 \\
\leq& \frac{1}{n_j}\sum_{i \in \mathcal{C}_j}\mathbb{E}\left\|\nabla F_i\left(\bar{\boldsymbol{x}}_j^{t,0}\right) - \nabla F_i\left(\bar{\boldsymbol{x}}_j^{t,0}, \xi_{i,0}^{t,0}\right)\right\|^2 \leq \sigma^2,
\end{aligned}
\tag{34}
$$

where the first inequality follows Lemma F.1.1 and the second inequality holds due to Assumption 2.

In addition, when $e \geq 0$, $\boldsymbol{z}_i^{t,e}$ obeys the following iteration

$$
\begin{aligned}
\boldsymbol{z}_i^{t,e+1} =& \boldsymbol{z}_i^{t,e} + \frac{1}{H\gamma}\left(\boldsymbol{x}_{i,H}^{t,e} - \boldsymbol{x}_{i,0}^{t,e} - \frac{1}{n_j}\sum_{i \in \mathcal{C}_j}\left(\boldsymbol{x}_{i,H}^{t,e} - \boldsymbol{x}_{i,0}^{t,e}\right)\right) \\
=& \boldsymbol{z}_i^{t,e} - \frac{1}{H}\sum_{h=0}^{H-1}\left(\left(\nabla F_i\left(\boldsymbol{x}_{i,h}^{t,e}, \xi_{i,h}^{t,e}\right) + \boldsymbol{y}_j^t + \boldsymbol{z}_i^{t,e}\right) - \frac{1}{n_j}\sum_{i \in \mathcal{C}_j}\left(\nabla F_i\left(\boldsymbol{x}_{i,h}^{t,e}, \xi_{i,h}^{t,e}\right) + \boldsymbol{y}_j^t + \boldsymbol{z}_i^{t,e}\right)\right).
\end{aligned}
$$

As $\sum_{i \in \mathcal{C}_j}\boldsymbol{z}_i^{t,e} = \mathbf{0}$, we thus have

$$
\boldsymbol{z}_i^{t,e+1} = \frac{1}{H}\sum_{h=0}^{H-1}\left(\frac{1}{n_j}\sum_{i \in \mathcal{C}_j}\nabla F_i\left(\boldsymbol{x}_{i,h}^{t,e}, \xi_{i,h}^{t,e}\right) - \nabla F_i\left(\boldsymbol{x}_{i,h}^{t,e}, \xi_{i,h}^{t,e}\right)\right).
$$

To establish an upper bound for $\sum_{e=0}^{E-1}\frac{1}{N}\sum_{j=1}^{N}Z_j^{t,e+1}$, we start with bounding $Z_j^{t,e}$ as follows

$$
\begin{aligned}
Z_j^{t,e+1} =& \frac{1}{n_j}\sum_{i \in \mathcal{C}_j}\mathbb{E}\left\|\frac{1}{H}\sum_{h=0}^{H-1}\left(\frac{1}{n_j}\sum_{i \in \mathcal{C}_j}\nabla F_i\left(\boldsymbol{x}_{i,h}^{t,e}, \xi_{i,h}^{t,e}\right) - \nabla F_i\left(\boldsymbol{x}_{i,h}^{t,e}, \xi_{i,h}^{t,e}\right)\right) + \nabla F_i\left(\bar{\boldsymbol{x}}_j^{t,e+1}\right) - \nabla f_j\left(\bar{\boldsymbol{x}}_j^{t,e+1}\right)\right\|^2 \\
=& \frac{1}{n_j}\sum_{i \in \mathcal{C}_j}\mathbb{E}\left\|\nabla F_i\left(\bar{\boldsymbol{x}}_j^{t,e+1}\right) - \frac{1}{H}\sum_{h=0}^{H-1}\nabla F_i\left(\boldsymbol{x}_{i,h}^{t,e}, \xi_{i,h}^{t,e}\right) - \nabla f_j\left(\bar{\boldsymbol{x}}_j^{t,e+1}\right) + \frac{1}{n_j}\sum_{i \in \mathcal{C}_j}\left(\frac{1}{H}\sum_{h=0}^{H-1}\nabla F_i\left(\boldsymbol{x}_{i,h}^{t,e}, \xi_{i,h}^{t,e}\right)\right)\right\|^2 \\
\leq& \frac{1}{n_j}\sum_{i \in \mathcal{C}_j}\mathbb{E}\left\|\nabla F_i\left(\bar{\boldsymbol{x}}_j^{t,e+1}\right) - \frac{1}{H}\sum_{h=0}^{H-1}\nabla F_i\left(\boldsymbol{x}_{i,h}^{t,e}, \xi_{i,h}^{t,e}\right)\right\|^2 \\
=& \frac{1}{n_j}\sum_{i \in \mathcal{C}_j}\mathbb{E}\left\|\frac{1}{H}\sum_{h=0}^{H-1}\left(\nabla F_i\left(\bar{\boldsymbol{x}}_j^{t,e+1}\right) - \nabla F_i\left(\boldsymbol{x}_{i,h}^{t,e}, \xi_{i,h}^{t,e}\right) \mp \nabla F_i\left(\boldsymbol{x}_{i,h}^{t,e}\right)\right)\right\|^2 \\
\leq& 2\frac{1}{n_j}\sum_{i \in \mathcal{C}_j}\mathbb{E}\left\|\frac{1}{H}\sum_{h=0}^{H-1}\left(\nabla F_i\left(\bar{\boldsymbol{x}}_j^{t,e+1}\right) - \nabla F_i\left(\boldsymbol{x}_{i,h}^{t,e}\right)\right)\right\|^2 + 2\frac{1}{n_j}\sum_{i \in \mathcal{C}_j}\mathbb{E}\left\|\frac{1}{H}\sum_{h=0}^{H-1}\left(\nabla F_i\left(\boldsymbol{x}_{i,h}^{t,e}\right) - \nabla F_i\left(\boldsymbol{x}_{i,h}^{t,e}, \xi_{i,h}^{t,e}\right)\right)\right\|^2 \\
\leq& 2\frac{1}{n_j}\sum_{i \in \mathcal{C}_j}\frac{1}{H}\sum_{h=0}^{H-1}\mathbb{E}\left\|\left(\nabla F_i\left(\bar{\boldsymbol{x}}_j^{t,e+1}\right) - \nabla F_i\left(\boldsymbol{x}_{i,h}^{t,e}\right)\right)\right\|^2 + 2\frac{\sigma^2}{H} \\
\leq& 2L^2\frac{1}{n_j}\sum_{i \in \mathcal{C}_j}\frac{1}{H}\sum_{h=0}^{H-1}\mathbb{E}\left\|\bar{\boldsymbol{x}}_j^{t,e+1} \mp \bar{\boldsymbol{x}}_j^{t,e} - \boldsymbol{x}_{i,h}^{t,e}\right\|^2 + 2\frac{\sigma^2}{H} \\
\leq& 4L^2\frac{1}{H}\sum_{h=0}^{H-1}\frac{1}{n_j}\sum_{i \in \mathcal{C}_j}\mathbb{E}\left\|\boldsymbol{x}_{i,h}^{t,e} - \bar{\boldsymbol{x}}_j^{t,e}\right\|^2 + 4L^2\mathbb{E}\left\|\bar{\boldsymbol{x}}_j^{t,e+1} - \bar{\boldsymbol{x}}_j^{t,e}\right\|^2 + 2\frac{\sigma^2}{H},
\end{aligned}
\tag{35}
$$

where the first inequality comes from Lemma F.1.1, the third inequality follows Lemma F.1.4 and Assumption 2, and the fourth inequality follows Assumption 1. Combining this bound with (34), we thus obtain Lemma F.2.4, i.e.,

$$
\sum_{e=0}^{E-1}\frac{1}{N}\sum_{j=1}^{N}Z_j^{t,e} \leq 4L^2 Q_t + 4L^2\sum_{e=0}^{E-1}\frac{1}{N}\sum_{j=1}^{N}\Theta_j^{t,e} + 2\frac{E}{H}\sigma^2 + \sigma^2.
\tag{36}
$$

### F.3.5 Proof of Lemma F.2.5

**Part I ($t \geq 1$):** As $\sum_{i \in \mathcal{C}_j} z_i^{t,e} = \mathbf{0}$, the updating rule of $y_j^{t+1}$ can be simplified as

$$y_j^{t+1} = \frac{1}{EH} \sum_{\tau=0}^{E-1} \sum_{h=0}^{H-1} \frac{1}{N} \sum_{j=1}^{N} \frac{1}{n_j} \sum_{i \in \mathcal{C}_j} \nabla F_i \left( x_{i,h}^{t,\tau}, \xi_{i,h}^{t,\tau} \right) - \frac{1}{EH} \sum_{\tau=0}^{E-1} \sum_{h=0}^{H-1} \frac{1}{n_j} \sum_{i \in \mathcal{C}_j} \nabla F_i \left( x_{i,h}^{t,\tau}, \xi_{i,h}^{t,\tau} \right). \quad (37)$$

We next bound $Y_j^{t+1,e}$ as follows

$$
\begin{aligned}
Y_j^{t+1,e} =& \frac{1}{N} \sum_{j=1}^{N} \mathbb{E} \left\| y_j^{t+1} + \nabla f_j \left( \hat{x}^{t+1,e} \right) - \nabla f \left( \hat{x}^{t+1,e} \right) \right\|^2 \\
=& \frac{1}{N} \sum_{j=1}^{N} \mathbb{E} \left\| \frac{1}{EH} \sum_{\tau=0}^{E-1} \sum_{h=0}^{H-1} \frac{1}{N} \sum_{j=1}^{N} \frac{1}{n_j} \sum_{i \in \mathcal{C}_j} \nabla F_i \left( x_{i,h}^{t,\tau}, \xi_{i,h}^{t,\tau} \right) - \frac{1}{EH} \sum_{\tau=0}^{E-1} \sum_{h=0}^{H-1} \frac{1}{n_j} \sum_{i \in \mathcal{C}_j} \nabla F_i \left( x_{i,h}^{t,\tau}, \xi_{i,h}^{t,\tau} \right) \right. \\
& \left. + \nabla f_j \left( \hat{x}^{t+1,e} \right) - \nabla f \left( \hat{x}^{t+1,e} \right) \right\|^2 \\
\leq& \frac{1}{N} \sum_{j=1}^{N} \mathbb{E} \left\| \nabla f_j \left( \hat{x}^{t+1,e} \right) - \frac{1}{EH} \sum_{\tau=0}^{E-1} \sum_{h=0}^{H-1} \frac{1}{n_j} \sum_{i \in \mathcal{C}_j} \left( \nabla F_i \left( x_{i,h}^{t,\tau}, \xi_{i,h}^{t,\tau} \right) \mp \nabla F_i \left( x_{i,h}^{t,\tau} \right) \right) \right\|^2 \\
\leq& \frac{2}{N} \sum_{j=1}^{N} \mathbb{E} \left\| \nabla f_j \left( \hat{x}^{t+1,e} \right) - \frac{1}{EH} \sum_{\tau=0}^{E-1} \sum_{h=0}^{H-1} \frac{1}{n_j} \sum_{i \in \mathcal{C}_j} \nabla F_i \left( x_{i,h}^{t,\tau} \right) \right\|^2 \\
& + \frac{2}{NE^2H^2} \sum_{j=1}^{N} \frac{1}{n_j^2} \sum_{i \in \mathcal{C}_j} \mathbb{E} \left\| \sum_{\tau=0}^{E-1} \sum_{h=0}^{H-1} \left( \nabla F_i \left( x_{i,h}^{t,\tau}, \xi_{i,h}^{t,\tau} \right) - \nabla F_i \left( x_{i,h}^{t,\tau} \right) \right) \right\|^2 \\
\leq& \frac{2}{NEH} \sum_{\tau=0}^{E-1} \sum_{h=0}^{H-1} \sum_{j=1}^{N} \mathbb{E} \left\| \frac{1}{n_j} \sum_{i \in \mathcal{C}_j} \nabla F_i \left( x_{i,h}^{t,\tau} \right) - \nabla f_j \left( \hat{x}^{t+1,e} \right) \right\|^2 + \frac{2}{EH} \frac{1}{N} \sum_{j=1}^{N} \frac{1}{n_j} \sigma^2 \\
\leq& \frac{2}{NEH} \sum_{\tau=0}^{E-1} \sum_{h=0}^{H-1} \sum_{j=1}^{N} \frac{1}{n_j} \sum_{i \in \mathcal{C}_j} \mathbb{E} \left\| \nabla F_i \left( x_{i,h}^{t,\tau} \right) - \nabla F_i \left( \hat{x}^{t+1,e} \right) \right\|^2 + \frac{2}{EH} \frac{1}{N} \sum_{j=1}^{N} \frac{1}{n_j} \sigma^2 \\
\leq& \frac{2L^2}{NEH} \sum_{\tau=0}^{E-1} \sum_{h=0}^{H-1} \sum_{j=1}^{N} \frac{1}{n_j} \sum_{i \in \mathcal{C}_j} \mathbb{E} \left\| x_{i,h}^{t,\tau} - \hat{x}^{t+1,e} \right\|^2 + \frac{2}{EH} \frac{1}{N} \sum_{j=1}^{N} \frac{1}{n_j} \sigma^2,
\end{aligned}
$$
$$(38)$$

where the first inequality holds due to Lemma F.1.1, the second inequality follows Lemmas F.1.1 and F.1.3, the third inequality follows Lemmas F.1.1 and F.1.4 and Assumption 2, the fourth inequality follows Lemma F.1.1, and the final one comes from Assumption 1.

Given the above inequality, it follows that

$$\sum_{e=0}^{E-1} \frac{1}{N} \sum_{j=1}^{N} Y_j^{t+1,e} \leq 2L^2 \sum_{e=0}^{E-1} \frac{1}{NEH} \sum_{\tau=0}^{E-1} \sum_{h=0}^{H-1} \sum_{j=1}^{N} \frac{1}{n_j} \sum_{i \in \mathcal{C}_j} \mathbb{E} \left\| x_{i,h}^{t,\tau} - \hat{x}^{t+1,e} \right\|^2 + \frac{2}{H} \frac{1}{N} \sum_{j=1}^{N} \frac{1}{n_j} \sigma^2. \quad (39)$$

Additionally,

$$
\begin{aligned}
\mathbb{E} \left\| x_{i,h}^{t,\tau} - \hat{x}^{t+1,e} \right\|^2 \leq& \mathbb{E} \left\| x_{i,h}^{t,\tau} \mp \bar{x}_j^{t,\tau} \mp \hat{x}^{t,\tau} \mp \bar{x}^{t+1} - \hat{x}^{t+1,e} \right\|^2 \\
\leq& 4\mathbb{E} \left\| x_{i,h}^{t,\tau} - \bar{x}_j^{t,\tau} \right\|^2 + 4\mathbb{E} \left\| \bar{x}_j^{t,\tau} - \hat{x}^{t,\tau} \right\|^2 \\
& + 4\mathbb{E} \left\| \hat{x}^{t,\tau} - \bar{x}^{t+1} \right\|^2 + 4\mathbb{E} \left\| \bar{x}^{t+1} - \hat{x}^{t+1,e} \right\|^2,
\end{aligned}
$$
$$(40)$$

which implies that we need further to bound $\mathbb{E} \left\| \hat{x}^{t,\tau} - \bar{x}^{t+1} \right\|^2$ and $\mathbb{E} \left\| \bar{x}^{t+1} - \hat{x}^{t+1,e} \right\|^2$.

Under the framework of MTGC, the virtual global model obeys the following iteration:

$$\hat{x}^{t+1,e} = \bar{x}^{t+1} - \gamma \sum_{j=1}^{N} \frac{1}{n_j} \sum_{i \in \mathcal{C}_j} \sum_{\tau=0}^{e-1} \sum_{h=0}^{H-1} \left( \nabla F_i \left( x_{i,h}^{t+1,\tau}, \xi_{i,h}^{t+1,\tau} \right) + z_i^{t+1,\tau} + y_j^{t+1} \right).$$

As $\sum_{i\in\mathcal{C}_j} \boldsymbol{z}_i^{t+1,\tau} = \boldsymbol{0}$ and $\sum_{j=1}^{N} \boldsymbol{y}_j^{t+1} = \boldsymbol{0}$, the the virtual global model iteration reduces to

$$\hat{\boldsymbol{x}}^{t+1,e} = \bar{\boldsymbol{x}}^{t+1} - \gamma \sum_{j=1}^{N} \frac{1}{n_j} \sum_{i\in\mathcal{C}_j} \sum_{\tau=0}^{e-1} \sum_{h=0}^{H-1} \nabla F_i\left(\boldsymbol{x}_{i,h}^{t+1,\tau}, \xi_{i,h}^{t+1,\tau}\right).$$

For $\mathbb{E}\left\|\bar{\boldsymbol{x}}^{t+1} - \hat{\boldsymbol{x}}^{t+1,e}\right\|^2$, we have

$$
\begin{aligned}
&\mathbb{E}\left\|\bar{\boldsymbol{x}}^{t+1} - \hat{\boldsymbol{x}}^{t+1,e}\right\|^2 \\
=&\gamma^2 \mathbb{E}\left\|\frac{1}{N}\sum_{j=1}^{N}\frac{1}{n_j}\sum_{i\in\mathcal{C}_j}\sum_{\tau=0}^{e-1}\sum_{h=0}^{H-1}\nabla F_i(\boldsymbol{x}_{i,h}^{t+1,\tau},\xi_{i,h}^{t+1,\tau}) \mp \frac{1}{N}\sum_{j=1}^{N}\frac{1}{n_j}\sum_{i\in\mathcal{C}_j}\sum_{\tau=0}^{e-1}\sum_{h=0}^{H-1}\nabla F_i(\boldsymbol{x}_{i,h}^{t+1,\tau})\right\|^2 \\
\leq&2\gamma^2\mathbb{E}\left\|\frac{1}{N}\sum_{j=1}^{N}\frac{1}{n_j}\sum_{i\in\mathcal{C}_j}\sum_{\tau=0}^{e-1}\sum_{h=0}^{H-1}\nabla F_i(\boldsymbol{x}_{i,h}^{t+1,\tau},\xi_{i,h}^{t+1,\tau}) - \frac{1}{N}\sum_{j=1}^{N}\frac{1}{n_j}\sum_{i\in\mathcal{C}_j}\sum_{\tau=0}^{e-1}\sum_{h=0}^{H-1}\nabla F_i(\boldsymbol{x}_{i,h}^{t+1,\tau})\right\|^2 \\
&+2\gamma^2\mathbb{E}\left\|\frac{1}{N}\sum_{j=1}^{N}\frac{1}{n_j}\sum_{i\in\mathcal{C}_j}\sum_{\tau=0}^{e-1}\sum_{h=0}^{H-1}\nabla F_i(\boldsymbol{x}_{i,h}^{t+1,\tau})\right\|^2 \\
=&2\gamma^2\frac{1}{N^2}\sum_{j=1}^{N}\frac{1}{n_j^2}\sum_{i\in\mathcal{C}_j}\mathbb{E}\left\|\sum_{\tau=0}^{e-1}\sum_{h=0}^{H-1}\nabla F_i(\boldsymbol{x}_{i,h}^{t+1,\tau},\xi_{i,h}^{t+1,\tau}) - \sum_{\tau=0}^{e-1}\sum_{h=0}^{H-1}\nabla F_i(\boldsymbol{x}_{i,h}^{t+1,\tau})\right\|^2 \\
&+2\gamma^2\mathbb{E}\left\|\frac{1}{N}\sum_{j=1}^{N}\frac{1}{n_j}\sum_{i\in\mathcal{C}_j}\sum_{\tau=0}^{e-1}\sum_{h=0}^{H-1}\nabla F_i(\boldsymbol{x}_{i,h}^{t+1,\tau})\right\|^2 \\
\leq&2\gamma^2 eH\sum_{\tau=0}^{e-1}\sum_{h=0}^{H-1}\mathbb{E}\left\|\frac{1}{N}\sum_{j=1}^{N}\frac{1}{n_j}\sum_{i\in\mathcal{C}_j}\nabla F_i(\boldsymbol{x}_{i,h}^{t+1,\tau}) \mp \frac{1}{N}\sum_{j=1}^{N}\nabla f_j(\bar{\boldsymbol{x}}_j^{t+1,\tau}) \mp \nabla f(\hat{\boldsymbol{x}}^{t+1,\tau})\right\|^2 + 2\gamma^2\frac{EH}{N^2}\sum_{j=1}^{N}\frac{1}{n_j}\sigma^2 \\
\leq&6\gamma^2 eH\sum_{\tau=0}^{e-1}\sum_{h=0}^{H-1}\frac{1}{N}\sum_{j=1}^{N}\mathbb{E}\left\|\frac{1}{n_j}\sum_{i\in\mathcal{C}_j}\nabla F_i(\boldsymbol{x}_{i,h}^{t+1,\tau}) - \nabla f_j(\bar{\boldsymbol{x}}_j^{t+1,\tau})\right\|^2 + 2\gamma^2\frac{EH}{N^2}\sum_{j=1}^{N}\frac{1}{n_j}\sigma^2 \\
&+6\gamma^2 eH\sum_{\tau=0}^{e-1}\sum_{h=0}^{H-1}\mathbb{E}\left\|\frac{1}{N}\sum_{j=1}^{N}\nabla f_j(\bar{\boldsymbol{x}}_j^{t+1,\tau}) - \nabla f(\hat{\boldsymbol{x}}^{t+1,\tau})\right\|^2 + 6\gamma^2 eH\sum_{\tau=0}^{e-1}\sum_{h=0}^{H-1}\mathbb{E}\left\|\nabla f(\hat{\boldsymbol{x}}^{t+1,\tau})\right\|^2 \\
\leq&6\gamma^2 L^2 EH\sum_{\tau=0}^{E-1}\sum_{h=0}^{H-1}\frac{1}{N}\sum_{j=1}^{N}\frac{1}{n_j}\sum_{i\in\mathcal{C}_j}\mathbb{E}\left\|\boldsymbol{x}_{i,h}^{t+1,\tau} - \bar{\boldsymbol{x}}_j^{t+1,\tau}\right\|^2 + 6\gamma^2 L^2 EH^2\sum_{\tau=0}^{E-1}\frac{1}{N}\sum_{j=1}^{N}\mathbb{E}\left\|\bar{\boldsymbol{x}}_j^{t+1,\tau} - \hat{\boldsymbol{x}}^{t+1,\tau}\right\|^2 \\
&+6\gamma^2 EH^2\sum_{\tau=0}^{E-1}\mathbb{E}\left\|\nabla f(\hat{\boldsymbol{x}}^{t+1,\tau})\right\|^2 + 2\gamma^2\frac{EH}{N^2}\sum_{j=1}^{N}\frac{1}{n_j}\sigma^2,
\end{aligned}
\tag{41}
$$

where the second equality holds due to Lemma F.1.3, the second inequality follows Lemma F.1.4 and Assumption 2, the third inequality comes from Lemma F.1.1, the last inequality follows Lemma F.1.1 and Assumption 1.

Similarly, we can bound $\mathbb{E}\left\|\hat{\boldsymbol{x}}^{t,\tau} - \bar{\boldsymbol{x}}^{t+1}\right\|^2$ as

$$
\begin{aligned}
&\mathbb{E}\left\|\hat{\boldsymbol{x}}^{t,\tau} - \bar{\boldsymbol{x}}^{t+1}\right\|^2 \\
\leq&6\gamma^2 L^2 EH\sum_{\tau'=0}^{E-1}\sum_{h=0}^{H-1}\frac{1}{N}\sum_{j=1}^{N}\frac{1}{n_j}\sum_{i\in\mathcal{C}_j}\mathbb{E}\left\|\boldsymbol{x}_{i,h}^{t,\tau'} - \bar{\boldsymbol{x}}_j^{t,\tau'}\right\|^2 + 6\gamma^2 L^2 EH^2\sum_{\tau'=0}^{E-1}\frac{1}{N}\sum_{j=1}^{N}\mathbb{E}\left\|\bar{\boldsymbol{x}}_j^{t,\tau'} - \hat{\boldsymbol{x}}^{t,\tau'}\right\|^2 \\
&+6\gamma^2 EH^2\sum_{\tau'=0}^{E-1}\mathbb{E}\left\|\nabla f(\hat{\boldsymbol{x}}^{t,\tau'})\right\|^2 + 2\gamma^2\frac{EH}{N^2}\sum_{j=1}^{N}\frac{1}{n_j}\sigma^2.
\end{aligned}
\tag{42}
$$

Given (41) and (42), it follows that

$$
\mathbb{E}\left\|\hat{\boldsymbol{x}}^{t,\tau}-\bar{\boldsymbol{x}}^{t+1}\right\|^2+\mathbb{E}\left\|\bar{\boldsymbol{x}}^{t+1}-\hat{\boldsymbol{x}}^{t+1,e}\right\|^2
$$

$$
\leq 6\gamma^2 L^2 EH^2 Q_{t+1}+6\gamma^2 L^2 EH^2 D_{t+1}+6\gamma^2 EH^2\sum_{\tau=0}^{E-1}\mathbb{E}\left\|\nabla f(\hat{\boldsymbol{x}}^{t+1,\tau})\right\|^2+2\gamma^2\frac{EH}{N^2}\sum_{j=1}^{N}\frac{1}{n_j}\sigma^2
$$

$$
+6\gamma^2 L^2 EH^2 Q_t+6\gamma^2 L^2 EH^2 D_t+6\gamma^2 EH^2\sum_{\tau=0}^{E-1}\mathbb{E}\left\|\nabla f(\hat{\boldsymbol{x}}^{t,\tau})\right\|^2+2\gamma^2\frac{EH}{N^2}\sum_{j=1}^{N}\frac{1}{n_j}\sigma^2. \tag{43}
$$

Combining (39) and (40), we can obtain

$$
\sum_{e=0}^{E-1}\frac{1}{N}\sum_{j=1}^{N}Y_j^{t+1,e}
$$

$$
\leq 2L^2\sum_{e=0}^{E-1}\frac{1}{NEH}\sum_{\tau=0}^{E-1}\sum_{h=0}^{H-1}\sum_{j=1}^{N}\frac{1}{n_j}\sum_{i\in\mathcal{C}_j}\mathbb{E}\left\|\boldsymbol{x}_{i,h}^{t,\tau}-\hat{\boldsymbol{x}}^{t+1,e}\right\|^2+\frac{2}{H}\frac{1}{N}\sum_{j=1}^{N}\frac{1}{n_j}\sigma^2
$$

$$
\leq 8L^2\frac{1}{NH}\sum_{\tau=0}^{E-1}\sum_{h=0}^{H-1}\sum_{j=1}^{N}\frac{1}{n_j}\sum_{i\in\mathcal{C}_j}\mathbb{E}\left\|\boldsymbol{x}_{i,h}^{t,\tau}-\bar{\boldsymbol{x}}_j^{t,\tau}\right\|^2+8L^2\frac{1}{N}\sum_{\tau=0}^{E-1}\sum_{j=1}^{N}\mathbb{E}\left\|\bar{\boldsymbol{x}}_j^{t,\tau}-\hat{\boldsymbol{x}}^{t,\tau}\right\|^2 \tag{44}
$$

$$
+8L^2\sum_{\tau=0}^{E-1}\mathbb{E}\left\|\hat{\boldsymbol{x}}^{t,\tau}-\bar{\boldsymbol{x}}^{t+1}\right\|^2+8L^2\sum_{e=0}^{E-1}\mathbb{E}\left\|\bar{\boldsymbol{x}}^{t+1}-\hat{\boldsymbol{x}}^{t+1,e}\right\|^2+\frac{2}{H}\frac{1}{N}\sum_{j=1}^{N}\frac{1}{n_j}\sigma^2.
$$

Plugging (43) into (44), we have

$$
\sum_{e=0}^{E-1}\frac{1}{N}\sum_{j=1}^{N}Y_j^{t+1,e}\leq 8L^2 Q_t+8L^2 D_t+32\gamma^2 L^2 E^2 H\frac{1}{N^2}\sum_{j=1}^{N}\frac{1}{n_j}\sigma^2+\frac{2}{H}\frac{1}{N}\sum_{j=1}^{N}\frac{1}{n_j}\sigma^2
$$

$$
+48\gamma^2 L^4 E^2 H^2 Q_t+48\gamma^2 L^4 E^2 H^2 D_t+48\gamma^2 L^2 E^2 H^2\sum_{\tau=0}^{E-1}\mathbb{E}\left\|\nabla f(\hat{\boldsymbol{x}}^{t,\tau})\right\|^2
$$

$$
+48\gamma^2 L^4 E^2 H^2 Q_{t+1}+48\gamma^2 L^4 E^2 H^2 D_{t+1}+48\gamma^2 L^2 E^2 H^2\sum_{\tau=0}^{E-1}\mathbb{E}\left\|\nabla f(\hat{\boldsymbol{x}}^{t+1,\tau})\right\|^2.
$$

**Part II** ($t=0$): On the other hand, when $t=0$, we have

$$
\sum_{e=0}^{E-1}\frac{1}{N}\sum_{j=1}^{N}Y_j^{0,e}
$$

$$
=\sum_{e=0}^{E-1}\frac{1}{N}\sum_{j=1}^{N}\mathbb{E}\left\|-\frac{1}{n_j}\sum_{i\in\mathcal{C}_j}\nabla F_i(\bar{\boldsymbol{x}}^0,\xi_{i,0}^{0,0})+\frac{1}{N}\sum_{j=1}^{N}\frac{1}{n_j}\sum_{i\in\mathcal{C}_j}\nabla F_i(\bar{\boldsymbol{x}}^0,\xi_{i,0}^{t,0})+\nabla f_j\left(\hat{\boldsymbol{x}}^{0,e}\right)-\nabla f\left(\hat{\boldsymbol{x}}^{0,e}\right)\right\|^2
$$

$$
\leq\sum_{e=0}^{E-1}\frac{1}{N}\sum_{j=1}^{N}\mathbb{E}\left\|-\frac{1}{n_j}\sum_{i\in\mathcal{C}_j}\nabla F_i(\bar{\boldsymbol{x}}^0,\xi_{i,0}^{0,0})+\nabla f_j\left(\bar{\boldsymbol{x}}^0\right)+\frac{1}{N}\sum_{j=1}^{N}\frac{1}{n_j}\sum_{i\in\mathcal{C}_j}\nabla F_i(\bar{\boldsymbol{x}}^0,\xi_{i,0}^{t,0})-\nabla f\left(\bar{\boldsymbol{x}}^0\right)\right\|^2
$$

$$
+\sum_{e=0}^{E-1}\frac{1}{N}\sum_{j=1}^{N}\mathbb{E}\left\|\nabla f_j\left(\hat{\boldsymbol{x}}^{0,e}\right)-\nabla f_j\left(\bar{\boldsymbol{x}}^0\right)-\nabla f\left(\hat{\boldsymbol{x}}^{0,e}\right)+\nabla f\left(\bar{\boldsymbol{x}}^0\right)\right\|^2
$$

$$
\leq E\frac{1}{N}\sum_{j=1}^{N}\frac{1}{n_j}\sigma^2+L^2\sum_{e=0}^{E-1}\mathbb{E}\left\|\hat{\boldsymbol{x}}^{0,e}-\bar{\boldsymbol{x}}^0\right\|^2.
$$

Similar to (41), we have

$$
\sum_{e=0}^{E-1} \mathbb{E}\left\|\hat{\boldsymbol{x}}^{0,e} - \bar{\boldsymbol{x}}^0\right\|^2
$$

$$
\leq 6\gamma^2 L^2 E^2 H \sum_{\tau=0}^{E-1}\sum_{h=0}^{H-1}\frac{1}{N}\sum_{j=1}^{N}\frac{1}{n_j}\sum_{i\in\mathcal{C}_j}\mathbb{E}\left\|\boldsymbol{x}_{i,h}^{0,\tau}-\bar{\boldsymbol{x}}_j^{0,\tau}\right\|^2 + 6\gamma^2 L^2 E^2 H^2 \sum_{\tau=0}^{E-1}\frac{1}{N}\sum_{j=1}^{N}\mathbb{E}\left\|\bar{\boldsymbol{x}}_j^{0,\tau}-\hat{\boldsymbol{x}}_j^{0,\tau}\right\|^2
$$

$$
+ 6\gamma^2 E^2 H^2 \sum_{\tau=0}^{E-1}\mathbb{E}\left\|\nabla f(\hat{\boldsymbol{x}}^{0,\tau})\right\|^2 + 2\gamma^2 \frac{E^2 H}{N^2}\sum_{j=1}^{N}\frac{1}{n_j}\sigma^2.
$$

Combining the above two inequalities gives rise to

$$
\sum_{e=0}^{E-1}\frac{1}{N}\sum_{j=1}^{N}Y_j^{0,e} \leq E\frac{1}{N}\sum_{j=1}^{N}\frac{1}{n_j}\sigma^2 + 6\gamma^2 L^4 E^2 H^2\left(Q_0 + D_0\right) + 6\gamma^2 L^2 E^2 H^2 \sum_{e=0}^{E-1}\mathbb{E}\left\|\nabla f(\hat{\boldsymbol{x}}^{0,e})\right\|^2
$$

$$
+ 2\gamma^2 L^2 \frac{E^2 H}{N^2}\sum_{j=1}^{N}\frac{1}{n_j}\sigma^2.
$$

To this end, we complete the proof of Lemma F.2.5.

### F.3.6 Proof of Lemma F.2.7

**Part I** ($t \geq 1$): Replacing $\sum_{e=0}^{E-1}\frac{1}{N}\sum_{j=1}^{N}\Theta_j^{t,e}$ in (15) with an upper bound established in Lemma F.2.6 (i.e., (18)), and then plugging it into (13), we have

$$
\begin{aligned}
Q_t \leq{} & 24\gamma^2 H^2 L^2 D_t + 12\gamma^2 H^2\Bigg\{4L^2 Q_t + 4L^2\Bigg(8\gamma^2 H^2 L^2 Q_t + 8\gamma^2 H^2 L^2 D_t + 8\gamma^2 H^2 \sum_{e=0}^{E-1}\frac{1}{N}\sum_{j=1}^{N}Y_j^{t,e}\\
& + 8\gamma^2 H^2 \sum_{e=0}^{E-1}\mathbb{E}\left\|\nabla f\left(\hat{\boldsymbol{x}}^{t,e}\right)\right\|^2 + 2\gamma^2 EH\frac{1}{N}\sum_{j=1}^{N}\frac{1}{n_j}\sigma^2\Bigg) + 2\frac{E}{H}\sigma^2 + {\color{red}\sigma^2}\Bigg\}\\
& + 12\gamma^2 H^2 \sum_{e=0}^{E-1}\frac{1}{N}\sum_{j=1}^{N}Y_j^{t,e} + 24\gamma^2 H^2 \sum_{e=0}^{E-1}\mathbb{E}\left\|\nabla f\left(\hat{\boldsymbol{x}}^{t,e}\right)\right\|^2 + 3EH\gamma^2\sigma^2\\
={} & \left(48\gamma^2 H^2 L^2 + 384\gamma^4 H^4 L^4\right)Q_t + \left(24\gamma^2 H^2 L^2 + 384\gamma^4 H^4 L^4\right)D_t\\
& + \left(384\gamma^4 H^4 L^2 + 24\gamma^2 H^2\right)\sum_{e=0}^{E-1}\mathbb{E}\left\|\nabla f\left(\hat{\boldsymbol{x}}^{t,e}\right)\right\|^2 + \left(384\gamma^4 H^4 L^2 + 12\gamma^2 H^2\right)\sum_{e=0}^{E-1}\frac{1}{N}\sum_{j=1}^{N}Y_j^{t,e}\\
& + 96\gamma^4 EH^3 L^2\frac{1}{N}\sum_{j=1}^{N}\frac{1}{n_j}\sigma^2 + 27\gamma^2 EH\sigma^2 + 12\gamma^2 H^2\sigma^2.
\end{aligned}
\tag{45}
$$

Summing (45) with the inequality established in Lemma F.2.3 and utilizing the upper bound of $\sum_{e=0}^{E-1}\frac{1}{N}\sum_{j=1}^{N}Y_j^{t,e}$ established in Lemma F.2.5, we have

$$
\begin{aligned}
Q_t + D_t \leq{} & \left(48\gamma^2 H^2 L^2 + 384\gamma^4 H^4 L^4 + 24\gamma^2 E^2 H^2 L^2\right)Q_t + \left(24\gamma^2 H^2 L^2 + 384\gamma^4 H^4 L^4\right)D_t\\
& + \left(384\gamma^4 H^4 L^2 + 24\gamma^2 H^2\right)\sum_{e=0}^{E-1}\mathbb{E}\left\|\nabla f\left(\hat{\boldsymbol{x}}^{t,e}\right)\right\|^2\\
& + \left(384\gamma^4 H^4 L^2 + 12\gamma^2 H^2 + 12\gamma^2 E^2 H^2\right)\Bigg\{\left(8L^2 + 48\gamma^2 L^4 E^2 H^2\right)\left(Q_{t-1} + D_{t-1}\right)\\
& + 48\gamma^2 L^4 E^2 H^2\left(Q_t + D_t\right) + 48\gamma^2 L^2 E^2 H^2 \sum_{\tau=0}^{E-1}\left(\mathbb{E}\left\|\nabla f(\hat{\boldsymbol{x}}^{t-1,\tau})\right\|^2 + \mathbb{E}\left\|\nabla f(\hat{\boldsymbol{x}}^{t,\tau})\right\|^2\right)\\
& + 32\gamma^2 L^2 E^2 H\frac{1}{N^2}\sum_{j=1}^{N}\frac{1}{n_j}\sigma^2 + \frac{2}{H}\frac{1}{N}\sum_{j=1}^{N}\frac{1}{n_j}\sigma^2\Bigg\}\\
& + 96\gamma^4 EH^3 L^2\frac{1}{N}\sum_{j=1}^{N}\frac{1}{n_j}\sigma^2 + 27\gamma^2 EH\sigma^2 + 12\gamma^2 H^2\sigma^2 + 3\gamma^2 E^3 H\frac{N-1}{N^2}\sum_{j=1}^{N}\frac{1}{n_j}\sigma^2.
\end{aligned}
$$

Reorganizing the above inequality gives rise to

$$Q_t + D_t$$
$$\leq \left(48\gamma^2 H^2 L^2 + 384\gamma^4 H^4 L^4 + 24\gamma^2 E^2 H^2 L^2 + \left(384\gamma^4 H^4 L^2 + 12\gamma^2 H^2 + 12\gamma^2 E^2 H^2\right) \times 48\gamma^2 L^4 E^2 H^2\right)$$
$$\quad \times (Q_t + D_t) + \left(384\gamma^4 H^4 L^2 + 12\gamma^2 H^2 + 12\gamma^2 E^2 H^2\right)\left(8L^2 + 48\gamma^2 L^4 E^2 H^2\right)(Q_{t-1} + D_{t-1})$$
$$\quad + \left(384\gamma^4 H^4 L^2 + 24\gamma^2 H^2 + \left(384\gamma^4 H^4 L^2 + 12\gamma^2 H^2 + 12\gamma^2 E^2 H^2\right) \times 48\gamma^2 L^2 E^2 H^2\right)$$
$$\quad \times \sum_{\tau=0}^{E-1}\left(\mathbb{E}\big\|\nabla f(\hat{\boldsymbol{x}}^{t-1,\tau})\big\|^2 + \mathbb{E}\big\|\nabla f(\hat{\boldsymbol{x}}^{t,\tau})\big\|^2\right)$$
$$\quad + \left(384\gamma^4 H^4 L^2 + 12\gamma^2 H^2 + 12\gamma^2 E^2 H^2\right)\left\{32\gamma^2 L^2 E^2 H \frac{1}{N^2}\sum_{j=1}^{N}\frac{1}{n_j}\sigma^2 + \frac{2}{H}\frac{1}{N}\sum_{j=1}^{N}\frac{1}{n_j}\sigma^2\right\}$$
$$\quad + 96\gamma^4 EH^3 L^2 \frac{1}{N}\sum_{j=1}^{N}\frac{1}{n_j}\sigma^2 + 27\gamma^2 EH\sigma^2 + 12\gamma^2 H^2\sigma^2 + 3\gamma^2 E^3 H\frac{N-1}{N^2}\sum_{j=1}^{N}\frac{1}{n_j}\sigma^2.$$

When $\gamma \leq \frac{1}{7EHL}$, we have $48\gamma^2 L^2 E^2 H^2 \leq 1$ and

$$\left(1 - \left(60\gamma^2 H^2 L^2 + 768\gamma^4 H^4 L^4 + 36\gamma^2 E^2 H^2 L^2\right)\right)(Q_t + D_t)$$
$$\leq \left(3840\gamma^4 H^4 L^4 + 120\gamma^2 H^2 L^2 + 120\gamma^2 E^2 H^2 L^2\right)(Q_{t-1} + D_{t-1})$$
$$\quad + \left(768\gamma^4 H^4 L^2 + 36\gamma^2 H^2 + 12\gamma^2 E^2 H^2\right)\sum_{\tau=0}^{E-1}\left(\mathbb{E}\big\|\nabla f(\hat{\boldsymbol{x}}^{t-1,\tau})\big\|^2 + \mathbb{E}\big\|\nabla f(\hat{\boldsymbol{x}}^{t,\tau})\big\|^2\right) \tag{46}$$
$$\quad + \left(384\gamma^4 H^4 L^2 + 12\gamma^2 H^2 + 12\gamma^2 E^2 H^2\right)\frac{3}{H}\frac{1}{N}\sum_{j=1}^{N}\frac{1}{n_j}\sigma^2$$
$$\quad + 96\gamma^4 EH^3 L^2 \frac{1}{N}\sum_{j=1}^{N}\frac{1}{n_j}\sigma^2 + 27\gamma^2 EH\sigma^2 + 12\gamma^2 H^2\sigma^2 + 3\gamma^2 E^3 H\frac{N-1}{N^2}\sum_{j=1}^{N}\frac{1}{n_j}\sigma^2.$$

As $\gamma \leq \frac{1}{33EHL}$, we have $1 - \left(60\gamma^2 H^2 L^2 + 768\gamma^4 H^4 L^4 + 36\gamma^2 E^2 H^2 L^2\right) \geq \frac{2}{3}$ and $3840\gamma^4 H^4 L^4 + 120\gamma^2 H^2 L^2 + 120\gamma^2 E^2 H^2 L^2 \leq \frac{1}{3}$. Hence, (46) can be simplified as

$$Q_t + D_t \leq \frac{1}{2}(Q_{t-1} + D_{t-1}) + \left(1152\gamma^4 H^4 L^2 + 72\gamma^2 E^2 H^2\right)$$
$$\quad \times \sum_{\tau=0}^{E-1}\left(\mathbb{E}\big\|\nabla f(\hat{\boldsymbol{x}}^{t-1,\tau})\big\|^2 + \mathbb{E}\big\|\nabla f(\hat{\boldsymbol{x}}^{t,\tau})\big\|^2\right) + \phi(\gamma, \sigma^2),$$

where

$$\phi(\gamma, \sigma^2) = \left(1728\gamma^4 H^4 L^2 + 216\gamma^2 E^2 H^2\right)\frac{1}{H}\frac{1}{N}\sum_{j=1}^{N}\frac{1}{n_j}\sigma^2 + 144\gamma^4 EH^3 L^2 \frac{1}{N}\sum_{j=1}^{N}\frac{1}{n_j}\sigma^2$$
$$\quad + 42\gamma^2 EH\sigma^2 + 18\gamma^2 H^2\sigma^2 + 5\gamma^2 E^3 H\frac{1}{N}\sum_{j=1}^{N}\frac{1}{n_j}\sigma^2.$$

Further, we establish an upper bound for $\phi(\gamma, \sigma^2)$ as follows,

$$\phi(\gamma, \sigma^2) = 1728\gamma^4 H^3 L^2 \frac{1}{\bar{n}}\sigma^2 + 144\gamma^4 \frac{1}{\bar{n}}EH^3 L^2\sigma^2 + \gamma^2\sigma^2\left\{216\frac{E^2 H}{\bar{n}} + 42EH + 18H^2 + 5\frac{E^3 H}{\bar{n}}\right\}$$
$$\quad \leq 1728\gamma^4 EH^3 L^2 \frac{1}{\bar{n}}\sigma^2 + 144\gamma^4 \frac{1}{\bar{n}}EH^3 L^2\sigma^2 + 281\gamma^2 E^3 H\sigma^2$$
$$\quad \leq 294\gamma^2 E^3 H\sigma^2,$$

where the last inequality holds due to $\frac{1}{\bar{n}} = \frac{1}{N}\sum_{j=1}^{N}\frac{1}{n_j} \leq 1$ and $\gamma \leq \frac{\sqrt{\bar{n}}E}{12HL} \leq \frac{1}{33EHL}$.

**Part II** ($t = 0$): Summing (45) with the inequality established in Lemma F.2.3 and utilizing the upper bound of $\sum_{e=0}^{E-1} \frac{1}{N} \sum_{j=1}^{N} Y_j^{t,e}$ established in Lemma F.2.5, we have

$$\Gamma_0 \leq \left(48\gamma^2 H^2 L^2 + 384\gamma^4 H^4 L^4 + 24\gamma^2 E^2 H^2 L^2\right)\Gamma_0 + \left(384\gamma^4 H^4 L^2 + 24\gamma^2 H^2\right)\sum_{e=0}^{E-1} \mathbb{E}\left\|\nabla f\left(\hat{\boldsymbol{x}}^{0,e}\right)\right\|^2$$

$$+ \left(384\gamma^4 H^4 L^2 + 12\gamma^2 H^2 + 12\gamma^2 E^2 H^2\right)\left\{E\frac{1}{N}\sum_{j=1}^{N}\frac{1}{n_j}\sigma^2 + 6\gamma^2 L^4 E^2 H^2 \Gamma_1\right.$$

$$\left. + 6\gamma^2 L^2 E^2 H^2 \sum_{e=0}^{E-1} \mathbb{E}\left\|\nabla f(\hat{\boldsymbol{x}}^{0,e})\right\|^2 + 2\gamma^2 L^2 \frac{E^2 H}{N^2}\sum_{j=1}^{N}\frac{1}{n_j}\sigma^2\right\}$$

$$+ 96\gamma^4 E H^3 L^2 \frac{1}{N}\sum_{j=1}^{N}\frac{1}{n_j}\sigma^2 + 27\gamma^2 E H \sigma^2 + 12\gamma^2 H^2 \sigma^2 + 3\gamma^2 E^3 H\frac{N-1}{N^2}\sum_{j=1}^{N}\frac{1}{n_j}\sigma^2.$$

Reorganizing the above inequality gives rise to

$$\Gamma_0 \leq \left(50\gamma^2 H^2 L^2 + 432\gamma^4 H^4 L^4 + 26\gamma^2 E^2 H^2 L^2\right)\Gamma_0 + \left(26\gamma^2 H^2 + 432\gamma^4 H^4 L^2 + 2\gamma^2 E^2 H^2\right)\sum_{e=0}^{E-1} \mathbb{E}\left\|\nabla f\left(\hat{\boldsymbol{x}}^{0,e}\right)\right\|^2$$

$$+ \left(384\gamma^4 H^4 L^2 + 12\gamma^2 H^2 + 12\gamma^2 E^2 H^2\right)\left\{E\frac{1}{N}\sum_{j=1}^{N}\frac{1}{n_j}\sigma^2 + 2\gamma^2 L^2 \frac{E^2 H}{N^2}\sum_{j=1}^{N}\frac{1}{n_j}\sigma^2\right\}$$

$$+ 96\gamma^4 E H^3 L^2 \frac{1}{N}\sum_{j=1}^{N}\frac{1}{n_j}\sigma^2 + 27\gamma^2 E H \sigma^2 + 12\gamma^2 H^2 \sigma^2 + 3\gamma^2 E^3 H\frac{1}{N}\sum_{j=1}^{N}\frac{1}{n_j}\sigma^2.$$

Following the same derivation as in Part I, we obtain

$$\Gamma_0 \leq \left(648\gamma^4 H^4 L^2 + 42\gamma^2 E^2 H^2\right)\sum_{e=0}^{E-1} \mathbb{E}\left\|\nabla f\left(\hat{\boldsymbol{x}}^{0,e}\right)\right\|^2 + \psi(\gamma, \sigma^2).$$

where

$$\psi(\gamma, \sigma^2) = \left(1044\gamma^4 H^4 L^2 + 72\gamma^2 E^2 H^2\right)E\frac{1}{N}\sum_{j=1}^{N}\frac{1}{n_j}\sigma^2 + 144\gamma^4 E H^3 L^2 \frac{1}{N}\sum_{j=1}^{N}\frac{1}{n_j}\sigma^2$$

$$+ 42\gamma^2 E H \sigma^2 + 18\gamma^2 H^2 \sigma^2 + 5\gamma^2 E^3 H\frac{1}{N}\sum_{j=1}^{N}\frac{1}{n_j}\sigma^2.$$

Utilizing $\frac{1}{\bar{n}} = \frac{1}{N}\sum_{j=1}^{N}\frac{1}{n_j} \leq 1$ and $\gamma \leq \frac{\sqrt{\bar{n}}E}{12HL} \leq \frac{1}{33EHL}$, we can further bound $\psi(\gamma, \sigma^2)$ as

$$\psi(\gamma, \sigma^2) = \left(1044\gamma^4 H^4 L^2 + 72\gamma^2 E^2 H^2\right)E\frac{1}{N}\sum_{j=1}^{N}\frac{1}{n_j}\sigma^2 + 144\gamma^4 E H^3 L^2 \frac{1}{N}\sum_{j=1}^{N}\frac{1}{n_j}\sigma^2$$

$$+ 42\gamma^2 E H \sigma^2 + 18\gamma^2 H^2 \sigma^2 + 5\gamma^2 E^3 H\frac{1}{N}\sum_{j=1}^{N}\frac{1}{n_j}\sigma^2$$

$$= 1044\gamma^4 E H^4 L^2 \frac{1}{\bar{n}}\sigma^2 + 144\gamma^4 L^2 E H^3 \frac{1}{\bar{n}}\sigma^2 + \gamma^2 \sigma^2 \left\{72\frac{E^3 H^2}{\bar{n}} + 42EH + 18H^2 + 5\frac{E^3 H}{\bar{n}}\right\}$$

$$\leq 146\gamma^2 E^3 H^2 \sigma^2.$$

To this end, we complete the proof of Lemma F.2.7.

## G   Recovering SCAFFOLD's Results

By comparing the communication complexity $T$ required to achieve a $\epsilon$-stationary solution, we can see that our result recovers that of SCAFFOLD when $N = 1$ and $E = 1$. Specifically, for our MTGC algorithm, to achieve an $\epsilon$ error bound, according to Corollary 4.1, we can find a $T$ to satisfy

$$\sqrt{\frac{\mathcal{F}_0 L\sigma^2}{\tilde{N}TEH}} \leq \frac{\epsilon}{3}, \quad \left(\frac{\mathcal{F}_0 L\sigma}{T}\right)^{\frac{2}{3}} \leq \frac{\epsilon}{3}, \quad \frac{L\mathcal{F}_0}{T} \leq \frac{\epsilon}{3}.$$

Equivalently, $T \geq \frac{L\sigma^2 \mathcal{F}_0}{\tilde{N}EH\epsilon^2}$, $T \geq \frac{L\sigma \mathcal{F}_0}{(\epsilon)^{\frac{3}{2}}}$, and $T \geq \frac{L\mathcal{F}_0}{\epsilon}$. In other words, the MTGC algorithm will have an expected error smaller than $\epsilon$ if $T$ satisfies

$$T = \mathcal{O}\left(\frac{L\sigma^2 \mathcal{F}_0}{\tilde{N}EH\epsilon^2} + \frac{L\sigma \mathcal{F}_0}{(\epsilon)^{\frac{3}{2}}} + \frac{L\mathcal{F}_0}{\epsilon}\right).$$

According to Theorem II of [18], to achieve the $\epsilon$ error bound, the number of global communication rounds SCAFFOLD needs to take can be expressed as

$$T = \mathcal{O}\left(\frac{L\sigma^2 \mathcal{F}_0}{n_j H\epsilon^2} + \frac{L\mathcal{F}_0}{\epsilon}\right),$$

where we have converted the notation from [18] to our notation.

We see that the dominating term of the MTGC, $\mathcal{O}\left(\frac{L\sigma^2 \mathcal{F}_0}{\tilde{N}EH\epsilon^2}\right)$, recovers that of SCAFFOLD when $N = 1$ (i.e., $\tilde{N} = n_j$) and $E = 1$, which corresponds to the case of a single group with a single (global) aggregator.

