# OpenReview forum: "Hierarchical Federated Learning with Multi-Timescale Gradient Correction"
_NeurIPS.cc/2024/Conference — NeurIPS 2024 poster_

### Official Review · Reviewer_zFr1 · 2024-06-27

**Soundness:** 2
**Presentation:** 2
**Contribution:** 2
**Rating:** 6
**Confidence:** 3

**Summary:**

This paper proposes a multi-timescale gradient correction (MTGC) methodology to deal with multi-timescale model drift. It introduces a distinct control variables to correct the client gradient towards the group gradient, and correct the group gradient towards the global gradient. Then, the stability of the proposed algorithm against multi-level non-i.i.d. data is shown empirically.

**Strengths:**

1. The idea of correcting gradients is interesting
2. The proof seems correct.

**Weaknesses:**

The model is tested on small datasets.

**Questions:**

1. What are the practical applications of HFL?
2. Are there any techniques implemented for preserving the privacy of the model?
3. The model has been tested on small datasets; how does it perform on larger datasets?
4. The ablation study is missing. How does the model perform when only one of the corrections is used?
5. How about the stability of the proposed method?

**Limitations:**

Yes

---

> ### Author Rebuttal · Authors · 2024-08-07
>
> We appreciate the reviewer for the valuable comments. We also thank the reviewer for acknowledging our idea as interesting and for taking a look at our proof. Our responses are given below.
>
> ### **Practical applications of HFL**
>
> HFL can play a key role in most of the real-world networks comprised of a hierarchical architecture such as edge/fog computing systems or software-defined networks. In autonomous vehicle applications, each small base station (local aggregator) aggregates the models sent from the vehicles (nodes) in its coverage region, and then sends the aggregated model to the macro base station (global aggregator). Other applications of Internet
> of Things (IoT) sensors also operate under similar settings.  In healthcare applications, individual hospitals (nodes) send the model to regional health centers (local
> aggregators), and the national health authority (global aggregator) aggregates all the models. This HFL process involves nation wide data points, improving the performance of disease diagnosis models. Finally, in financial fields, individual financial institutions (nodes) collect transaction data and train local models for fraud detection and credit scoring. These models are aggregated at regional financial hubs (local aggregators), and the aggregated models are then sent to national or global financial authority  (global aggregator) to create a global model. We will illustrate these key applications more clearly in the revised paper.
>
> ### **Implementation of privacy-preserving techniques**
>
> Since our primary focus was to tackle the data heterogeneity issue from the perspective of designing optimization algorithms in the hierarchical setting, we have not considered applying additional techniques like privacy preservation that are orthogonal to our approach. Nonetheless, existing privacy-preserving techniques such as secure aggregation or differential privacy (DP) can be incorporated into our framework. To be specific, secure aggregation can be directly applied during the local aggregation process, as the local aggregators only need to obtain the average of
> models within its coverage rather than individual local models. Similarly, secure aggregation can be directly applied during the global aggregation process to preserve privacy as well.
>
> For DP, we can apply noise injection to local and global model aggregation (Steps 8 and 10 of Algorithm 1 in the original manuscript) to protect privacy. For instance, in Step 8, clients upload local models to the group server. In practical implementation, they only need to upload the model update
> $\Delta_i = \boldsymbol{x}_{i,H}^{t,e} - \bar{\boldsymbol x}_j^{t,e}$.
> For applying Gaussian noise to protect the model's privacy, we first clip $\Delta_i$ to ensure that its norm does not exceed a predefined threshold $C$:
> $
> \tilde{\Delta}_i = \Delta_i \cdot \min\left(1, \frac{C}{||\Delta_i||_2}\right).
> $
> Next, we can apply Gaussian noise to the clipped model update
>  $
> \hat{\Delta}_i = \tilde{\Delta}_i + \mathcal{N}(0, \sigma^2 I).
> $
> Here, $\mathcal{N}(0, \sigma^2 I)$ represents Gaussian noise with mean 0 and covariance matrix $\sigma^2 I$. Note that the procedures of applying the DP mechanism to MTGC are similar to FedAvg [R1]. The specific analysis of the DP-assisted MTGC is beyond the scope of our work and is best left for future research.
>
> [R1] Understanding clipping for federated learning: Convergence and client-level differential privacy, ICML 2022.
>
>
> ### **Experiments on larger datasets**
>
> Please refer to the global response.
>
> ### **Ablation study with only one correction term**
>
> In Figure 4 of our original manuscript, we have already reported the results of using only one of the correction terms. Specifically, we applied either the local correction, $\boldsymbol z_i^{t,e}$, or the group correction, $\boldsymbol y_j^{t}$, to HFedAvg. It is observed that in the group i.i.d. \& client non-i.i.d. scenario, local correction performs better than the group correction, as data samples are non-i.i.d. across the clients.
> Conversely, in the group non-i.i.d. \& clients i.i.d. scenario, the opposite holds. More detailed discussions on these results are illustrated in lines 311-327 of our original manuscript.
>
> ### **Stability of our method**
>
> In the original manuscript, we proved the convergence of MTGC without relying on extra data similarity assumptions. In the simulation, we show the stability of our algorithm in 4 different ways: We conducted experiments (i) using 3 random realizations (with different initial models) in Figure 3, (ii) under 3 different data distribution settings in Figure 4, (iii) using 4 different datasets, (iv) using varying parameters in Table 5.1 and Figures 5 and 6. During the rebuttal period, we have also added two more datasets (Shakespeare and CINIC-10) to further validate the advantage of our method. Moreover, we ran experiments with four more random realizations corresponding to Figure 4. Due to space limitations, we have selected one representative figure for CIFAR-10 to show for the rebuttal. Please refer to **Figure R4 (c)** of the attached pdf.
> Our MTGC consistently outperforms baselines across all scenarios and the performance gains beyond the standard deviation, showcasing its stability.
>
> Again, we appreciate the reviewer for the helpful comments. In case there are remaining questions/concerns, we hope to be able to have an opportunity to further answer them.

---

> > ### Comment · Reviewer_zFr1 · 2024-08-09
> > **Hierarchical Federated Learning with Multi-Timescale Gradient Correction**
> >
> > I thank the authors for their detailed rebuttal.
> >
> > I appreciate the discussion on privacy-preserving methods. Although the primary focus of this study is to tackle the data heterogeneity issue, it is essential to recognize that this is a federated learning model, and the privacy of the data must be preserved. I assume that applying the methods mentioned by the authors could potentially degrade the model's performance.
> >
> > I've read the comments and feedback from the other reviewers and look forward to the discussions following the rebuttal. I will keep my score for now.

---

> ### Author Response · Authors · 2024-08-11
>
> Thanks for your feedback. We appreciate your time for the discussion. We agree with the reviewer that preserving privacy of clients is crucial in FL. Here, we would like to emphasize that implementing additional privacy-preserving techniques on top of each scheme would degrade not only the accuracy of our approach, but also that of the baselines (they also were not integrated with privacy-preserving techniques in their original work).
>
> To demonstrate this, we have conducted an experiment to compare the performance of our algorithm with HFedAvg under the same $(\epsilon, \delta)$-differential privacy (DP) guarantees. We apply DP to the gradient to ensure sample-level privacy.
> To implement DP, we employ gradient clipping followed by the addition of Gaussian noise, a standard practice in the literature [R1, R2]. This approach is applied to both our algorithm and the baseline.
>
> Specifically, to guarantee DP, we replace step 7 of Algorithm 1 in the manuscript with the following procedures:
>
> + Compute stochastic gradient: $g_{i,m}^{t,e,h}, m=1,\ldots B_s$, $\forall i $
>
> + Gradient clipping: $\hat{g}\_{i,m}^{t,e,h} = g_{i,m}^{t,e,h} \cdot \min\left(1, \frac{c}{||g_{i,m}^{t,e,h}||}\right), m=1,\ldots B_s$, $\forall i $
>
> + Applying Gaussian noise to the gradient: $G_i^{t,e,h} = \frac{1}{B_s} (\sum_{m=1}^{B_s} \hat{g}_i^{t,e,h} + \mathcal{N}(0,\sigma_g^2))$, $\forall i $
>
> + Local model update: $ \boldsymbol x_{i,h+1}^{t,e}= \boldsymbol x_{i,h}^{t,e} - \gamma( G_i^{t,e,h} +  \boldsymbol z_i^{t,e}  +  \boldsymbol y\_j^t), \forall i  \in   \mathcal{C}\_j,\forall j$
>
> To achieve $(\epsilon, \delta)$ privacy, the noise should be set at the following scale [R1, R2]:
> $$
> n \sim \mathcal{N}(0,\sigma_{g}^2) ~~ \text{while} ~~
> \sigma\_{g}^2 = \frac{c^2 \log (1 / \delta) THE}{(|\mathcal{D}\_i|/B_s)^2 \epsilon^2},
> $$
> where $B_s$ denotes the batch size.
>
> The table below shows the results using the Shakespeare and CIFAR-100 datasets. For the Shakespeare task, we set $T=30, E=30, H=35$, $c=1$, $\delta = 10^{-3}$, $|\mathcal{D}\_i| = 1500$, and $B\_s = 200$. We compare their performance under a privacy budget of $\epsilon = 15$.
> The standard deviation of the Gaussian noise is thus $\sigma_{g} = 1.37$. For CIFAR-100, we set $T=50, E=20, H=20$, $c=10$, $\delta = 10^{-3}$, $|\mathcal{D}\_i| = 500$, and $B\_s = 50$. Under the privacy budget of $\epsilon = 15$, the standard deviation of the Gaussian noise is $\sigma\_{g} = 1.63$.
>
>
> | Dataset  | MTGC  (w/o DP) | MTGC-DP | HFedAvg  (w/o DP) | HFedAvg-DP |
> |-------------------|-------|---------|---------|------------|
> | Shakespeare    | 46.42 | 45.50   | 43.16   | 42.95      |
> | CIFAR-100       | 53.53 | 50.72   | 41.69   | 39.70      |
>
> We see that, as expected, the noise injection process for guaranteeing DP slightly decreases the accuracy of both schemes. Importantly, MTGC performs better than HFedAvg with and without DP. We will include these new results in the revised version of our manuscript.
>
> [R1] Abadi, Martin, et al. Deep learning with differential privacy. ACM SIGSAC, 2016.
>
> [R2] Li, Bo, et al. An improved analysis of per-sample and per-update clipping in federated learning. ICLR, 2024

---

### Official Review · Reviewer_fhg9 · 2024-07-15

**Soundness:** 3
**Presentation:** 3
**Contribution:** 2
**Rating:** 6
**Confidence:** 3

**Summary:**

This paper proposes a novel algorithm called Multi-Timescale Gradient Correction (MTGC) for Hierarchical Federated Learning (HFL). The authors address the challenge of multi-timescale model drift in HFL systems, where data heterogeneity exists both within client groups and across groups. MTGC introduces coupled gradient correction terms to mitigate client model drift and group model drift. The authors provide theoretical convergence analysis for non-convex settings and demonstrate MTGC's effectiveness through experiments on various datasets and models.

**Strengths:**

- The proposed MTGC algorithm is simple and easy to implement, introducing client-group and group-global correction terms. The paper clearly defines the problem it aims to solve. The motivation is clear and well-articulated, highlighting the gap in existing HFL algorithms.
- The inclusion of theoretical analysis adds depth to the paper, providing a solid foundation for the proposed approach. The theoretical results show that MTGC achieves linear speedup in the number of local iterations, group aggregations, and clients.
- The convergence bound of the proposed algorithm is immune to the extent of data heterogeneity, which is a significant strength.

**Weaknesses:**

- There is a lack of comparisons and discussions on clustered federated learning [1,2], which share similar context to some extent and could provide valuable background for the readers.


- The experiments primarily focus on image classification tasks. Including other types of tasks (e.g., natural language processing) could strengthen the generalizability of the results. There is a lack of experiments on various types of distribution shifts such as domain shift, label shift etc. Exploring different non-i.i.d. scenarios could provide more insights into the algorithm's robustness.

- While the paper analyzes model drift theoretically, there is a lack of empirical and theoretical analysis on how model drift affects generalization performance, especially in relation to the number of hierarchical communication levels. The study also does not clearly demonstrate in which scenarios and to what extent increasing the number of hierarchical communication levels benefits federated learning performance.

- The paper could also be strengthened by providing more insights into the practical implications of the theoretical results.

[1] An Efficient Framework for Clustered Federated Learning. NeurIPS 2020.

[2] Optimizing the Collaboration Structure in Cross-Silo Federated Learning. ICML 2023

**Questions:**

Please refer to the weaknesses mentioned above.

**Limitations:**

The limitations of the study are discussed, and no potential negative societal impacts of the work have been identified.

---

> ### Author Rebuttal · Authors · 2024-08-07
>
> We appreciate the reviewer's positive comments and feedback. We are glad to receive your appreciation of our motivation and results. Our responses are below:
>
> ### **Discussion on CFL**
>
> Our work focuses on HFL, employing a multi-layered structure consisting of local nodes, local aggregators, and a central server. Both clustered FL and HFL aim to improve FL learning efficiency by leveraging structured client groupings. The difference between them lies in the grouping criteria. HFL focuses on collaborative training over a given network topology, where clients are generally grouped based on their geographical location or network connection status, and aims to build a single global model under this setting. CFL groups clients to optimize model training, with different global models constructed depending on the group. [R1] demonstrates how dynamic clustering based on data distributions can enhance model performance. [R2] explores alleviating negative transfer from collaboration by clustering clients into non-overlapping coalitions based on their distribution distances and data quantities.
>
> We will add this to our updated manuscript.
>
> ### **Experiments on other types of tasks**
>
> Please refer to the global response.
>
>
> ### **Experiments on distribution shifts**
>
> Please refer to the global response.
>
> ### **Generalization bound**
> During the rebuttal, inspired by the paper [R2] shared by the reviewer, we studied the generalization bound of our algorithm as follows:
>
> Given a model $\boldsymbol{x}$, we denote the expected risk, defined on $\mathcal{D}$, as $\mathcal{R}(\boldsymbol{x})$.
> In the training stage, we aim to minimize the empirical loss $f(\boldsymbol{x})$ defined on finite samples $\hat{\mathcal{D}}$. The generalization error $\mathcal{R}(\boldsymbol{x}^t)-\mathcal{R}(\boldsymbol{x}^*)$, $\boldsymbol{x}^* = \arg\min \; \mathcal{R}(\boldsymbol{x})$, can be expressed as
> $$
> \mathcal{R}(\boldsymbol{x}^t)-\mathcal{R}(\boldsymbol{x}^*) = [\mathcal{R}(\boldsymbol{x}^t)-f(\boldsymbol{x}^t)]
>  +[f(\boldsymbol{x}^t)-f(\hat{\boldsymbol{x}}^*)]
>  +[f(\hat{\boldsymbol{x}}^*)-f(\boldsymbol{x}^*)]
>  +[f(\boldsymbol{x}^*) - \mathcal{R}(\boldsymbol{x}^*)],
> $$
> where $\hat{\boldsymbol{x}}^* = \arg\min f(\boldsymbol{x})$. According to [R2], one can claim that for any $\delta \in (0,1)$, with probability at least $1-\delta$, there is a quantity-aware function $\phi$ such that
> $$
> |\mathcal{R}(\boldsymbol{x})-\mathcal{R}(\boldsymbol{x})| \leq \phi (|\hat{D}|,\delta), \forall \boldsymbol{x}.
> $$
>
> Therefore, with probability at least $1-\delta$, we have
> $$
> \mathcal{R}(\boldsymbol{x}^t)-\mathcal{R}(\boldsymbol{x}^*) \leq 2 \phi (|\hat{D}|,\delta)
>  +f(\boldsymbol{x}^t)-f(\hat{\boldsymbol{x}}^*).
> $$
> Note that this bound is influenced by $\phi (|\hat{D}|,\delta)$ and $f(\boldsymbol{x}^t)-f(\hat{\boldsymbol{x}}^*)$. The former is determined by $\hat{D}$ while the latter will be impacted by the training algorithm, local data distribution, and network topology.
>
> In our manuscript, we characterize the upper bound of $||\nabla f(\boldsymbol{x})||^2$ rather than $f(\boldsymbol{x}^t)-f(\hat{\boldsymbol{x}}^*)$ as we focus on non-convex setting. For illustration, suppose we convert the upper bound for $||\nabla f(\boldsymbol{x})||^2$ to $f(\boldsymbol{x}^t)-f(\hat{\boldsymbol{x}}^*)$ by assuming the PL condition $f(\boldsymbol{x}^t)-f(\hat{\boldsymbol{x}}^*) \leq \frac{1}{2\mu} ||\nabla f(\boldsymbol{x})||^2$. This gives a generalization error bound
> $$
> 2 \phi (|\hat{D}|,\delta) + \frac{\Delta}{2\mu},
> $$
> where $\Delta$ is the error derived in Corollary 4.1.
>
> Different from CFL [R2] which tries to approximately minimize $\phi (|\hat{D}|,\delta)$ by clustering clients, i.e, optimizing $\hat{D}$, in our setting, $|\hat{D}|$ is fixed, and thus $\phi (|\hat{D}|,\delta)$ is fixed.
>
> In addition, regarding the impact of model drift on the generalization bound: Unlike CFL, where the clients end up with different models, in HFL, model drift will converge to zero at the end of training for MTGC due to its convergence. Model drift is a characterization metric used in the intermediate process for convergence analysis. As you can see, there is no model drift term in our generalization bound.
>
>
> ### **Increasing hierarchical communication levels**
>
> Note that the main objective of our paper is to develop a convergent algorithm for a given HFL topology, treating the number of layers as an intrinsic system parameter. Hence, we predominantly assume the topology to be predetermined in our work. To address this comment, during the rebuttal period, we empirically studied the impact of hierarchical levels on FL performance, in terms of testing accuracy versus communication time. We find that the benefit of additional layers varies based on the configuration. Please refer to the global response for the experimental details.
>
>
> ### **Practical implications of the theoretical results**
>
> Thanks for the comment.  As shown in Corollary 4.1, the upper bound is mainly dominated by the first term $\mathcal{O}\left(\sqrt{\frac{\mathcal{F}_0L\sigma^2}{\tilde{N} T EH}} \right)$. Due to the speedup in the number of local iterations $H$ and the number of group aggregations  $E$, we can reduce the global communication round $T$ by increasing $H$ and $E$ in practice. We empirically validate this in Table 5.1 in the original manuscript and Figure 6 in the Appendix, connecting theory and practice. On the other hand notice that there is an upper bound for the learning rate, i.e., $\gamma \leq \frac{1}{40EHL}$. When we increase $E$ and $H$, the upper bound of the optimal learning rate will decrease. We will highlight these insights after Corollary 4.1 in the revised manuscript.
>
> [R1] An efficient framework for clustered federated learning. NeurIPS 2020.
>
> [R2] Optimizing the collaboration structure in cross-silo federated learning. ICML 2023.
>
> Again, thanks for your comments. If we could provide any more clarifications, we would be grateful if you could let us know.

---

> ### Comment · Area_Chair_XPt5 · 2024-08-12
>
> Dear Reviewer fhg9,
>
> Could you please respond with how you think about the authors' response? Please at least indicate that you have read their responses.
>
> Thank you,
> Area chair

---

### Official Review · Reviewer_r1S6 · 2024-07-16

**Soundness:** 3
**Presentation:** 3
**Contribution:** 2
**Rating:** 6
**Confidence:** 3

**Summary:**

This paper presents a method to address multi-timescale model drift in hierarchical federated learning. Specifically, it introduces two control variables to correct intra-group client drift and group model drift. The paper establishes the convergence bound in a non-convex setup and demonstrates its stability against multi-level data heterogeneity. Overall, the paper is well-written and easy to follow.

**Strengths:**

1. The writing is clear and easy to follow.
2. I have initially checked the proofs and did not find any issues so far.

**Weaknesses:**

1. It would be beneficial to include comparisons of additional computational and communication costs of MTGC in the experimental section.

**Questions:**

1. In Algorithm 1, consider using different color blocks to mark $x$, $z$, and $y$. This could help in better understanding the algorithm's process.

2. How about the runtime of the algorithm? Would it be possible to include comparisons of runtime with the baseline in the experiments?

3. In the experiments, does Eq.5 require the addition of regularization coefficients for correction terms $z$ and $y$?

4. Could you include some measurements of the additional communication costs in the experiments?

**Limitations:**

N.A.

---

> ### Author Rebuttal · Authors · 2024-08-06
>
> We appreciate the reviewer's constructive feedback. We appreciate the reviewer for carefully checking our proof and are glad to receive your positive feedback on the writing of our paper. Our responses are given as below:
>
> ### **Communication cost comparison**
>
> Compared to HFedAvg, MTGC requires initializing the correction variables at the start of each global round, which incurs additional communication overhead. Specifically, for every $E$ steps of group aggregation, MTGC incurs an additional communication cost equivalent to one transmission of the model parameters. In other words, **the per-aggregation communication complexity of MTGC is $\frac{E+1}{E}$ times that of HFedAvg**.
> To show this impact, we have added experiments comparing the communication cost and testing accuracy at the client side. This experiment was conducted on CIFAR-10 dataset with $E = 30$ and $H = 20$ under both client and group non-i.i.d. setup. The model and other parameters are the same as in the original manuscript.
> The results are shown in **Figure R1(a)** of the attached PDF. The results demonstrate that MTGC achieves higher testing accuracy for a given communication cost, highlighting the efficiency and effectiveness of our approach.
>
> ### **Running time and computational cost**
>
> During the rebuttal phase, we compared the runtime of our MTGC algorithm with the baselines. Using one NVIDIA A100 GPU with 40 GB memory, we conducted experiments on the CIFAR-10 dataset with $E = 30$ and $H = 20$ under client/group non-i.i.d. setup. The model and other parameters are the same as in the original manuscript. We report the required time for attaining a preset accuracy of $75 \\%$ and for running $100$ global rounds in **Figure R1(b)** of the attached PDF of the global response. We see that although our approach incurs extra operation induced by the correction variables, this is the cost for achieving a significant performance improvement by effectively handling the data heterogeneity problem. We also note that the computation cost incurred by the correction variable is relatively small compared to computing gradients in a neural network using backpropagation, which is a step that is required in all methods. Overall, the results confirm the advantage of our method compared with baselines.
>
> ### **Distinguishing the update of $  \boldsymbol{x}$, $ \boldsymbol{z}$ and $ \boldsymbol{y}$**
>
> Thanks for your kind reminder. We will highlight the updates of three variables with different colors in the future version.
>
> ### **Does Eq.5 require the addition of regularization coefficients for correction terms $ \boldsymbol{ z}$ and $ \boldsymbol{y}$?**
>
> Based on our theory and experiments, we don't need to apply coefficients to $\boldsymbol{z}$ and $\boldsymbol{y}$.
> For ease of explanation, we state equation (5) as follows:
>
> $$ {\boldsymbol{x}}^{t,e}_{i,h+1} = {\boldsymbol{x}}^{t,e}\_{i,h}  - \gamma \left( \nabla F_i({\boldsymbol{x}}^{t,e}\_{i,h}, \xi\_{i,h}^{t, e}) + \boldsymbol{z\_i}^{t,e} + \boldsymbol{y\_j}^t \right) $$
>
> $\boldsymbol{z}$ and $\boldsymbol{y}$ play the role of tracking the gradient difference between the local gradient and the group gradient and the difference between the group gradient and the global gradient. By utilizing $\boldsymbol{z}$ and $\boldsymbol{y}$, our aim is for the correction updating direction $\nabla F_i + \boldsymbol{z_i} + \boldsymbol{y_j} $ to approach the global gradient direction. On the other hand, if we apply coefficients $\lambda$ for $\boldsymbol{z}$ and $\boldsymbol{y}$, consider an ideal example where we have attained the optimal point $\boldsymbol{x^*}$ and
> $\boldsymbol{z_i} =  \nabla f_i(\boldsymbol{x^*}) - \nabla F_i(\boldsymbol{x^*})$ and $\boldsymbol{y_j }= \nabla f(\boldsymbol{ x^*}) - \nabla f_j(\boldsymbol{x^*}) $.
> If we apply a coefficient $\lambda$ into our iteration, equation (5) becomes
> $$\boldsymbol{x^*} - \gamma \left(\nabla F_i(\boldsymbol{x^*}) + \lambda \left(\nabla f_j(\boldsymbol{x^*}) - \nabla F_i(\boldsymbol{x^*}) +  \nabla f(\boldsymbol{x^*}) - \nabla f_j(\boldsymbol{x^*}) \right)  \right).$$
>
> This iteration is not stable at $x^*$ as $(1-\lambda) \nabla F_i(\boldsymbol{x^*})$ is generally not equal to zero unless $\lambda = 1$. Due to this, we did not apply coefficients to $\boldsymbol{z}$ and $\boldsymbol{y}$.
>
> Again, thank you for your time and efforts for reviewing our paper, and providing insightful comments. If there are any more clarifications we could provide, we would be grateful if you could let us know.

---

> ### Comment · Area_Chair_XPt5 · 2024-08-12
>
> Dear Reviewer r1S6,
>
> Could you please respond with how you think about the authors' response? Please at least indicate that you have read their responses.
>
> Thank you,
> Area chair

---

### Official Review · Reviewer_rcWg · 2024-07-29

**Soundness:** 3
**Presentation:** 3
**Contribution:** 1
**Rating:** 5
**Confidence:** 4

**Summary:**

This paper introduces the usage of the gradient correction scheme to hierarchical federated learning. Specifically, the authors propose and analyze the multi-timescale gradient correction MTGC algorithm which is a direct generalization of SCAFFOLD to the framework where local clients aggregate their models on group server and group servers aggregate their models to a global server. The authors introduce control variables that correct i) client model drift and ii) group model drift that arises due to data heterogeneity. Theoretical convergence results are presented in the non-convex regime and experimental results showcase the ability of MTGC to address data heterogeneity both from clients and from group-servers.

**Strengths:**

- The paper places itself correctly in the existing literature, is well-structured and easy to follow.
- The problem of hierarchical FL is relevant and fairly interesting for the ML community.
- The proposed algorithm is natural and easy to understand.
- Both theoretical and numerical results are provided.

**Weaknesses:**

- The main weakness of this work is its lack of novelty and technical contribution. The proposed MTGC is a straightforward extension of the well-known SCAFFOLD algorithm to a two-level hierarchical FL. Although the authors mention that the coupling of the two error correction variables is introducing new challenges I fail to see how this is the case. Indeed, the analysis of the correction variables appears to be largely the same as in SCAFFOLD and the authors to not make clear in the main body of their work what specific new challenges they faced and how they were circumvented. To summarize, although the paper is well written I do not see enough contribution to justify its acceptance to a top tier conference.

- Minor typo: on line 110 in "However, their is" should be "However, their algorithm is".

**Questions:**

See weaknesses section.

**Limitations:**

The authors adequately addressed the limitations  of their work.

---

> ### Author Rebuttal · Authors · 2024-08-06
>
> We appreciate the reviewer's constructive feedback. We are glad that the reviewer acknowledges the problem we studied is interesting.
>
> We would like to emphasize that theoretically showing that the proposed MTGC algorithm guarantees convergence and achieves linear speedup in terms of both $H$ (the number of local updates) and $E$ (the number of group aggregations) was quite non-trivial.
> In this response, we will provide more detailed descriptions of the theoretical challenges and our proof techniques to better highlight the contributions compared to SCAFFOLD, which was developed in a single-level scenario.
>
> ### **Difference in the analysis of the correction terms**
>
> In SCAFFOLD [12], the theoretical analysis of correction terms appears in the proof of Lemma 18.
> Here, the correction error can be easily bounded using $\beta$-smoothness:
> $$
> \mathbb{E}||c^{r-1}-\nabla f({x}^{r-1})||^2+ \frac{1}{N}\sum_{i=1}^N \mathbb{E}||c_i^{r-1}-\nabla f_i({x}^{r-1})||^2 \leq \frac{2 \beta^2}{K}\sum_{k=1}^K\mathbb{E}||y_{i, k-1}^{r-1}-x^{r-1}||^2.
> $$
>
> In our work (with multi-level, multi-timescale updates/aggregations), we
> bound the new correction errors
> $$
> Z\_j^{t,e} = \frac{1}{n\_j}\sum\_{i \in \mathcal{C}\_j} \mathbb{E} ||\boldsymbol{z}\_i^{t,e} + \nabla F\_i(\bar{\boldsymbol{x}}^{t,e}\_j) - \nabla f\_j(\bar{\boldsymbol{x}}^{t,e}\_j)||^2, \ \ \text{and} \ \ \
> Y\_j^{t,e} = \mathbb{E} || \boldsymbol{y}\_j^t + \nabla f_j(\hat{\boldsymbol{x}}^{t,e}) - \nabla f(\hat{\boldsymbol{x}}^{t,e})||^2
> $$
> in Lemmas C.2.4 and C.2.5,
> for characterizing client and group model drifts, respectively. As shown in the proof of Lemmas C.2.4 and C.2.5, the analysis of the correction variables in our work involves new difficulties. In particular, in the analysis of upper-level correction term $\boldsymbol{y}$, we need to bound the discrepancy between $\boldsymbol x_{i,h}^{t,\tau}$ and $\hat{\boldsymbol x}^{t+1,e}$ (see line 628 of our manuscript), where two models appear in two different global rounds. We note that different from SCAFFOLD, $\boldsymbol{y}_j^t$, $\nabla f_j(\hat{\boldsymbol{x}}^{t,e})$, and $\nabla f(\hat{\boldsymbol{x}}^{t,e})$ in $Y_j^{t,e}$ are in two different timescales. Moreover, the upper bound of $Y_j^{t,e}$ is influenced by the gradients and model drifts of two consecutive rounds, highlighting the increased complexity compared to the single-level case.
>
> ### **Detailed technical challenges**
>
> **First**, the upper bounds of correction errors $Y_j^{t,e}$ and $Z_j^{t,e}$ shown in Lemmas  C.2.4 and C.2.5, are impacted by model drifts $D_t$ and $Q_t$, and $\Theta_j^{t,e}$. Meanwhile, the upper bounds of $D_t$, $Q_t$, and $\Theta_j^{t,e}$ are also impacted by $Y_j^{t,e}$ and $Z_j^{t,e}$. The interplay between these terms makes the analysis non-trivial. **Second**, (i) global aggregation, (ii) the update of upper-level correction variable $\boldsymbol{y}$ and local aggregation, and (iii) the update of lower-level correction variable $\boldsymbol{z}$ are performed at different timescales. Note that in SCAFFOLD, since there is no local aggregation step, the convergence is presented using the global aggregation timescale, i.e., $\{\nabla f(\bar{\boldsymbol x}^{t})\}$. However, in MTGC, if we directly consider $\{\nabla f(\bar{\boldsymbol x}^{t})\}$, it is difficult to capture the effects of group aggregation and correction variable $\boldsymbol{z}$. Moreover, it is hard to establish a tight connection between $\nabla f(\bar{\boldsymbol x}^{t})$ and $||\boldsymbol x_{i,h}^{t,e} - \bar{\boldsymbol x}^{t}||$, $\forall e, h$ since there is a large lag between $ {\boldsymbol x}_{i,h}^{t,e}$ and $\bar{ \boldsymbol x}^{t}$.
>
> ### **Our approach**
>
> For the **first challenge**, we extracted a recursive relationship of $\Gamma_{t} = Q_t + D_t$ hidden behind Lemmas C.2.2-C.2.6, as summarized in Lemma C.2.7. With this recursion, we design a novel Lyapunov function as $\Phi_{t+1} = \mathbb{E} f(\bar{\boldsymbol x}^{t+1}) - f^* +\gamma L^2 H \Gamma_{t},$ to derive the recursive relationship between two global rounds. This new Lyapunov function with recursive components enabled us to mitigate the coupling effects between $Y_j^{t,e}$, $Z_j^{t,e}$ and $\Theta_j^{t,e}$ (see proof of Theorem 4.1 in our Appendix).
>
> For the **second challenge**, we introduce a new metric, which is the gradient $\nabla f(\hat{\boldsymbol x}^{t,e})$ at virtual sequence $\{\hat{\boldsymbol x}^{t,e}\}$, to characterize the convergence of MTGC. This introduction makes our analysis tractable by building connection between $\boldsymbol x_{i,h}^{t,e}$ and $\hat{\boldsymbol x}^{t,e}$ as follows: $\boldsymbol x\_{i,h}^{t,e} \rightarrow \bar{\boldsymbol x}^{t,e}\_j \rightarrow \hat{\boldsymbol x}^{t,e}$. Accordingly, we introduce two characterizations, $||\boldsymbol x_{i,h}^{t,e} - \bar{\boldsymbol x}_j^{t,e} ||^2$ and $||\hat{\boldsymbol x}^{t,e} - \bar{\boldsymbol x}_j^{t,e} ||^2$, to capture the local and group model drifts, respectively. The former quantifies the progress made by each client from $(t,e,0)$ to $(t,e,h)$-th iteration while the latter characterizes the group model deviation from the virtual global model at the $(t,e)$-th group aggregation, making our analysis distinct from SCAFFOLD.
>
> In the revised manuscript, we will illustrate these new challenges and our solution in more detail. Considering that the exploration of FL  under practical hierarchical setups is quite limited, we believe that our work presents a meaningful contribution to the community. Our work bridges this gap by developing the MTGC algorithm with desired theoretical properties (convergence guarantee with linear speedup in the numbers of clients, local updates, and edge aggregations), where the analysis from single-level to hierarchical is not a trivial extension as supported by our response above.
>
> ### **Typo**
>
> We double-checked the manuscript and corrected this typo.
>
> Again, thanks for your time and efforts! Please let us know if you need any further clarification.

---

> > ### Comment · Reviewer_rcWg · 2024-08-10
> > **Post Rebuttal**
> >
> > I appreciate the efforts of the authors to address my concerns in their rebuttal.
> >
> > After carefully reading the comments from the rest of the reviewers as well as the responses of the authors I have decided to increase my score. Specifically, the extensive description of the challenges and the techniques used to overcome these challenges (found in the rebuttal) are of crucial importance, helping the reader to better understand the technical contribution of this work. I believe that including this discussion in the main body of the paper will further improve its quality and presentation.
> >
> > That being said, I still consider the main weaknesses of this work to be its - somewhat limited - novelty and impact. Overall, I find this paper to be very close to the acceptance threshold leaning slightly towards acceptance (indicated by my updated score 5).

---

> > > ### Author Response · Authors · 2024-08-10
> > >
> > > We really appreciate your reply and score update. We will include these discussions in the main body of the revised manuscript. Thanks again for the helpful suggestion.

---

### Author Rebuttal · Authors · 2024-08-07

We appreciate all reviewers for providing constructive comments. In this global response, we will describe the additional experiments we have conducted, suggested by **Reviewer r1S6**, **Reviewer fhg9**, and **Reviewer zFr1**, that may be of interest to all reviewers. The figures are included in the attached PDF. The responses to all other comments are provided individually for each reviewer.

### **Experiments for Reviewer r1S6**

**Computation and communication costs**

We reported the computational and communication costs of MTGC and baselines in **Figure R1** in the attached PDF.

### **Experiments for Reviewer fhg9**

**Experiments on other types of tasks**

We have conducted additional experiments using the Shakespeare dataset, an NLP task. We use the LSTM model where the model starts with an $80$-character input sequence, which is converted to an $80 \times 8$ sequence through an embedding layer. This embedded sequence is then processed by a two-layer LSTM, each with $100$ hidden units. The final output is passed through a softmax layer to make predictions. The performance comparison is presented in **Figure R2(a)** of the attached pdf, where we set the learning rate $0.5$, $H=75$, and $E=30$. It is observed that our MTGC consistently outperforms baseline methods.

**Experiments on distribution shift datasets**

We have conducted additional experiments to include two different non-i.i.d. scenarios: label shift and feature shift, as referenced in [R1,R2]. These experiments were performed using the Fashion-MNIST dataset.

For **label shift** [R1,R2], we randomly assign 3 classes out of 10 classes to each group with a relatively balanced number of instances per class, and then assign 2 classes to each client. As discussed in [R1], label shift adds more heterogeneity to this system. According to the results shown in Figure R2(b), it is clear that the proposed algorithm is more robust against data heterogeneity. Specifically, there is less oscillation in MTGC compared with HFedAvg and the attained accuracy of MTGC in the given communication round is higher than all baselines.

For **feature shift** [R1], we first partition data following the group non-i.i.d. \& client non-i.i.d. case as in our original manuscript,  and then let clients at different groups rotate images for different angles. Concretely, for the clients at the $i$-th group, the angle is $-50+10 \times i$. Note that this rotation is only applied to the training set. The feature shift increases the diversity between the training set and the testing set, which thus adds difficulty to this classification task. In **Figure R2(c)**, we see that MTGC attains the best performance among these baselines.

[R1] Optimizing the collaboration structure in cross-silo federated learning, ICML 2023.

[R2] On the convergence of clustered federated learning, arXiv preprint 2022.

**Experiments on hierarchy levels**

To evaluate the impact of hierarchy levels, we compared the testing accuracy versus communication time. We considered three scenarios: single-level (100 clients), two-level (100 clients grouped into 10x10), and three-level (100 clients grouped into 4x5x5). The experiments were conducted using ResNet-18 (44.6 MB) and the CIFAR-10 dataset. We considered different cases of link bandwidths in each HFL architecture, which impact the speed of communication through the system.

The results are shown in **Figure R3** of the attached PDF. In **Figure R3(a)**, the communication bandwidths were set to 0.5 MB/s for single-level, and 0.6 MB/s and 20 MB/s for upper and lower links in two-level, and 0.7 MB/s, 5.5 MB/s, and 20 MB/s for three-level. Results show performance improvements from single-level to two-level, and from two-level to three-level.

In **Figure R3(b)**, considering a faster network, bandwidths were adjusted to 3 MB/s for single-level, and 3.5 MB/s and 20 MB/s for upper and lower links in two-level, and 4 MB/s, 5.5 MB/s, and 20 MB/s for three-level. Results indicated performance improvement from single-level to two-level but a decrease from two-level to three-level. This highlights that performance can vary with different configurations. Generally, increasing intermediate layers is beneficial when the central server is far from clients with high transmission latency.

### **Experiments for Reviewer zFr1**

**Experiments on larger datasets**

During the rebuttal, we have conducted additional experiments on the larger Shakespeare and CINIC-10 datasets.

For the **Shakespeare dataset**, we randomly pick 100 characters (people) in Shakespeare’s plays. We let each client have 1,500 samples, where each sample is a sequence of 80 characters (words). Considering that there are 100 clients in the system, there are 150,000 train samples in total. This means that the number of samples is 3 times that of CIFAR-10 (or CIFAR-100), which has 50,000 train samples. The performance comparison is presented in **Figure R4(a)** of the attached pdf, where we use the LSTM model and set the learning rate $0.5$, $H=75$, and $E=30$.
It is seen that MTGC consistently outperforms the baseline methods in larger datasets.

The **CINIC-10 dataset** contains 90,000 training images, 90,000 validation images, and 90,000 test images, significantly larger than CIFAR-10 and CIFAR-100 with 60,000 images. It includes images from both CIFAR-10 and ImageNet, enhancing diversity. We believe that the larger size and diversity of CINIC-10 further confirm the validity of our experiments. The model and hyperparameters used for the CINIC-10 dataset are the same as those of the CIFAR-10 task shown in the original manuscript. As illustrated in **Figure R4(b)** of the attached pdf, MTGC maintains its superior performance on the CINIC-10
dataset, consistent with its performance on other tasks.

---

### Decision · Program_Chairs · 2024-09-25

**Decision:**

Accept (poster)

**Comment:**

This paper consider a setting of FL where the clients have hierarchical architectures. Then they proposed a simple but effective algorithm for handling multi-timescale model drift based on multi-timescale gradient correction (MTGC) techniques. This paper is closely relevant to practices and fills the gap in existing literature on hierarchical FL. The theoretical results are strong and without any assumptions data heterogeneity. Also a linear speedup with the number of local iterations is obtained in theory.